# A model framework on atmosphere-snow water vapor exchange and the associated isotope effects at Dome Argus, Antarctica:part I the diurnal changes

Tianming Ma[1, 2], Zhuang Jiang[1], Minghu Ding[4, 3], Pengzhen He[5], Yuansheng Li[6], Wenqian Zhang[4] and Lei Geng[1, 3, 7]

[1] Deep Space Exploration Laboratory/School of Earth and Space Sciences, University of Science and Technology of China, Hefei 230026, China.

[2] School of Marine Science and Environment Engineering, Dalian Ocean University, Dalian 116023, China.

[3] State Key Laboratory of Cryospheric Science, Northwest Institute of Eco-Environment and Resources, Chinese Academy of Sciences, Lanzhou 730000, China.

[4] Chinese Academy of Meteorological Sciences, Beijing 100081, China.

[5] School of Environment and Tourism, West Anhui University, Lu'an, 237012, China.

[6] Polar Research Institute of China, Shanghai 200136, China.

[7] CAS Center for Excellence in Comparative Planetology, University of Science and Technology of China, Hefei 230026, Anhui, China.

*Correspondence to*: Lei Geng (genglei@ustc.edu.cn)

**Abstract.** Ice-core water isotopes contain valuable information on past climate changes. However, such information can be
altered by post-depositional processing after snow deposition. Atmosphere-snow water vapor exchange is one of such processes, but its influence remains poorly constrained. Here we constructed a box model to quantify the atmosphere-snow water vapor exchange fluxes and the associated isotope effects at sites with low snow accumulation rates where the effects of atmosphere-snow water vapor exchange are suspected to be large. The model reproduced the observed diurnal variations in $\delta^{18}O$, $\delta D$, and d-excess in water vapor at Dome C, East Antarctica. According to the same model framework, we found that
under average summer clear-sky conditions atmosphere-snow water vapor exchange at Dome A can cause diurnal variations in atmospheric water vapor $\delta^{18}O$ and $\delta D$ of 4.8±2.6 ‰ and 29±19 ‰, with corresponding diurnal variations in surface snow $\delta^{18}O$ and $\delta D$ of 0.80±0.35 ‰ and 1.6±2.7 ‰, respectively. The modeled results under summer cloudy conditions display similar patterns to those under clear-sky conditions but with much smaller magnitudes of diurnal variations. However, under winter conditions at Dome A, the model predicts little to no diurnal changes in snow isotopes, consistent with the stable
boundary condition in winter which inhibits effective vapor exchange between the atmosphere and snow. In addition, after 24-hours, and continuous simulations of 11 days, the model predicts significant enrichments in snow isotopes under summer conditions, while in winter the depletions also accumulate after each 24-hour simulation but with a much smaller magnitude of change compared to the results from summer simulations. If the modeled snow isotope enrichments in summer conditions and the depletions in winter conditions represent general situations at Dome A, this likely suggests that atmosphere-snow water
vapor exchange tends to increase snow isotope seasonality, and the annual net effect would be overall enrichments in snow

isotopes since the effects in summer appear to be greater than those in winter. This trend will need to be further explored in the future with more comprehensive model studies and/or field observations/experiments.

## 1 Introduction

Water stable isotopes ($\delta^{18}O$ and $\delta D$) in snow and rain are valuable proxies to inform atmospheric temperatures at the time of
precipitation (Craig, 1961; Dansgaard, 1964). In Antarctica, the isotopic composition of snowfall, as well as that of surface snow, is correlated with local air temperature (Fujita & Abe, 2006; Masson-Delmotte et al., 2008; Stenni et al., 2016). These findings permit past temperature reconstructions using ice-core $\delta^{18}O$ and $\delta D$ records across different time scales (e.g., from millennium to glacial-interglacial) (Petit et al., 1999; EPICA community members, 2004; WAIS Divide project members, 2013). Temperature information at shorter time scales (e.g., seasonal to decadal or longer) is critical for understanding climate
variabilities and probing driving forces, and thus many studies have focused on high-resolution temperature reconstructions using water isotope profiles (e.g., Stenni et al., 2017). However, there are an increasing number of observations indicating that air temperature and snow/ice-core water isotopes are not always covarying, especially at decadal or shorter timescales, and the disconnection is particularly obvious at low snow accumulation rate sites such as Vostok, Dome F and Dome C, Antarctica (Hoshina et al., 2014; Ekaykin et al., 2017; Casado et al., 2018). Such observations suggest changes in snow water isotopes
after deposition, which not only inhibits temperature reconstructions at decadal or shorter timescales using ice core $\delta^{18}O$ and/or $\delta D$ records, but also undermines reconstructions at longer timescales such as millennium and glacial-interglacial climate changes (Touzeau et al., 2016; Casado et al., 2018; Laepple et al., 2018; Markle & Steig, 2022).

It is well known that after snow deposition, a combination of post-depositional processes can induce significant changes in the water isotopic composition of snow (Steen-Larsen et al., 2013; Casado et al., 2018; Laepple et al., 2018). Such changes have
been demonstrated by the gradual weakening of snow isotope-temperature relationships as reflected by surface and buried snow samples (Casado et al., 2018). Atmosphere-snow water vapor exchange is one of such processes but there are only limited observations/modeling studies focusing on this process at the diurnal scale in polar summers (Ritter et al., 2016; Casado et al., 2018; Madsen et al., 2019; Hughes et al., 2021; Wahl et al., 2021; Hu et al., 2022; Wahl et al., 2022). The isotopic effects associated with atmosphere-snow water vapor exchange at longer time scales have been investigated at Greenland Ice Sheet
(Dietrich et al., 2023), but not yet in Antarctica.

Atmosphere-snow water vapor exchange is the snow sublimation-water vapor deposition cycle occurring at the atmosphere-snow interface. It is driven by near-surface vapor pressure gradients and influenced by temperature, wind speed, and humidity (Neumann et al., 2009; Sokratov & Golubev, 2009; Ritter et al., 2016; Wahl et al., 2021; Wahl et al., 2022). Dansgaard (1973) proposed that the layer-by-layer sublimation of snow and ice does not induce isotopic fractionation, but this was suggested to
be invalid based on laboratory experiments and field observations in which sublimation was found to modify surface snow isotopic compositions under natural conditions (Sokratov & Golubev, 2009; Ebner et al., 2017; Hughes et al., 2021; Wahl et al., 2021). Water vapor sublimated from snow can be transferred to the overlying atmosphere where it affects the atmospheric

water vapor concentration and isotopic composition. Moreover, the inverse part of sublimation, i.e., deposition, can also lead to changes in the isotopic composition of surface snow as well as atmospheric water vapor due to preferential deposition of heavy isotopes (e.g., $H_2^{18}O$ and HDO) (Wahl et al., 2021). Given fluctuations in surface temperature, humidity and other meteorological conditions, the relative degree of sublimation vs. deposition could vary, leading to variations in the isotopic compositions of surface snow and atmospheric boundary layer water vapor (Neumann et al., 2009; Sokratov & Golubev, 2009; Ritter et al., 2016; Wahl et al., 2021; Hughes et al., 2021; Wahl et al., 2022). Parallel variations in the isotopic composition of atmospheric water vapor and surface snow (0.2-1.5 cm depth) have been observed at multiple polar sites (e.g., Dome C, Kohnen station, NEEM, and EastGrip) in summer for short durations (Steen-Larsen et al., 2013; Casado et al., 2016; Casado et al., 2018; Madsen et al., 2019; Bréant et al., 2019), and such co-variations have been suggested to be due to atmosphere-snow water vapor exchange.

Given the difficulties in conducting continuous high-resolution observations in polar regions, a model frame describing the atmosphere-snow water vapor exchange processes and the associated isotope effects would be useful in terms of snow and ice-core water isotope record interpretation across different sites. Such models, if fully resolving the physical mechanisms of atmosphere-snow water vapor exchange processes with appropriate parameterizations, can be incorporated into snowpack and climate models to assess the effects of atmosphere-snow water vapor exchange on the preservation of snow water isotope signals. Several empirical models have been developed to evaluate the isotope effects of atmosphere-snow water vapor exchange. They incorporate atmospheric stratification and climatological boundary conditions to calculate water mass and isotope exchanges at the atmosphere-snow interface by assuming a closed system with a one-dimensional box model (Ritter et al., 2016; Casado et al., 2018; Pang et al., 2019).

As the interior dome of East Antarctica, Dome Argus (80.42°S, 77.12°E; 4093 m above sea level, Dome A hereafter) has a more southerly moisture source than other sites on the East Antarctic Plateau (Wang et al., 2012). This makes ice core records of water isotopes from Dome A special in terms of recording southern mid-altitude moisture influence. In addition, Dome A is a candidate site in the search for ancient ice with 1.0-1.5 million years old (Sun et al., 2008; Van Liefferinge et al., 2018). Since 2009, the Kunlun deep ice coring project has been conducted at Dome A. By the 2015/2016 field season, an 800-m ice core had been drilled (Hu et al., 2021), and a preliminary analysis of water isotopic records of the top 109 meters reflected a long-term cooling trend at Dome A over the last 2 kyr (Hou et al., 2012; Jiang et al., 2012; An et al., 2021). Given the extremely low snow accumulation rate (18-23 mm w. eq. y. from Ding et al., 2016) at Dome A, water isotopes preserved in firn and ice cores at this site are presumably influenced by post-depositional processing. In particular, the effects of atmosphere-snow water vapor exchange might become important, as snow can stay at the surface for a relatively long period. This characteristic not only means that water isotope records from Dome A should be carefully evaluated for the effects of atmosphere-snow water vapor exchange before interpretation, but also makes Dome A a promising site for elucidating the isotopic effects of atmosphere-snow water vapor exchange. In addition, reanalysis data indicate that at Dome A the time interval between two precipitation events can reach ~ 80 days (estimated based on ERA5 reanalysis dataset), which means that snow can sit at the surface for a substantially long period before burial, and is subject to extensive atmosphere-snow water vapor exchange, which

consequently affects the isotopic composition of the buried snow. Pang et al. (2019) estimated the potential influence of summer (November to January) sublimation on the isotopic composition of surface snow at Dome A using a simple Rayleigh distillation model. They found that on average surface snow $\delta^{18}O$ was enriched by 2.0 ‰ compared to fresh snow $\delta^{18}O$.

However, this evaluation may underestimate the isotopic effects since it did not consider the potential effects of atmospheric dynamic conditions and clouds. A new model is thus needed to provide a more comprehensive evaluation on the isotopic effect of atmosphere-snow water vapor exchange at Dome A, especially for seasons other than summer months when observations are not available.

To provide a more comprehensive assessment on the effects of atmosphere-snow water vapor exchange for snow and

atmospheric water isotope variations at Dome A, we constructed an improved one-dimensional box model based on previous work (Ritter et al., 2016; Casado et al., 2018; Touzeau et al., 2018) to predict changes in snow and water vapor isotopic compositions at Dome A at the diurnal scale. The main characteristics compared to models in the literature include the use of the bulk aerodynamic method to parameterize atmosphere-snow water vapor exchange. This model was first validated using observations at Dome C and then applied under Dome A conditions.

**2 Method**

**2.1 Model construction and description**

Similar to the model developed by Casado et al., (2018), the model presented in this study contains three water reservoirs, i.e., the free atmospheric water vapor layer, the atmospheric boundary layer and the topmost snow layer (Fig. 1). Their masses and isotopic compositions are considered to be associated only with atmosphere-snow water vapor exchange occurring at the

atmosphere-snow interface and the exchange of air masses occurring between the free atmosphere and boundary layer. These two processes can cause changes in the masses and isotopic compositions of water vapor in the boundary layer, whereas the masses and isotopic compositions of snow are influenced only by atmosphere-snow water vapor exchange.

The atmosphere-snow water vapor exchange consists of two processes: sublimation and deposition (Fig. 1). During sublimation, water vapor is released from snow, transported into the atmospheric layer via turbulent mixing and molecular

diffusion, and immediately mixed with the water vapor already in the boundary layer. During deposition, water vapor is influenced by aerodynamic resistance from turbulence and molecular diffusion, and the deposit is mixed with the surface snow layer. While water vapor transportation at the atmosphere-snow interface relies on two different diffusion pathways, turbulence plays a more crucial role in mass and energy exchanges (Brun et al., 2011; Vignon et al., 2017).

In the box model, atmosphere-snow water vapor exchange flux is calculated by turbulent quantities at each time step of 1 hour,

as detailed in Section 2.1.1. Based on atmosphere-snow water vapor exchange flux parameterization, the model further calculates temporal variations in snow and water vapor isotopic compositions according to isotopic mass balance (detailed in Section 2.1.2).

Model inputs mainly include meteorological conditions, e.g., air temperature ($T_a$), surface temperature ($T_s$), humidity (relative humidity ($RH_w$) or specific humidity ($q_a$)), and wind speed ($u_a$). Additional model inputs include the mixing-layer height ($H_0$), snow layer thickness ($h_0$), and the initial isotopic values, i.e., the snow isotopic composition ($\delta_{s0}$), water vapor isotopic composition in the boundary layer ($\delta_{v0}$), and water vapor isotopic composition in the free atmosphere ($\delta_{f0}$).

### 2.1.1 Atmosphere-snow water vapor exchange flux parametrization

We used the bulk aerodynamic method and Monin-Obukhov similarity theory (Monin & Obukhov, 1954) to estimate turbulent fluxes. This approach calculates the net effects of sublimation and deposition at each time step using meteorological data, avoiding to parameterize the individual fluxes of sublimation and deposition.

The bulk aerodynamic method estimates the atmosphere-snow water vapor exchange flux (Ex) through calculation of latent heat (LE) between the surface and one reference height (z) in the boundary layer (Berkowicz & Prahm, 1982). The expression is as follows:

$$Ex = LE/L_s = -\rho_a u^* q^* \tag{1}$$

where $\rho_a$ is the dry air density varying with the observed air temperature ($T_a$) and pressure ($P_a$), $L_s$ is the sublimation heat constant, and $u^*$ and $q^*$ are the friction velocity and specific humidity turbulence scale, respectively. The $u^*$ and $q^*$ are defined as:

$$u^* = \frac{ku_a}{log\left(\frac{z}{z_0}\right) - \Psi_M\left(\frac{z}{L}\right)} \tag{2}$$

$$q^* = \frac{k(q_a - q_0)}{log\left(\frac{z}{z_0}\right) - \Psi_M\left(\frac{z}{L}\right)} \tag{3}$$

where $k$ denotes the von-Karman constant, $u_a$ is the wind speed at the reference height in the boundary layer ($z = 4m$), $q_0$ is the saturated specific humidity at the snow surface derived from the Clausius-Clapeyron equation, $q_a$ is the specific humidity that can be estimated from the observed relative humidity over the ice surface ($RH_i$) once the saturated specific humidity at the reference height ($q_s$) is known from the August–Roche–Magnus Formula at a given temperature ($T_a$), $z_0$ represents the surface roughness length for humidity exchange, and $\Psi_M$ is diabatic correction term with respect to the ratio of the reference layer height (z) and Monin-Obukhov length (L), where $L$ is defined as:

$$L = \frac{\overline{\theta}}{g} \frac{u_*^2}{k\theta_*} \tag{4}$$

where $\overline{\theta}$ is the mean potential temperature between the snow surface ($\theta_0$) and the reference height in the boundary layer ($\theta_a$), $g$ is the gravitational acceleration, and $u^*$ and $\theta^*$ are the friction velocity and temperature turbulence scale, respectively. The $\theta^*$ is analogous to $u^*$ and $q^*$, using $\theta_a$, $z_0$, and $z/L$:

$$\theta^* = \frac{k(\theta_a - \theta_0)}{log\left(\frac{z}{z_0}\right) - \Psi_M\left(\frac{z}{L}\right)} \tag{5}$$

In Eq:(2), Eq:(3), and Eq:(5), $z_0$ can be estimated using least square fitting with the observed wind speed at three different heights under neutral atmospheric stratification. The $\Psi_M$ is calculated for stable, unstable and neutral boundary layers using

the functions taken from Louis (1979). The determination of atmospheric stability depends on the Richardson number (Ri), which is defined as follows:

$$Ri = \frac{g}{\theta_z} \frac{z \Delta \theta}{u_z^2} \tag{6}$$

Based on Eqs: (1)-(6), the atmosphere-snow water vapor exchange flux, $Ex$, can be calculated in the model with appropriate inputs. A positive value of $Ex$ represents net sublimation (i.e., sublimation > deposition), while a negative value of $Ex$ corresponds to net deposition (i.e., sublimation < deposition).

### 2.1.2 Isotopic Mass Balance

Assuming that the snow reservoir is influenced only by atmosphere-snow water vapor exchange (Fig. 1), temporal variations in snow mass per unit surface area (S) can be expressed as:

$$M_s^t = M_s^{t-1} - Ex \tag{7}$$

where $Ex$ is the exchange flux as calculated in the previous section, $M_s$ is the mass of the defined surface snow, and the superscript $t$ denotes time. From Eq: (7), $M_s$ at time $t$ can be calculated from the initial snow masses (i.e., masses at t=0) and the accumulated $E_x$ at time $t$. In the model, $M_s$ at t=0 relies on the initial snow height ($h_0$) and snow density ($\rho_s$). The water vapor mass in the boundary layer ($M_v$) at time $t$ and the unit area (S) can be computed from the initial boundary height ($H_0$), dry air density ($\rho_a$), and specific humidity ($q_a$) at the reference height in the boundary layer:

$$M_v^t = \rho_a H_0 q_a^t S \tag{8}$$

where $q_a$ at time $t$ can be determined by direct measurements or the observed relative humidity ($RH_i$). In Eq: (8), we neglect the temporal changes in the height of the boundary layer, given that the boundary heights in polar inland regions are relatively stable (Bonner et al., 2009; Ma B. et al., 2020). According to Eq: (8), the mass changes in the atmospheric boundary water vapor layer at each time interval are $\rho_v H_0 (q_a^t - q_a^{t-1})$. This quantity is influenced by atmosphere-snow water vapor exchange ($E_x$) and the water vapor exchange flux from the free atmosphere to the boundary layer ($M_f$). Thus, $M_f$ at any time can be quantified as follows:

$$M_f^t = M_v^t - M_v^{t-1} - Ex = \rho_v H_0 (q_a^t - q_a^{t-1}) - Ex \tag{9}$$

Note that the exchange between the boundary layer and the free atmosphere can occur under the unstable conditions or weak stable conditions (Zilitinkevich & Esau, 2007). In the model, we consider that $M_f$ can contribute to the atmospheric boundary water vapor reservoir when the Richardson number is less than 0.1 (i.e., including weak stable conditions in addition to unstable conditions).

Based on the calculation of mass changes in the three reservoirs (Eqs: (7-9)), the isotopic mass equations are:

$$M_s^t R_s^t = M_s^{t-1} R_s^{t-1} - R_{Ex}^t \times Ex \tag{10.a}$$

$$M_v^t R_v^t = M_v^{t-1} R_v^{t-1} + R_{Ex}^t \times Ex + R_f^t \times M_f^t \tag{10.b}$$

where $R_s$, $R_v$, $R_f$, and $R_{Ex}$ represent the ratios of heavy isotopes ($^{18}O$ and D) and light isotopes ($^{16}O$ and H) in the snow layer, atmospheric boundary layer, free atmospheric layer, and exchange flux, respectively.

The calculation of $R_{Ex}$ differs between the sublimation-dominated (i.e., net sublimation) period and deposition-dominated (i.e., net deposition) period. For the sublimation-dominated phase ($E_x>0$), kinetic fractionation is assumed to occur when the sub-saturation condition is considered. The isotopic composition of the sublimated vapor is calculated from Merlivat & Jouzel (1979), combining $R_s$, $R_v$, the diffusion coefficient (k'), the equilibrium coefficient ($\alpha_e$), and the relative humidity of the air with respect to the surface temperature (h) as follows:

$$R_{Ex}^t = \frac{1-k'}{1-h}\left(\frac{R_s^t}{\alpha_e} - h \times R_v^t\right) \tag{11}$$

The isotopic composition of the condensed vapor ($E_x<0$) is in equilibrium with that of the water vapor above -20°C. However, kinetic fractionation will also occur due to vapor supersaturation over ice on the East Antarctic Plateau. This effect can reduce the effective fractionation of water isotopes. Therefore, the equilibrium coefficient ($\alpha_e$) is replaced by the effective fractionation coefficient ($\alpha_f$) when calculating the $R_{Ex}$ of condensed vapor. The $\alpha_f$ is defined by the product of the kinetic fractionation

coefficient ($\alpha_k$) and $\alpha_e$. The $R_{Ex}$ of condensed vapor is thus expressed as:

$$R_{Ex}^t = \alpha_f(R_v^t + 1) - 1 \tag{12}$$

The $\alpha_e$ with respect to ice is given by Ellehoj et al. (2013) as a function of temperature (Eq: (13)).

$$\alpha_e^{{}^{18}O} = exp\left(0.0831 - \frac{49.192}{T} + \frac{8312.5}{T^2}\right) \tag{13.a}$$

$$\alpha_e^{D} = exp\left(0.2133 - \frac{203.10}{T} + \frac{48888}{T^2}\right) \tag{13.b}$$

The $\alpha_f$ is deduced from $\alpha_e$ as follows:

$$\alpha_f = \alpha_e \frac{RH_i}{1+\alpha_e(RH_i-1)\left(\frac{D_i}{D_i'}\right)} \tag{14}$$

where $D_i$ is the diffusivity of the water molecule and $D_i'$ is the same as $D_i$ but for heavy isotopes. The ratios of $D_i / D_i'$ are given by Jouzel & Merlivat (1984), with values of 1.0285 for $^{18}O$ and 1.0251 for D.

The key variables in the model are summarized and listed in Table S1.

**2.2 Model simulations**

We first used the above-mentioned model to simulate atmosphere-snow water vapor exchange and the associated isotope effects at Dome C (75.10°S, 123.33°E; 3233 m above sea level) where diurnal variations in water vapor isotopic compositions as well as surface snow water isotopes are available from observations (Casado et al., 2016; Touzeau et al., 2016). We then applied the model to Dome A conditions to investigate the isotopic effects due to atmosphere-snow water vapor exchange at

diurnal scales. The initial model values, including mixing-layer height ($H_0$), snow layer height ($h_0$), snow isotopic composition ($\delta s_0$), water vapor isotopic composition in the boundary layer ($\delta v_0$), water vapor isotopic composition in the free atmosphere layer ($\delta f_0$), and snow density ($\rho_s$) are listed in Table 1. These values were justified according to the conditions discussed in the following sections.

### 2.2.1 Diurnal simulations under Dome C conditions

At Dome C, previous observations revealed a clear diurnal cycle of water vapor isotopic composition from 5 to 16 January 2015 (Casado et al., 2016). This diurnal cycle was attributed to the effects of atmosphere-snow water vapor exchange under clear-sky conditions (Casado et al., 2018). To compare the modeled results with the observations, we performed a continuous simulation using observed meteorological data over the same period (11 days). Meteorological parameters (e.g., temperature, humidity, and wind speed) during the observed period were downloaded from the CALVA program (Genthon et al., 2010).

The surface snow temperature ($T_s$) data are available in a previous publication (Casado et al., 2016). The boundary height, $H_0$, was determined by Doppler Sodar measurements from an on-site iron tower at Dome C (Vignon et al., 2017). The surface snow layer height, $h_0$, was set to be the thickness of surface snow collected (i.e., 1.5 cm) for isotopic composition analysis at this site (Casado et al., 2018). The initial vapor isotopic compositions in the boundary layer, $\delta v_0$, were set as the observations of water vapor $\delta^{18}O$, $\delta D$, and d-excess at the beginning of the modeling period during the 2014/2015 field season (Casado et

al., 2016), while snow isotopes, $\delta s_0$, were set as the mean isotopic values of summer surface snow samples (Casado et al., 2018). The water vapor isotopic composition in the free atmosphere layer ($\delta f_0$) was not reported at this site. Here we expect that $\delta f_0$ is greater than $\delta v_0$. Although there are currently no vertical observations of water vapor isotopic composition in Antarctica, vertical isotopic profiles ($\delta^{18}O$) observed at the summit of Greenland have indicated that the isotopic composition of water vapor in the free atmosphere is slightly higher than that within the boundary layer (Berkelhammer et al., 2016). In

order to explain the water vapor and snow isotope observations at Dome C, Casado et al. (2018) assumed that the contribution from the free atmosphere can increase the ratio of $H_2^{18}O$ molecules in the boundary layer (Casado et al., 2018) and set $\delta f_0$ as the highest observed value of water vapor isotopic composition at Dome C. Note that $\delta f_0$ was a constant value for the simplicity of model calculations. The density of the topmost 5 cm of surface snow ($\rho_s$) was reported by Champollion et al. (2019).

### 2.2.2 Simulations under Dome A summer conditions

Previous studies have shown that a diurnal cycle clearly occurs in surface snow and water vapor isotopic compositions during clear-sky days, whereas this feature is not significant on highly cloudy periods (Casado et al., 2016; Ritter et al., 2016; Hughes et al., 2021). Clouds play an important role in modulating atmospheric thermal and dynamic conditions (Haynes et al., 2013), and cloudy conditions may also mean more moisture present in the atmosphere. Under cloudy conditions, extra moisture and downward radiation from clouds likely disturb local temperature and/or humidity variabilities, resulting in smaller differences

between day and night atmosphere-snow water vapor exchange and thus the isotopic effects are less pronounced. Therefore, in the model simulations for Dome A summer conditions, we not only simulated continuous changes in surface snow and water vapor isotopic composition over a multiday timescale, but also incorporated two representative scenarios (i.e., cloudy vs. clear-sky conditions) to ensure a rigorous assessment of the isotopic variations associated with atmosphere-snow water vapor exchange processes.

The simulations with continuous meteorological input were conducted without considering the influence of clouds. The selected period for summer simulations was from 5 to 16 January for each year from 2006 to 2011 (with the exception of 2005 for which data were not available). The model was thus run for 11 days each year, consistent with the Dome C simulations. By averaging the six simulated results obtained from the simulations, we were able to estimate the continuous changes in water vapor and snow isotopic composition. This approach allowed for a more robust analysis of the simulated data and enabled a

direct comparison of the results across different cases (results shown in Section 3.2.4 and Figure 7).

The hourly averages of total cloud cover (Tcc) were used to select days with clear-sky and highly cloudy conditions. These data were retrieved from the ERA-5 reanalysis dataset, with a spatial resolution of $1.25° \times 1.25°$. Based on previous studies, the classification criteria are as follows: $Tcc \leq 0.3$ for clear-sky conditions, and $Tcc \geq 0.8$ for highly cloudy conditions (Qian et al., 2012). Following this criterion, we selected 20 clear-sky days during the summer period (December to February) of 2005-

2011. Then, the hourly meteorological data from those selected days were stacked to create a representative cycle for model initialization. For highly cloudy conditions, a stack of 102 diurnal cycles of meteorological variables was also produced for modeling at the diurnal scale.

Meteorological data were obtained from an automatic weather station (AWS) installed near the summit of Dome A. The hourly surface air pressure, air temperature at heights of 1 m, 2 m and 4 m, relative humidity at 4 m, wind speed at heights of 1 m, 2

m and 4 m, and wind direction are available for the period of 2005-2011 (Ma et al., 2010; Ding et al., 2022). The surface snow temperature ($T_s$) was not available observations at Dome A. Thus, we performed $T_s$ calculations based on the method from Brun et al. (2011). The equation for $T_s$ calculations is shown as follows:

$$Ts = \left(\frac{LW_{up} + (\epsilon - 1)LW_{dn}}{\epsilon\sigma}\right)^{0.25} \tag{15}$$

where $\sigma$ is the Stefan–Boltzmann constant, $\epsilon$ is the snow emissivity (0.93), and $LW_{dn}$ and $LW_{up}$ are the downward and upward

longwave radiative fluxes respectively. The hourly longwave radiative flux data were retrieved from ERA5 reanalysis dataset.

The stacked hourly mean values of the meteorological conditions at Dome A are shown in Fig. 2a. During clear-sky conditions, the air temperature at the 4m level ($T_a$) shows a diurnal cycle with an amplitude of 10°C and an average of -31°C. The diurnal $T_s$ follows a similar pattern to that of $T_a$, varying between -39°C and -28°C. The ranges of diurnal cycles for specific humidity ($q_a$) and relative humidity ($RH_i$) are $1.8$-$3.7 \times 10^{-4}$ kg·kg$^{-1}$ and 66-130%, respectively. $q_a$ is also parallel to $T_a$, whereas $RH_i$

shows an opposite trend. In contrast to temperature and humidity, the daily air pressure near the surface is stable (~584 hPa). The wind speed ($u_a$) and latent heat flux reached the daily maxima of 3.0 m/s and 3.3 W·m$^{-2}$ respectively at 10:00 UTC, coinciding with the peaks in $T_a$, $T_s$ and $q_a$ on the diurnal scale. Under highly cloudy conditions, the latent heat exhibits less variability, yet $q_a$ and $u_a$ display greater diurnal variations (Fig. 2b).

The initial model values of $H_0$, $h_0$, $\delta s_0$, $\delta v_0$, and $\rho_s$ for the Dome A simulations are listed in Table 1. $H_0$ was estimated as the

median thickness of the boundary layer (15 m) based on sonic radar and visual observations of the angular size of stellar images during summer (Bonner et al., 2010; Ma B. et al., 2020). The surface snow thickness, $h_0$, was set to 1.5 cm according to summer snow accumulation at Dome A (calculated from the annual mean snow accumulation of 18-23 mm·yr$^{-1}$). $\delta s_0$ values were

obtained from the average precipitation isotopic composition measurements during 2009/2010 field season at Dome A (Pang et al., 2019). The $\delta v_0$ can be calculated from $\delta s_0$ assuming atmosphere-snow equilibrium and using the equilibrium fractionation coefficient at the surface temperature of the beginning of the diurnal cycle. $\delta f_0$ was set equal to the value at Dome C, since there are no measurements available at Dome A. The $\rho_s$ in Table 1 was from the measurements taken during 2014/2015 field season (Ma T. et al., 2020).

### 2.2.3 Simulations under Dome A winter conditions

Given the different meteorological conditions in winter compared to summer, the degree of atmosphere-snow water vapor exchange and the associated isotope effects could be different. Therefore, we also conducted multiday and diurnal simulations for winter at Dome A, similar to the summer simulations. This may shed light on assessments of the effects of atmosphere-snow water vapor exchange on seasonal and annual scales.

Winter simulations that incorporated continuous meteorological data were executed for a duration of 11 days, spanning from 5 to 16 July for each year between 2006 and 2011. This enabled the acquisition of 6 simulated results, which were subsequently averaged to provide a comprehensive understanding of the continuous changes in water vapor and snow isotopic composition over a multi-day timescale.

The stacked hourly mean values of winter meteorological conditions at Dome A were extracted in the same way as we did for the summer conditions. As shown in Fig. 2c, the average temperature, specific humidity, and atmospheric pressure are lower than those in summer, but the relative humidity increases during winter. These changes result in the negative values of latent heat flux during winter. In addition, the winter meteorological parameters and latent heat flux do not show any apparent diurnal variations.

The initial model values for the winter simulations are also listed in Table 1. The initial value of the snow isotopic composition ($\delta^{18}O_{s0}$) is the average of the precipitation isotopic composition at the starting month for the winter season. Due to the lack of observations, $\delta^{18}O_{s0}$ was estimated from the monthly mean temperature and the δ-T slopes in non-summer seasons (0.64±0.02) according to the compiled data in Pang et al. (2019). We also further evaluated these estimations of $\delta^{18}O_{s0}$ by comparison with snowfall $\delta^{18}O$ modeled using the ECHAM5-wiso model (Werner et al., 2011). The results of the two methods agree with each other (Supplementary Texts S3), suggesting that $\delta^{18}O_{s0}$ estimation using the regression line is reliable. The initial value for the water vapor isotopic composition ($\delta^{18}O_{v0}$) was also estimated assuming isotope equilibrium with $\delta^{18}O_{s0}$. $\delta f_0$ was set to be the calculated $\delta^{18}O_{v0}$ using $\delta^{18}O_{s0}$ and the highest temperature observed in winter during the studied period. $h_0$ is kept the same as that in summer to simplify the calculations. The median $H_0$ at Dome A varies little throughout most of the year according to Bonner et al. (2010) and Ma B. et al. (2020), so in the model we used the same $H_0$ in winter as that in summer. The $\rho_s$ is the annual mean snow density based on measurements (Ma T. et al., 2020) and we did not consider seasonal variations to simplify the calculations.

**2.2.4 Sensitivity simulations**

Changes in initial parameters could influence the isotopic effects of atmosphere-snow water vapor exchange. For example, previous field experiments have indicated that isotopic enrichment caused by atmosphere-snow water vapor exchange tends to decrease as snow thickness increases (Hughes et al., 2021). Ritter et al. (2016) noted that diurnal variations in water vapor isotopic composition decrease as the mixing layer height (i.e., $H_0$) increases. These previous findings motivate us to investigate the sensitivity of the modeled results to these boundary conditions and/or initial values.

The sensitivity tests included three groups of comparative experiments for the Dome A site and were run for a 24-h period under summer clear-sky conditions. The first group focuses on the sensitivity of surface and water vapor $\delta^{18}O$ to varying $h_0$ and $H_0$. In the experiment, we vary $h_0$ between 0.1 and 3.0 cm (Ritter et al., 2016; Hughes et al., 2021) and $H_0$ from 1 to 100 m (Bonner et al., 2010; Ritter et al., 2016). The second group is designed to investigate how the uncertainties in $\delta^{18}O_{s0}$ and $\delta^{18}O_{v0}$, influence the isotopic effects of atmosphere-snow water vapor exchange, especially when $\delta^{18}O_{s0}$ and $\delta^{18}O_{v0}$ are not in

equilibrium. We varied $\delta^{18}O_{s0}$ and $\delta^{18}O_{v0}$ from -53~-43 ‰ (the range of summer precipitation $\delta^{18}O$ at Dome A, Pang et al. (2019)) and -85~-55 ‰, respectively. The range of $\delta^{18}O_{v0}$ is estimated from $\delta^{18}O_{s0}$ and the equilibrium fractionation coefficient under summer conditions, and $\delta^{18}O_{s0}$ and $\delta^{18}O_{v0}$ in thermodynamic imbalance are included. In the third group, $\delta^{18}O_{f0}$ and snow density were varied to test their influence on the diurnal changes in surface snow and water vapor $\delta^{18}O$, respectively. The selection of -68~-58 ‰ for the $\delta^{18}O_{f0}$ range refers to the summer observations of water vapor isotopic composition at Dome C

(Casado et al., 2016). According to field observations at Dome A and other interior domes (Laepple et al., 2018), the range of snow density was set to 300-400 kg/m$^3$ for sensitivity simulations. Note that the isotope effects are greater in summer than in winter, we only used summer conditions and values to illustrate the sensitivity of the modeled results to these parameters.

**3 Results**

**3.1 Modeled diurnal and multi-day variations at Dome C**

On the diurnal scale at Dome C, the modeled water vapor $\delta^{18}O$ increase from -68 ‰ at 00:00 UTC to -66 ‰ at 09:00 UTC and then decreases to -75 ‰ at 16:00 UTC (Fig. 3a). The diurnal patterns in water vapor $\delta D$ are similar to that in water vapor $\delta^{18}O$ and their max-min difference is ~54‰ (Fig. 3b). The water vapor d-excess, defined by d-excess (‰) $\equiv \delta D$-8*$\delta^{18}O$ (Dansgaard, 1964), varies between 52 ‰ and 72 ‰ during the 24-h period (Fig. 3c). Its diurnal pattern is opposite to that of $\delta^{18}O$ and $\delta D$. The modeled snow $\delta^{18}O$ and $\delta D$ also exhibit a diurnal pattern where higher values occur during the warming phase and lower

values occur during the cooling phase (Fig. 3d). The diurnal range of simulated snow $\delta^{18}O$ is ~1.5‰ on average, but its value is close to that of the observations (2.0‰) during a typical frost event from 6 to 7 January, 2015. In addition, the diurnal variations in snow d-excess are opposite to those in snow $\delta^{18}O$ and $\delta D$, similar to the relationship between vapor $\delta^{18}O$ and d-excess. Overall, the modeled diurnal variations in vapor $\delta^{18}O$ and $\delta D$ capture the observations well, while their magnitudes are slightly different from those in observations.

The continuous simulations at the multiday scale are shown in Fig. 3e. The simulated water vapor $\delta^{18}O$ exhibits periodic changes on the diurnal scale, but its daily mean value remains unchanged over the course of the simulation. This trend is consistent with the observations reported by Casado et al. (2016), as evidenced by a high correlation coefficient (R > 0.6). The snow $\delta^{18}O$ values display a noticeable enrichment trend compared to its initial state, which is different from that of the water vapor $\delta^{18}O$.

## 3.2 Modeled results at Dome A

### 3.2.1 Diurnal variations under summer clear-sky conditions

At Dome A, the Richardson number ($R_i$) varies between -0.01 and 0.02 during the 24-h period (Fig. 4a). The friction velocity of water molecules (u*) ranges from 0.11 to 0.19 m/s, with a mean value of 0.14 m/s (Fig. 4b). The atmosphere-snow water vapor exchange flux ($E_x$) calculated from $R_i$ and $u*$ varies in parallel with temperature (Fig. 4c). In general, negative $R_i$ values
represent relatively unstable atmospheric conditions, which corresponds to the phase of sublimation (i.e., net vapor flux from snow to the atmosphere, Fig. 4c). In contrast, $R_i$ appears to be positive during most of the cooling phase (i.e., the net vapor flux from the atmosphere to snow, Fig. 4c), suggesting stable atmospheric conditions.

Figs. 4d-4f display the modeled surface snow and water vapor isotopic compositions and the uncertainties. All the isotopes display apparent diurnal cycles. In particular, water vapor $\delta^{18}O$ and $\delta D$ indicate enrichments in the sublimation period, followed
by depletions during the rest of the day when condensation (vapor deposition) dominates (Figs. 4d and 4e). The snow $\delta^{18}O$ and $\delta D$ values exhibit similar but somewhat opposite patterns within 24 hours (Figs. 4d and 4e). The diurnal pattern of d-excess is opposite to that of $\delta^{18}O$ and $\delta D$ in snow and vapor (Fig. 4f). Overall, the diurnal patterns of snow and water vapor isotopes at Dome A are similar to those at Dome C during summer cloudless conditions.

The magnitudes of the diurnal range in water vapor isotopic composition are 4.8 ‰ for $\delta^{18}O$, 29 ‰ for $\delta D$ and 9.3 ‰ for d-
excess. In comparison, the modeled diurnal isotope variations in surface snow are much smaller with magnitudes of 0.80 ‰ for $\delta^{18}O$, 1.6 ‰ for $\delta D$ and 4.9 ‰ for d-excess. In addition, after 24-hours of model operation, the water vapor $\delta^{18}O$, $\delta D$, and d-excess increase by 2.4 ‰, 16 ‰, and 3.1 ‰, respectively (Figs. 4d-4f). Moreover, after 24 hours, the snow isotopic compositions display enrichments of 0.29 ‰ for $\delta^{18}O$ and 1.1 ‰ for $\delta D$, and a depletion of 1.3 ‰ for d-excess.

### 3.2.2 Diurnal variations under highly cloudy summer conditions

Under highly cloudy conditions, the Richardson number ($R_i$) is almost neutral or unstable at the diurnal scale (Fig. 5a). The the friction velocity (u*) exhibits a diurnal cycle varying between 0.11 m/s and 0.13 m/s (Fig. 5b), which is much smaller than that under clear-sky conditions. We also found a diurnal cycle in the atmosphere-snow water vapor exchange flux ($E_x$), as shown in Fig. 5c. Overall, the diurnal changes in $u*$, $R_i$ and $E_x$ are less pronounced compared with those under clear-sky conditions.

The diurnal cycle patterns in water and surface snow isotopic compositions are also apparent under cloudy conditions (Figs. 5d-5f), but the magnitudes are smaller than those under clear-sky conditions. In particular, the diurnal peak-to-valley differences in water vapor isotopic compositions are 3.0 ‰ for $\delta^{18}O$, 21 ‰ for $\delta D$ and 4.0 ‰ for d-excess. The diurnal variations in the surface snow isotopic composition have a magnitude of 0.28 ‰ for $\delta^{18}O$, 0.87 ‰ for $\delta D$, and 2.2 ‰ for d-excess. In addition, the same as in clear-sky conditions, after 24-hours, snow water isotopes were enriched in the model.

### 3.2.3 Diurnal variations under winter conditions

The winter simulation results are plotted in Fig. 6. Under winter conditions, the Richardson number (Ri) and the friction velocity (u*) remain stable over a full 24-hour period (Figs. 6a and 6b). The atmosphere-snow water vapor exchange flux ($E_x$) shows negative values throughout 24 hours (Fig. 6c), suggesting that sublimation does not occur under Dome A winter conditions. As a result, in contrast to the simulated results in summer, there are no significant diurnal variations in snow

isotopes in winter, but the changes in water vapor isotopic composition in winter are comparable to those in summer. This can be associated with the almost constant meteorological conditions and the relatively weak exchange between snow and atmospheric water vapor during a diurnal period, as displayed in Fig. 2c. In addition, because the isotopic composition of deposited vapor is much lower than that of surface snow, the winter snow layer experiences small but steady depletions in $\delta^{18}O$ and $\delta D$ (Figs. 6d and 6e). In contrast, snow d-excess becomes more enriched under the effects of the atmosphere-snow

water vapor exchange flux (Fig. 6f). The water vapor isotopic composition also displays a depletion because heavier isotopes tend to deposit faster.

### 3.2.4 Continuous changes at the multiday scale at Dome A

The continuous simulations presented in Fig. 7 reveal that the water vapor isotopic composition ($\delta^{18}O$) exhibits substantial interannual differences in absolute values, even during the same period (Fig. 7a and 7c). In addition, these simulations and

their averages display distinct diurnal periodicity. On the multi-day scale, the average water vapor $\delta^{18}O$ values do not show a significant trend with increasing simulation time. Their values fluctuate around -72 ‰ in the summer and -105 ‰ in the winter. The diurnal cycles shown in the Dome A continuous simulations are consistent with the simulated results at Dome C. The snow isotopic composition ($\delta^{18}O$) simulations in each year exhibit a striking similarity in their trend during the summer and winter seasons. Specifically, the snow $\delta^{18}O$ values at the end of the simulation are consistently higher than the initial values

during the summer (Fig. 7b). Conversely, a slightly negative trend can be observed in the winter simulations (Fig. 7d).

### 3.3 Sensitivity to model parameters

In the first group of sensitivity tests (Fig. 8a), the water vapor and snow isotopic composition displayed distinct patterns in response to variations in snow depth ($h_0$) and the boundary layer height ($H_0$). The magnitude of the diurnal variations in water vapor $\delta^{18}O$ ($\delta^{18}O_v$) is highly influenced by the $H_0$ but not by $h_0$. This finding aligns with previous calculations at Kohnen

Station, which demonstrated a decrease in the magnitude of $\delta^{18}O_v$ with increasing mixing layer height (Ritter et al., 2016). On

the other hand, the magnitude of diurnal variations in snow $\delta^{18}O$ ($\delta^{18}O_s$) exhibits a greater sensitivity to $h_0$ (Fig. 8b). This finding is consistent with field experiments showing that isotopic enrichment induced by atmosphere-snow water vapor exchange tends to decrease with increasing snow thickness (Hughes et al., 2021). Similar to the magnitude of $\delta^{18}O_s$, the changes in $\delta^{18}O_s$ after a diurnal cycle are more sensitive to $h_0$ (Fig. 8c).

In the second group of tests, where $\delta^{18}O_{s0}$ and $\delta^{18}O_{v0}$ varies, the magnitude of $\delta^{18}O_v$ diurnal changes is more sensitive to $\delta^{18}O_{v0}$ than $\delta^{18}O_{s0}$ (Fig. 8d). As $\delta^{18}O_{s0}$ decreases, the magnitude of $\delta^{18}O_s$ diurnal changes increases, emphasizing the influence of $\delta^{18}O_{s0}$ on snow isotopic variations (< 0.05‰ in Fig. 8e). In addition, the value of $\delta^{18}O_s$ after a diurnal cycle shows a greater sensitivity to $\delta^{18}O_{s0}$, while such a change remains small (<0.01‰ in Fig. 8f).

In the third group, with varying $\delta^{18}O_{f0}$ and snow density ($\rho_s$), changes in $\delta^{18}O_{f0}$ significantly influence the magnitude of diurnal

variations in $\delta^{18}O_v$ (Fig. 8g). In contrast, these changes have a lesser effect on the magnitude of diurnal $\delta^{18}O_s$ variations and $\delta^{18}O_s$ changes after a diurnal cycle (Figs. 8h and 8i). The snow density has a considerable effect on $\delta^{18}O_s$, while it induces only a small change in the magnitude of diurnal $\delta^{18}O_v$ fluctuations.

## 4 Discussion

Despite differences in the magnitudes, under summer clear-sky and highly cloudy conditions the modeled isotopes in surface
snow and water vapor display clear diurnal patterns at Dome A. In both of these cases, the water vapor isotopes show a smaller magnitude of diurnal variations with respect to the snow isotopes. In general, in the period of mass exchange dominated by sublimation, snow $\delta^{18}O$ and $\delta D$ are enriched because lighter isotopes are preferentially sublimated to the atmosphere. Moreover, sublimates mixing with vapor water leads to increases in vapor $\delta^{18}O$ and $\delta D$ because they have higher $\delta^{18}O$ and $\delta D$ values than atmospheric vapor. During periods of mass exchange dominated by deposition, water vapor $\delta^{18}O$ and $\delta D$ are significantly
depleted (Ritter et al., 2016). Note that the effects on snow $\delta^{18}O$ and $\delta D$ are smaller than those on vapor $\delta^{18}O$ and $\delta D$. This is because the surface snow mass reservoir is much larger than the mass of deposition, so the associated isotope effects on surface snow are very small (Steen-Larsen et al., 2013; Casado et al., 2018).

Based on Figs. 2, 4c, 4d, 5c, and 5d, it is evident that the diurnal isotope cycles in surface snow and water vapor have a strong correlation with surface temperature and humidity. As described in Section 2.1, surface temperature can modify local
atmospheric dynamic conditions and specific humidity, leading to synchronous responses in atmosphere-snow water vapor exchange fluxes. Temperature can also affect isotope fractionation during phase exchange. Atmosphere-snow water vapor exchange is associated with equilibrium and kinetic isotope fractionation between snow and water vapor (Ritter et al., 2016; Hughes et al., 2021; Wahl et al., 2021). The degree of isotopic equilibrium fractionation is directly dependent on the local surface temperature (Ellehoj et al., 2013), while kinetic isotope fractionation is mainly driven by the vapor pressure gradient
between the snow surface and atmosphere (Jouzel & Merlivat, 1984; Surma et al., 2021; Passey & Levin, 2021). The specific humidity is also crucial because it represents the size of the water vapor reservoir with which snow can exchange (Casado et al., 2018). However, it is only important for atmospheric vapor $\delta^{18}O$ and $\delta D$ as surface snow is a much larger mass reservoir

that buffers the effects of atmospheric vapor change. Wind speed also plays a key role in driving isotopic variations at Dome A, because its increase can amplify the variations in latent heat, leading to more pronounced diurnal changes in water vapor and snow isotopic composition (Supplementary Texts S5, Bréant et al., 2019).

The diurnal variations of water vapor isotopic composition, resulting from the exchange between the atmosphere and snow surface, are subject to influences beyond mere meteorological conditions. Specifically, fluctuations in the boundary layer height ($H_0$) can result in either an attenuation or an amplification of the magnitude of variations in water vapor isotopic composition (Ritter et al., 2016), as evidenced by Fig. 8a. Furthermore, the interaction between the free atmosphere and the boundary layer can significantly impact the diurnal variations in the water vapor isotopic composition (Casado et al., 2018). Specifically, during periods of intense mixing, the variations in water vapor isotopic composition become more pronounced (Fig. 8g and Supplementary Texts S4). However, in the model employed for this study, the boundary layer height ($H_0$) and water vapor isotopic composition in the free atmosphere layer ($\delta f_0$) are maintained as constants to simplify the calculations, whereas they vary daily in reality. This simplification for model calculations may lead to a reduction in the interday variability of simulated water vapor isotopic compositions (Fig. 3e).

We also compared our modeled water vapor $\delta^{18}O$, $\delta D$, and d-excess data at Dome A with water vapor $\delta^{18}O$, $\delta D$, and d-excess data from other East Antarctic interior sites from observations, such as the Kohnen station, Dome C, and a location approximately 100 km away from Dome A (Ritter et al., 2016; Casado et al., 2016; Liu et al., 2022). Both our simulations and observations have similar diurnal patterns, with high values occurring during daytime warming and low values occurring during nighttime cooling. However, it is worth noting that the magnitudes differ between the diurnal simulations at Dome A and the observations at other sites. Our modeled $\delta D$ variations at Dome A (29±19‰) are lower than the observed diurnal variations in water vapor $\delta D$ at Kohnen station (36±6‰ from Ritter et al., (2016)) and at Dome C (38±2‰ from Casado et al., (2016)). This difference can be attributed to the atmospheric dynamic conditions at Dome A, which are characterized by a lower daily mean wind speed (2.8 m/s) than those to Dome C (3.3 m/s) and Kohnen station (4.5 m/s) during summer season (Casado et al., 2018). A lower wind speed corresponds to relatively weak air convection in the vertical direction. Due to the coupling between upper and lower atmospheric layers, vertical turbulent mixing may decrease with weakened air convection in the atmospheric near-surface layer (Casado et al., 2018). This change can attenuate molecular exchange between surface snow and water vapor, leading to a muted fluctuation in the modeled water vapor $\delta D$ in combination with less mass exchange. In addition, the simulated diurnal changes in water vapor isotopic composition are lower than those observed at sites near Dome A (>40‰ for $\delta^{18}O$ and 200‰ for d-excess). This large discrepancy may be due to calibration drifts caused by the low water vapor content during the measurements at the nearest Dome A site (Liu et al., 2022).

The magnitudes of the modeled diurnal changes in snow $\delta^{18}O$ and $\delta D$ are different between highly cloudy and clear-sky conditions, with apparently small magnitudes under cloudy conditions. It seems that when clouds are present, surface snow will receive longwave radiation from clouds and be less influenced by solar radiation. As a result, the diurnal radiation budget cycle is less variable than that on days without clouds, as otherwise, solar radiation with a strong diurnal cycle becomes the most significant variable. On days with clouds, the diurnal variations in air temperature and surface temperature are also

smaller and the differences between the air temperature and surface temperature during the day and night become less pronounced (Fig. 2). This could have a negative impact on the changes in atmospheric dynamics between day and night. The diurnal variations in the wind speed and friction velocity are thus not significant (Figs. 2, 4b, and 5b). The vertical turbulent mixing between surface snow and water vapor in a diurnal cycle is relatively stable, leading to less mass exchange as well as isotope effects between the two reservoirs.

The model results for summer clear-sky and highly cloudy conditions also indicate that after a 24-hour simulation, $\delta^{18}O$ and $\delta D$ in surface snow are enriched mainly due to isotope fractionation during sublimation. Notably, although water vapor with much lighter $\delta^{18}O$ and $\delta D$ values than snow are deposited in the deposition period, the masses are negligible compared to those in the snow mass reservoir so the effects on snow isotopes in the 24-hour simulation period are dominated by the effects of sublimation. The enrichments in snow isotopes caused by sublimation are consistent with previous studies (e.g., Ritter et al., 2016; Casado et al., 2018; Hughes et al., 2021). In addition, sublimation is associated with snow mass loss. Many studies also indicate significant surface snow mass loss during summer due to sublimation at inland Antarctic sites including Dome A (e.g., Frezzotti et al., 2004; Ding et al., 2016). As such, at Dome A, surface snow isotopes are presumably enriched during summer. Using a simple Rayleigh distillation model, Pang et al. (2019) predicted that over summer ~2 ‰ enrichments in surface snow $\delta^{18}O$ can be caused under mean Dome A summer conditions.

Based on the results of the sensitivity tests, diurnal variations in isotopic composition of snow due to water vapor exchange processes can also be influenced by several parameters, such as snow thickness, snowfall isotopic composition, snowfall density, and surface roughness (refer to Fig. 8 and Supplementary Texts S5). Among these factors, changes in snowpack thickness exhibit the most pronounced impact on the isotopic effects of water vapor exchange processes. Specifically, when the snow thickness exceeds 3 cm, the water vapor exchange effect struggles to induce interday variations in snow isotopes. On the other hand, the effects of snowfall isotopic composition, snowfall density, and surface roughness on the isotopic composition of surface snow may be limited during the Dome A summer season (Supplementary Texts S5), given the realistic range of potential variations in snowpack parameters.

Under the typical winter conditions at Dome A, temperature and humidity remain relatively constant throughout the day (i.e., during a 24-hour simulation period). The Richardson number (Ri) is positive throughout the day, indicating stable atmospheric conditions. As a result, the diurnal variations in the exchange of atmospheric water vapor and snow isotopes are less pronounced. Specifically, the model simulations suggest that under these conditions, only deposition occurs, leading to a depletion of snow isotopes ($\delta^{18}O$ and $\delta D$) after the 24-hour simulation period.

Because the diurnal variations in snow isotopic composition induced by atmosphere-snow water vapor exchange in summer and winter are different, seasonal snow isotopic changes can be affected. In particular, according to the modeled results, in summer surface snow $\delta^{18}O$ and $\delta D$ should become more enriched than fresh snow, while in winter surface snow isotopes should be less abundant than fresh snow. This effect appears to be distinct from what can be expected from other post-depositional processes. For example, Town et al. (2008) demonstrated that wind-driven ventilation after snowfall can result in isotope enrichment in winter snow layers and depletion in summer snow layers, decreasing the magnitude of seasonal

variations. Vapor diffusion in snow pores also contributes to the attenuation of $\delta^{18}O$ or $\delta D$ seasonal variations by smoothing (Johnsen et al., 2000; Casado et al., 2020). To evaluate the annual net effect of atmosphere-snow vapor exchange, the potential mass loss in summer and gain in winter must be estimated. From just the continuous simulations of this study, it appears that the annual net effects should lead to isotopic enrichment in the snow layer, since the magnitudes of isotopic changes in summer are much larger than those in winter. However, we note that the continuous simulation in this study was conducted without differentiating between clear-sky and cloudy conditions and was considerably affected by abrupt temperature fluctuations observed at Dome A. Therefore, further exploration of continuous simulations is required, which can be achieved through improvements in model refinement and the capabilities of observational techniques with more precise data available.

## 5 Conclusions

Atmosphere-snow water vapor exchange is important for snow isotope preservation as suggested by previous studies (Ritter et al., 2016; Hughes et al., 2021; Hu et al., 2022). In this study, we constructed a new box model based on the bulk aerodynamic method to predict changes in surface snow and water vapor isotopic compositions in response to diurnal fluctuations in local meteorological conditions. The model was validated by the agreement between the modeled and observed diurnal cycles of water vapor $\delta^{18}O$, $\delta D$, and d-excess at Dome C and then applied to investigate the degree of atmosphere-snow water vapor exchange and the associated isotope effects at Dome A on diurnal scales. The model results show that atmosphere-snow water vapor exchange at Dome A can also lead to similar diurnal isotope variations in atmospheric water vapor $\delta^{18}O$ and $\delta D$ under summer conditions, with corresponding diurnal variations in surface snow $\delta^{18}O$ and $\delta D$. For clear-sky conditions, the magnitudes of the diurnal cycles in snow and water vapor isotopes are greater than those in simulations under highly cloudy conditions. In addition, we performed diurnal simulations under Dome A winter conditions. The results indicate that the diurnal isotope variations over the 24-hour simulation period are less significant due to the stable atmospheric conditions with low and relatively stable air temperature and specific humidity. However, the model results suggest that snow isotope depletion can occur in winter. The modeled opposite isotope effects on snow after 24 hours in winter and summer at Dome A suggest that atmosphere-snow water vapor exchange could increase the seasonal snow isotope variations. The modeled changes in winter are smaller than those in summer, due to the highly stable boundary layer conditions in winter. This means that the effects in summer cannot be offset by those in winter, leading to overall enrichments in snow isotopes.

We also acknowledge the limitations inherent to our simulations with a one-dimensional model. The air mass exchange process between the free atmosphere layer and boundary layer may play an important role in atmosphere-snow water vapor exchange as observed during some frost events (Casado et al., 2018). Although the influence of the free atmosphere has been incorporated into our model, it is worth refining the underlying assumptions for the air mass exchange process and improving the accuracy of the model simulations. Further, observational validation of the model results for the winter season is unavailable due to the extreme harsh conditions at Dome A. Although it is currently difficult to conduct field work at the diurnal scale there, observations on longer timescales (e.g., weekly resolved sampling of surface snow and precipitation over a year along with a

snowpack to reconstruct the changes after deposition) could be possible. These results are important for validating the model's ability to predict on the associated isotope effects of atmosphere-snow water vapor exchange, especially considering that the model implies atmosphere-snow water vapor exchange may have few isotope effects at the annual scale but tend to increase snow water isotope seasonality. The latter is opposite to other post-depositional processes such as wind-driven ventilation (Town et al., 2008) and vapor diffusion in snow pores (Johnsen et al., 2000).

## Data availability statement

The simulated data and model code are available upon request to Tianming Ma (Email: mtm@ustc.edu.cn). Other data and software used in this study are also available online. We also acknowledge the use of Dome C data from the CALVA project and CENECLAM and GLACIOCLIM observatories (http://www-lgge.ujf-grenoble.fr/~christo/calva/). Meteorological observations at Dome A can be downloaded from the Australian Antarctic Data Centre (https://data.aad.gov.au/metadata/records/DomeA_AWS). The hourly averages of total cloud cover and longwave radiative fluxes are sourced from the ERA5 reanalysis dataset (https://cds.climate.copernicus.eu/cdsapp#!/dataset/reanalysis-era5-single-levels).

## Author contributions

LG and TM conceived this study. TM and PH performed the model simulations, analyzed the data and wrote the manuscript with LG. LG and JZ provided helped the model construction. MD, WZ and YL provided available data for the simulations. All the authors contributed to the data interpretation and writing.

## Competing interests

The authors declare that there are no conflicts of interest.

## Acknowledgments

The research leading to these results has received financial support from the National Natural Science Foundation of China (42206242 to T. M.; 42206245 to P. H.), the State Key Laboratory of Cryospheric Science for the Open Fund (SKLCS-OP-2020-06 to T. M.), the Nature Science Research Project of Anhui Province (2108085QD158 to T. M.), and the Fundamental Research Funds for the Central Universities. L.G. acknowledges financial support from the National Natural Science Foundation of China (Awards: 41822605, 41871051 and 41727901), the Fundamental Research Funds for Central Universities, the Strategic Priority Research Program of Chinese Academy of Sciences (XDB 41000000), and the National Key R&D Program of China (2019YFC1509100). This research was also supported in part by National Natural Science Foundation of

China (49973006, 40773074 and 40703019 to Y. L.) and the Ministry of Science and Technology of China (2006BAB18B01 to Y. L.). The authors are grateful for the   data collected by Chinese National Antarctic Research Expedition during the summer of 2005-2011.

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

**Table 1: Key initial values for diurnal simulations.**

| Site | | Dome C | Dome A | Dome A |
|---|---|---|---|---|
| Period | | Summer (5th-16th, January) | Summer (December-February) | Winter (June-August) |
| $H_0$ (m) | | 10 | 15 | 15 |
| $h_0$ (cm) | | 1.50 | 1.50 | 1.50 |
| Snow isotopic composition (‰) | $\delta^{18}O_{s0}$ | -51.16[a] | -48.18[b] | -61.92[b] |
| | $\delta D_{s0}$ | -394.00 | -372.90 | -474.72 |
| | d-ex$_{s0}$ | 15.28 | 12.54 | 20.64 |
| Water vapor isotopic Composition in the near-surface boundary layer (‰) | $\delta^{18}O_{v0}$ | -68.00 | -70.40[c]/-70.4[d] | -94.69[e] |
| | $\delta D_{v0}$ | -490.00 | -500.59/-500.64 | -625.54 |
| | d-ex$_{v0}$ | 52.00 | 62.64/62.67 | 131.98 |
| Water vapor isotopic composition in the free atmosphere (‰) | $\delta^{18}O_{f0}$ | -63.00 | -63.00 | -88.00 |
| | $\delta D_{f0}$ | -440.00 | -440.00 | -574.00 |
| | d-ex$_{f0}$ | 64.00 | 64.00 | 130.00 |
| $\rho_s$ (kg·m$^{-3}$) | | 329 | 380 | 380 |

[a, b] Observations for surface snow isotopes and calculations for fresh snow isotopes, respectively; [c, d] Values correspond to clear sky and highly cloudy conditions, respectively; [e] Some of the winter conditions were set the same as those in summer (see details in Section 2.2.3).

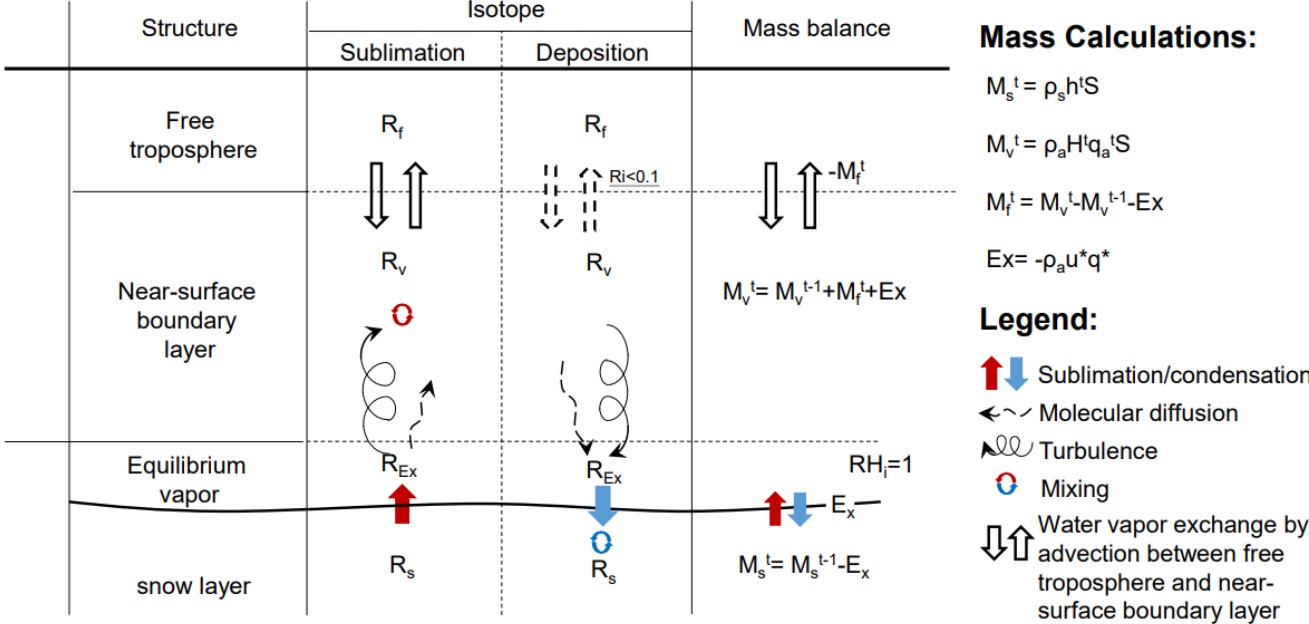


**Figure 1: Schematic diagram of the box model used in this study. Please note that the dotted arrow between the free atmosphere and the boundary layer indicates that exchange can only take place under the condition that the Richardson number (Ri) is less than 0.1.**


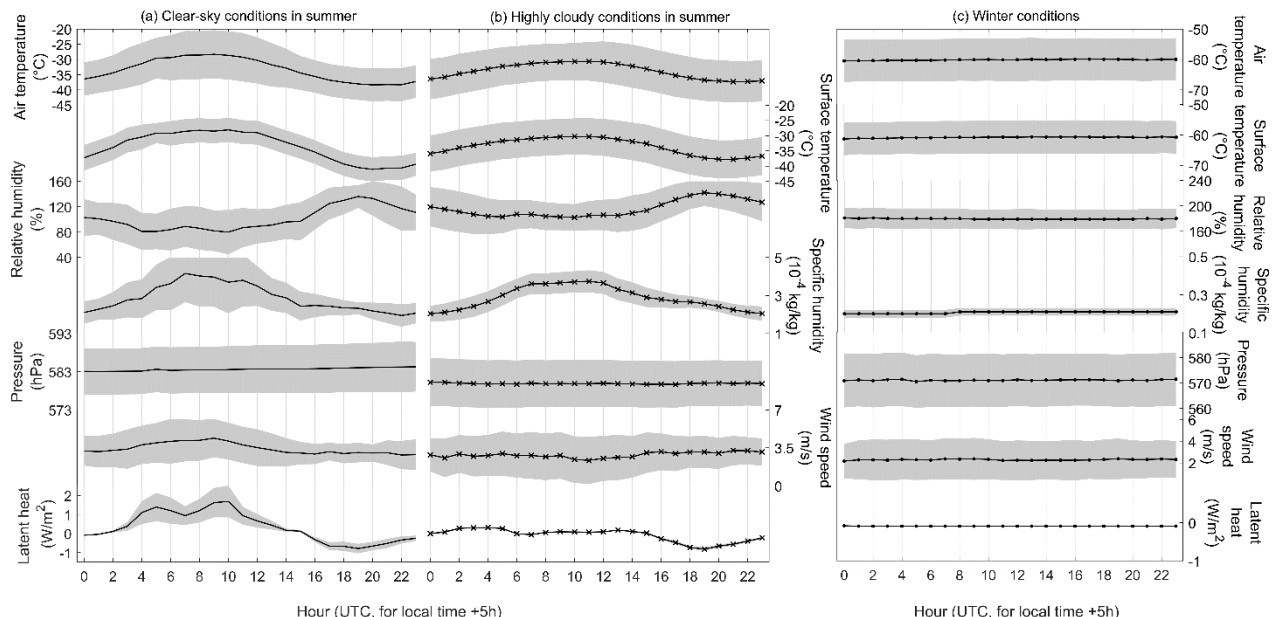

Figure 2: Stacks of diurnal cycles of meteorological parameters and the calculated latent heat under summer clear-sky conditions (a), summer highly cloudy conditions (b), and winter conditions (c) at Dome A. The hourly air temperature, relative humidity, air pressure and wind speed data were averaged from the AWS observations on the selected days. The diurnal variations in the other three parameters were calculated based on hourly observations. In each panel, the solid line with marks represents the average and the gray shadow represents the standard deviation ($\pm 1\sigma$).


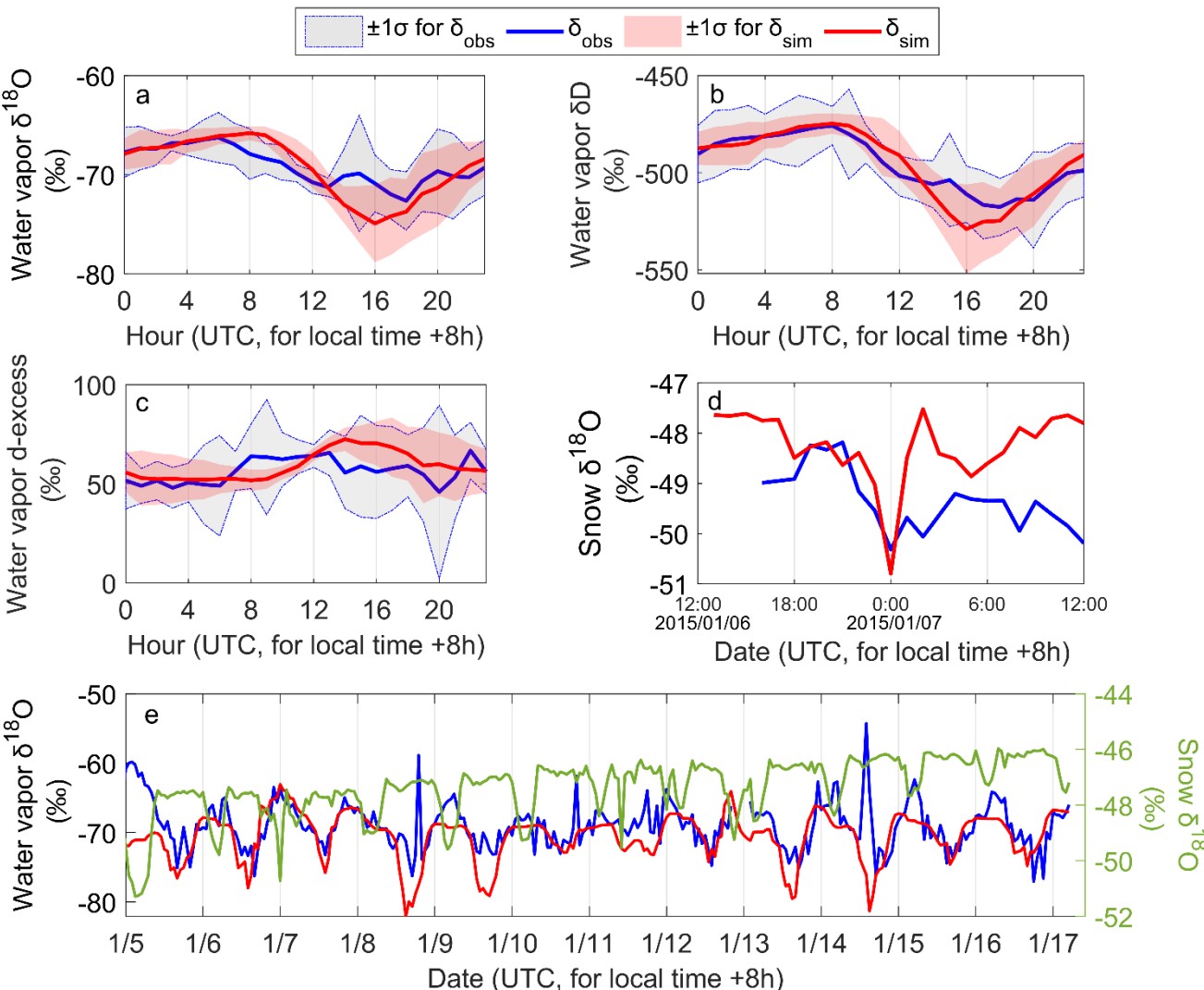

**Figure 3: Modeled variations in water vapor and snow isotopic compositions at Dome C along with the observations. (a) Diurnal variations in water vapor $\delta^{18}O$, (b) diurnal variations in water vapor $\delta D$, (c) diurnal variations in water vapor d-excess, (d) diurnal variations in snow isotopes during 6-7 January 2015, and (e) continuous variations in water vapor and snow $\delta^{18}O$ during 5-16 January 2015. In all panels, the blue solid line represents the observations ($\delta_{obs}$) with the light gray shaded area as the uncertainties**

 **($\pm 1\sigma$). The red solid line and the light red shaded area depict the simulation ($\delta_{sim}$) and corresponding uncertainties ($\pm 1\sigma$), respectively. In panel (e), the green solid line represents the modeled snow $\delta^{18}O$. Note that snow $\delta^{18}O$ observations at Dome C are available only from 6 to 7 January 2015 (Casado et al., 2018). The method for uncertainty estimation can be found in supplementary (Supplementary Texts S2).**

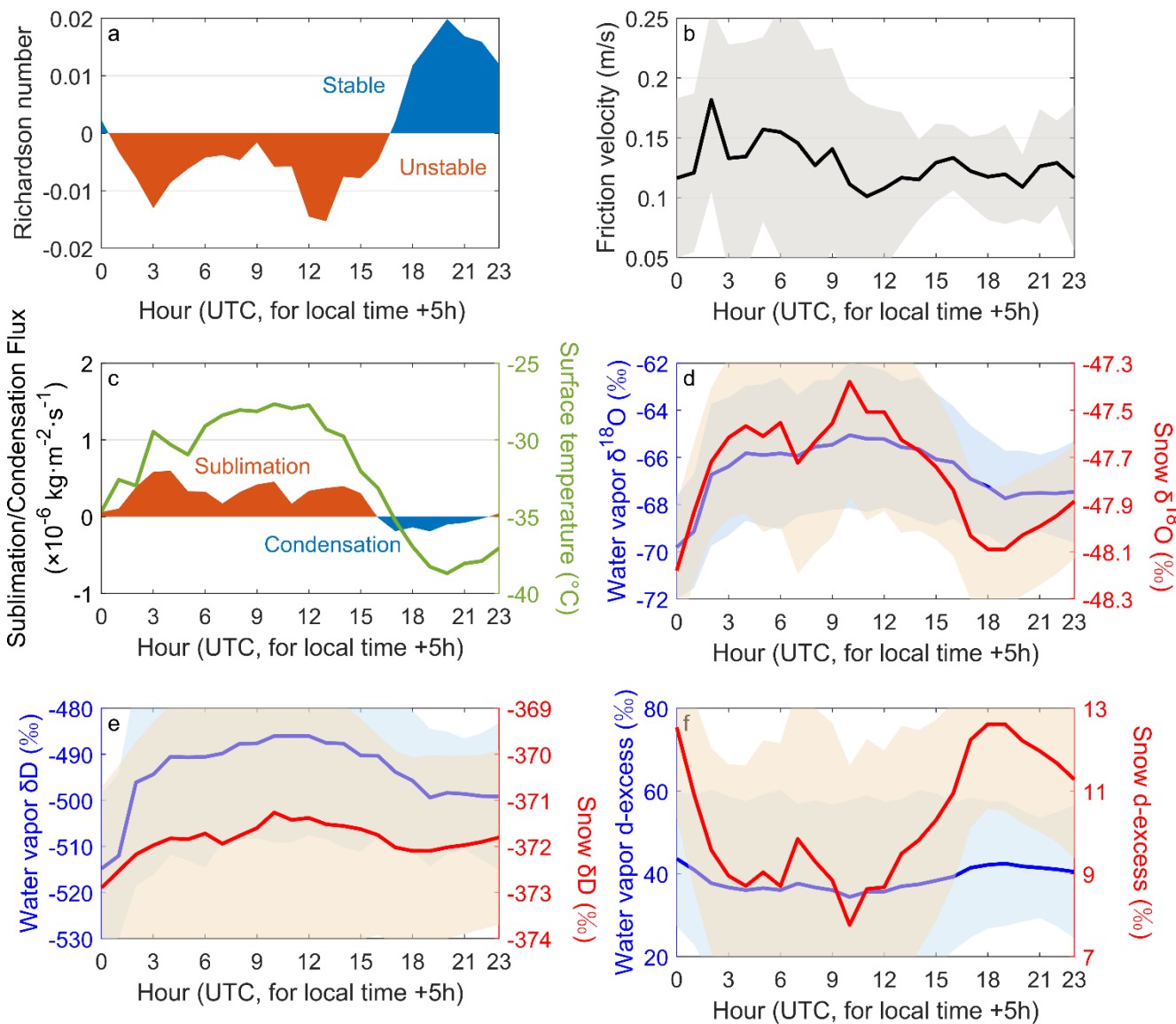


**Figure 4: The simulated hourly mean vapor exchange flux and variations in atmospheric water vapor and snow isotopes under summer clear-sky conditions at Dome A: (a) Richardson number, (b) friction velocity, (c) vapor exchange flux, (d) snow and water vapor δ¹⁸O, (e) snow and water vapor δD, (f) snow and water vapor d-excess. The uncertainties for each variable are displayed by shaded areas in each subpanel.**

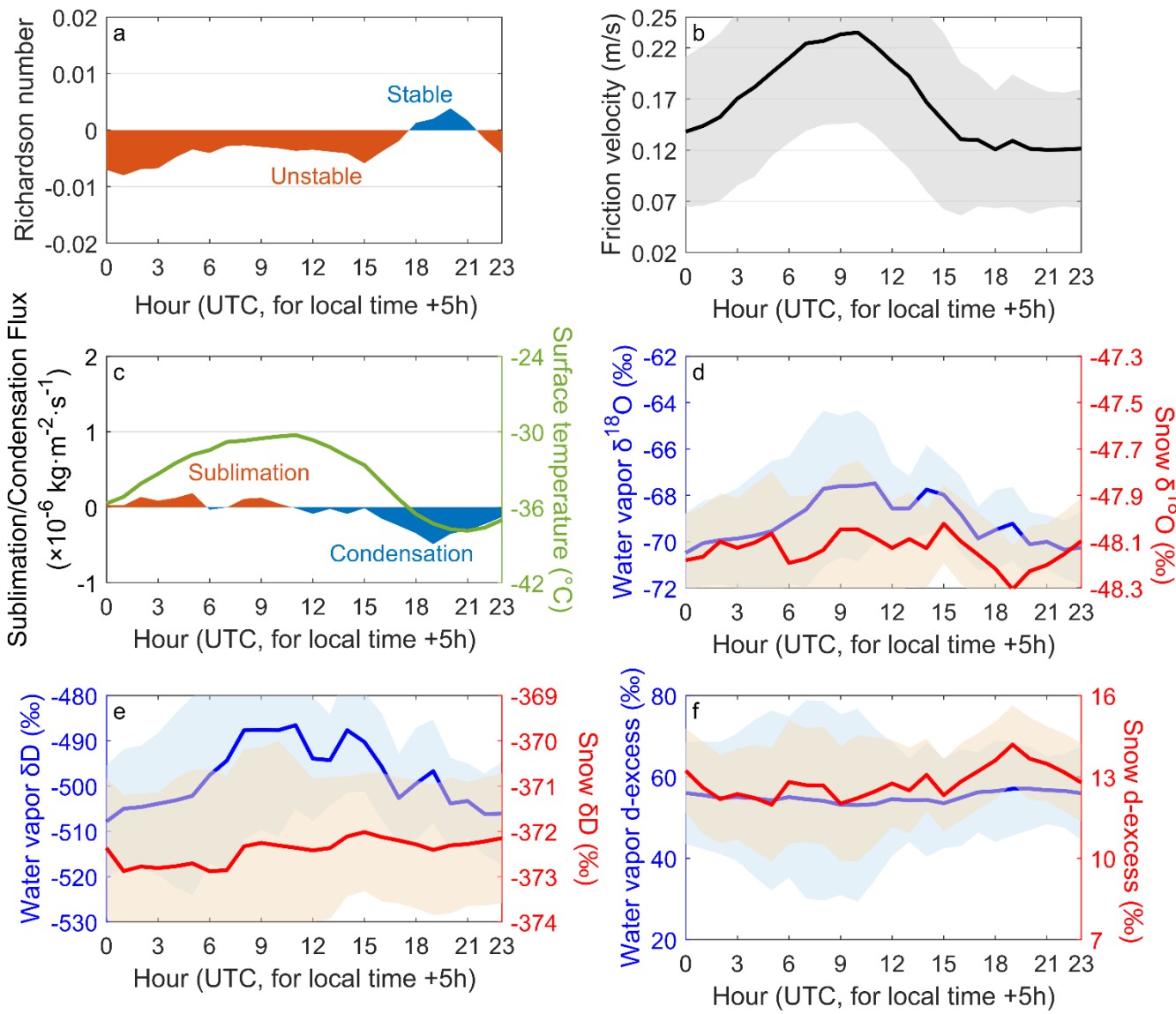

**Figure 5: Same as Figure 4 but for Dome A under highly cloudy conditions in summer.**

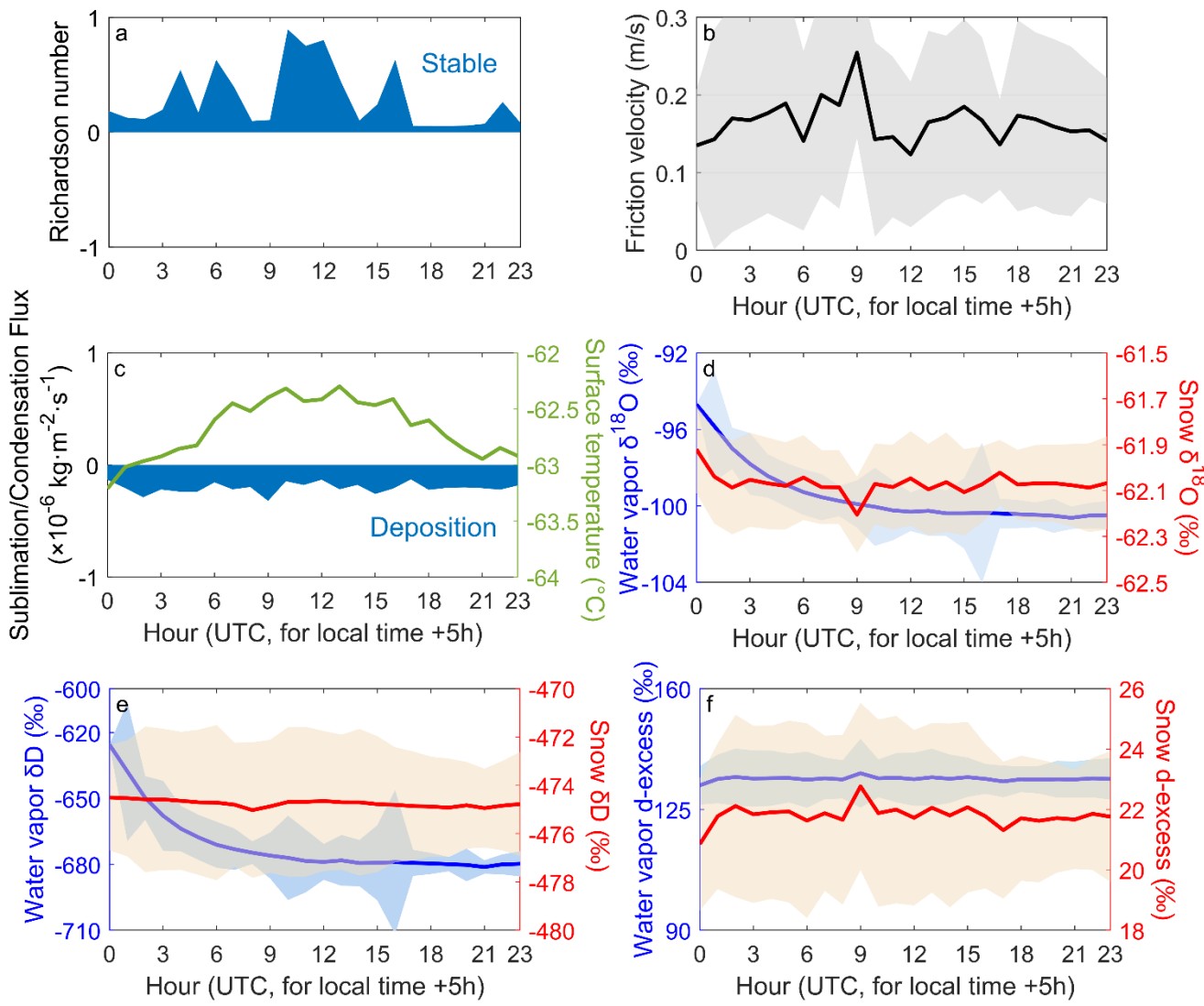

**Figure 6: Same as Figure 4 but for Dome A under winter conditions.**

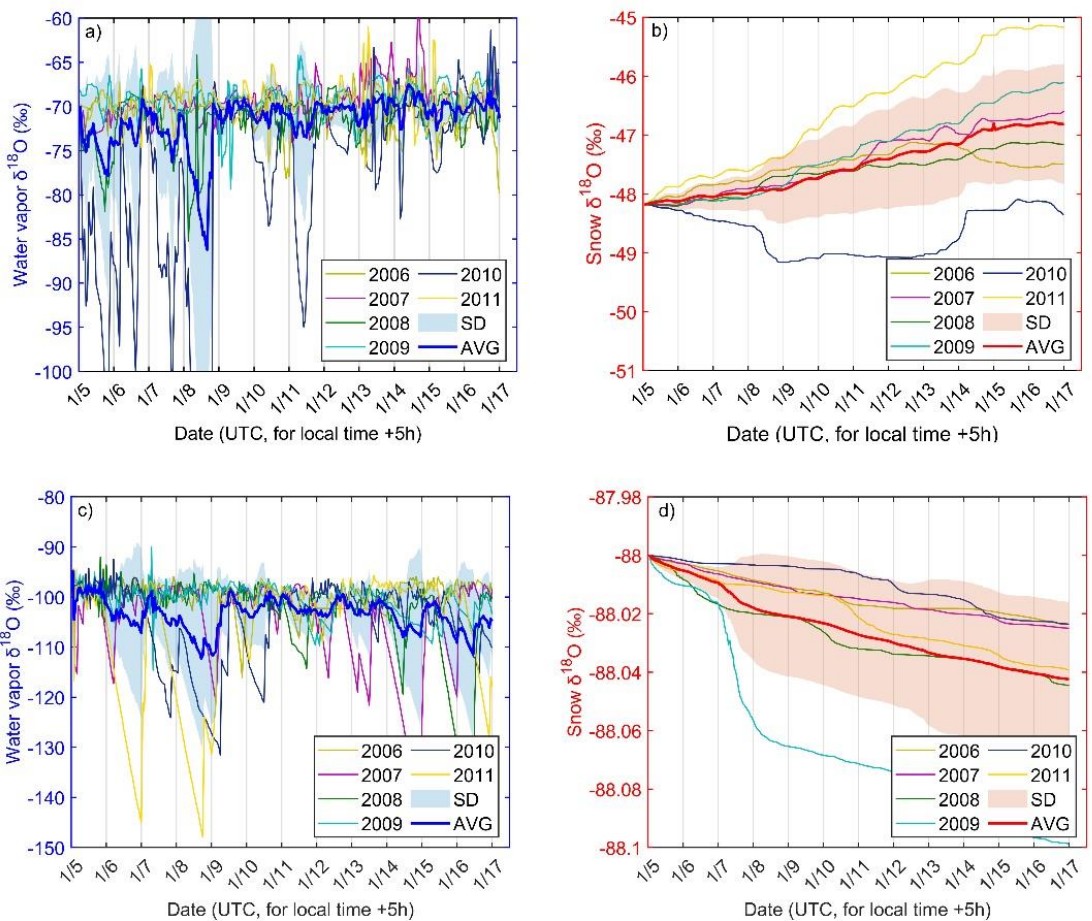


**Figure 7: Continuous simulations of snow and water vapor isotopes at Dome A. Panels a) and b) respectively represent summer simulations over an 11-day period (January 5-16th, 2006-2011), and Panel c) and d) are the same as Panel a) and b), but for wintertime (July 5-16th, 2006-2011). In all panels, the light lines represent the simulated results of water vapor δ18O for each year during the simulation period. The bold solid line and the light blue shadow are the averages (AVGs) and standard deviations (SDs) of the δ18O**
**simulations in each year, respectively.**

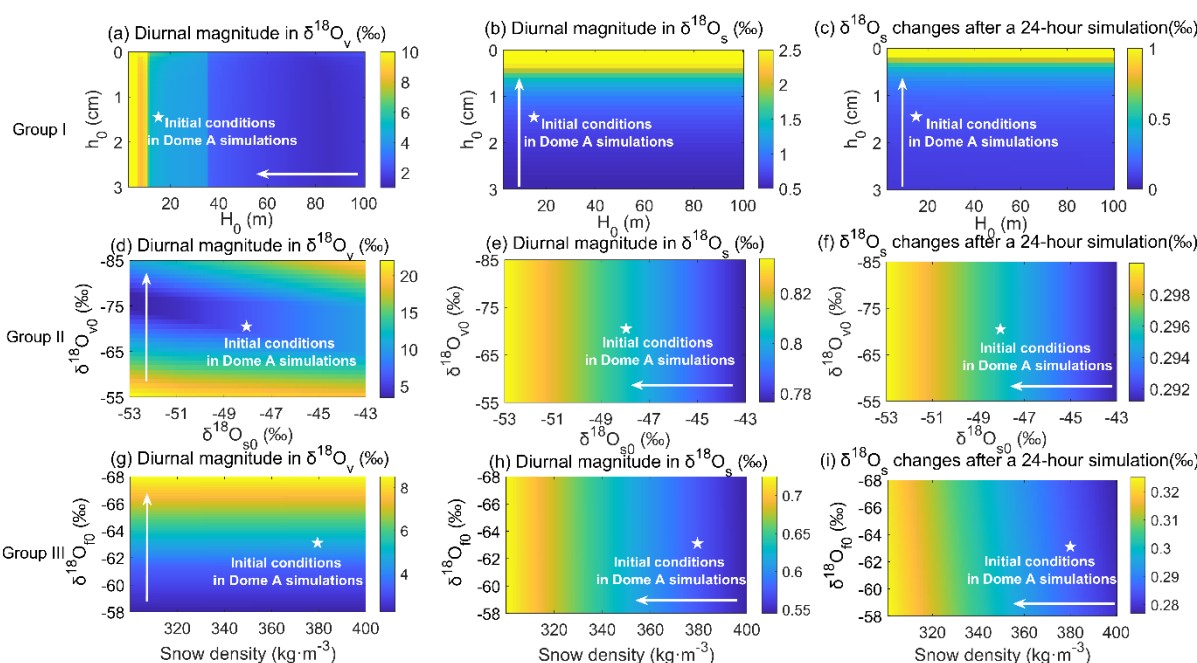

Figure 8: Sensitivity of the modeled results to changes in initial conditions. Panels 8a-8c display the modeled magnitude of $\delta^{18}O$ diurnal variations in water vapor ($\delta^{18}O_v$), the modeled magnitude of $\delta^{18}O$ diurnal variations in surface snow ($\delta^{18}O_s$), and $\delta^{18}O_s$ differences between the ending and starting values varying with different surface snow thicknesses ($h_0$) and boundary layer heights ($H_0$). Panels 8d-8f show the sensitivities of the simulated results to changes in initial water vapor ($\delta^{18}O_{v0}$) and surface snow isotopic composition ($\delta^{18}O_{s0}$), respectively. Panels 8g -8i are the same as 8d-8f, but show the sensitivities to changes in the water vapor isotopic composition of free atmosphere ($\delta^{18}O_{f0}$) and snow density ($\rho_s$). In each subpanel, the white star indicates the initial conditions used in the Dome A simulations with summer clear-sky conditions. The white arrows correspond to the direction of the simulated results with the higher sensitivity.