# Peer review of "A model framework on atmosphere-snow water vapor exchange and the associated isotope effects at Dome Argus, Antarctica:part I the diurnal changes"

_The Cryosphere, 2023_

## Referee Comment (RC2)

**Review of manuscript tc-2023-76 "A model framework on atmosphere-snow water vapor exchange and the associated isotope effects at Dome Argus, Antarctica: part I the diurnal changes" by Tianming Ma et al.**

**General comments:**

The impact of vapor exchange on the surface snow isotopes is uncertain. In their study, the authors present a box model to simulate the impact of vapor exchange on the isotopic composition of the vapor and surface snow. They evaluate a simulation driven by average meteorological diurnal conditions from 11 summer days at DOME-C using vapor isotope measurements and conclude that the model reproduces diurnal variations of the isotopes in the vapor. A similar simulation is repeated for DOME-A for cloudy summer, non-cloudy summer, and winter conditions. A sensitivity analysis to the model input parameters is performed.

The topic is relevant and has the potential to improve our understanding of the impact of water vapor exchange on the surface snow isotopes, in particular, because simulations of the vapor exchange impact on the vapor/snow isotopes are rare. However, there are severe mistakes in the model theory, and the conclusions drawn by the authors are not supported by the applied methodology and presented results. I suggest major revisions in their methodology prior to publication.

My main concern is that the way the authors word their conclusions and their title suggests they provided model estimates of the diurnal variations in the snow and vapor isotopes. In fact, presented simulations are driven by average diurnal cycles of the meteorological parameters. Thus, instead, the authors provide the impact of an average day on initialized snow and vapor isotopes. The presented current results show how a given initial surface snow and vapor isotopic composition could develop within the first 24 hours when applying water vapor exchange.

It is unclear to me why the authors didn't run the simulation based on the meteorological input of individual days instead of stacking and averaging the input data. This limits the simulation time to only 24 hours. Such a short time does not allow for the development of the snow surface over several days. I would consider a minimum of a week spinup time to perform a model simulation in a more equilibrated state as could be expected in nature.

The intuitive approach to obtain an estimate of the average diurnal impact on the isotopes would be to run a longer simulation over several days and give the average daily impact. It seems to me that the authors have the needed data and tools to provide a model simulation over several days, as suggested above. This will improve the manuscript's relevance and provide better applicability of their results to explain observed changes in the snow isotopic composition.

Secondly, there are errors in the calculation of the latent heat flux as well as the calculation of the isotopic flux. Please see the details below. In addition, to my understanding, the latent heat flux is calculated based on already stacked and averaged meteorological data. Since the latent heat flux is non-linearly dependent on these meteorological parameters, the resulting flux based on the averages can diverge severely from a diurnal average of the latent heat flux resulting from hourly calculations. The presented simulations need to be re-run using the corrected latent heat flux calculation.

Another concern is that even when the above-mentioned errors in the latent heat flux calculation are corrected, the conditions for the Monin-Obukov similarity theory (MOST) are often violated under polar conditions. The present study does not discuss the quality of the calculated latent heat flux. If the authors pursue the goal of providing as realistic estimates of the water vapor exchange on the isotopes as possible, they have to make sure that the quality of the driving parameter, the latent heat flux, is well evaluated for similar conditions.

The figures are generally well-made and described. Relevant previous research is cited and discussed. The structure of the manuscript is clear, but the language could be clearer in some parts. The description of the humidity post-processing and the uncertainty analysis needs to be more detailed and clearer. I am looking forward to the revised version of this study.

**Detailed comments:**

- L20-22: This is misleading because the given values refer to the simulated changes when applying one average summer day. The way it is currently written suggests that the given values correspond to the average daily impact on the isotopes when simulating many different summer days.

- L26: I disagree with this statement. Although, in contrast to summer, the meteorological variables don't seem to have a diurnal cycle in winter, the simulation of the isotopic changes shows similar magnitudes to the simulated changes in summer. How do you come to the conclusion that there are no relevant isotopic changes simulated on a diurnal scale in winter? Please clarify what this statement refers to. In that context, please reconsider the use of the term "diurnal cycle" or "diurnal pattern" in the manuscript. For me, a diurnal cycle is a repetitive pattern, i.e., similar values are found at the same time of the day. However, the authors use that term when describing the simulated isotopic change within 24 hours (e.g., L26, L295, L296, L310, L314, L319, L328, L330, L335, L339, L353-355, L361, L403,...). But since the simulated isotopic values are different at 00:00 and 24:00 of the simulated day, the isotopes do not show a diurnal cycle but a change during one day.

- L114-116: This sentence lacks clarity, please reformulate it. The calculation of sublimation and deposition is based on the same formula in the model, so why are two different formulations used here? And please change "followed by a mixing procedure and then uptake of surface snow", e.g., to "and the deposit is mixed into the snow surface layer".

- L124: What does "mainly" and "etc" refer to? Are further input parameters required to run the model? If so, please provide a complete list of all input parameters. If not, please remove the "etc".

- L129-130: Please provide a sufficient discussion of the uncertainty of the calculated latent heat fluxes beyond what is presented in S2 in the supplements. Is there a way to evaluate the quality of the latent heat flux calculations using another dataset (e.g., measured with an eddy covariance system)?

- L134, Eq 1.: The formula that the authors use to calculate the latent heat flux is not correct. Following Berkowicz and Prahm (1982) (B&P82) from solving Eq. 22 for LE, then using H from Eq. 11d with $u^*$ and $\Theta^*$ from Eqs. 11a and 11b, $\Delta u = u_{air} - u_{surface}$ with $u_{surface} = 0$, and $\gamma = \frac{c_p}{L_s}$ you obtain:

$$LE = \rho L_s \kappa^2 \cdot \frac{u_{air}}{log\left(\frac{z_{u,a}}{z_{u,0}}\right) - \Psi_m\left(\frac{z_{u,2}}{L}\right) + \Psi_m\left(\frac{z_{u,1}}{L}\right)} \cdot \frac{q_a - q_s}{R \cdot log\left(\frac{z_{t,a}}{z_{t,0}}\right) - \Psi_h\left(\frac{z_{t,2}}{L}\right) + \Psi_h\left(\frac{z_{t,1}}{L}\right)} \tag{1}$$

Additionally, $L_s$ should not show up on the right side of the formula when giving the expression for $\frac{LE}{L_s}$. Please correct the theory of the box model calculation and re-run all simulations of the study. Furthermore, in Eq. 1, in L134 and L138: There is no time derivative given in B&P82, they use $\Delta$ to indicate the vertical gradient. When using the MOST, the latent heat flux depends on the wind speed as well as the vertical humidity gradient ($q_a - q_s$).

- L135: Please change "$\rho_V$" to "$\rho_a$"

- L145: Where does the chosen value of 0.244 mm for the roughness length come from? The latent heat flux is highly sensitive to the choice of the roughness length. Please provide a sensitivity analysis of the simulated results to the choice of a range of roughness lengths, e.g., 0.1 mm to 2 mm.

- L172: Above (in L138), $RH_i$ is defined as the relative humidity over ice, not for the specific humidity.

- L182, L183: The "$h$" in Merlivat and Jouzel (1979) (M&J79) does not refer to the relative humidity of the air, but to the relative humidity of the air with respect to the surface temperature, i.e., $h = \frac{q_{air}}{q_{sat,surface}}$ (instead of $RH_{air} = \frac{q_{air}}{q_{sat,air}}$). The formulation in M&J79 is really confusing, but their $q_s$ in the formula of $h = \frac{q}{q_s}$ (below Eq. 9 in M&J79), in fact, refers to the "saturated specific humidity at the air-water interface (z=0)", i.e., the saturation specific humidity with respect to the surface temperature, while $q$ is the air specific humidity. It is, thus, not correct to use the relative humidity here, but instead $h = \frac{q_{air}}{q_{s,surface}}$. If this was not the case in the simulations, please correct and re-run them. Otherwise, please be more precise in the description of $RH_i$.

- L189: Where does the expression for $R^t_{EX}$ come from? Because Eq. 2 in Jouzel and Merlivat (1984) is $R^t_{EX} = \alpha_f(R^t_v + 1) - 1$. Please correct this.

- L200-201: Casado et al. (2016) does not present a snow dataset. If the authors refer to the Touzeau et al. (2016) dataset, please add the reference.

- L209: I suggest replacing "representative" with "average". It was initially unclear to me what the authors meant by "stacking" the observed cycles.

- L210: Please remove the "e.g." and "etc." in the parenthesis since the given parameters are the only ones that can be downloaded from the CALVA program.

- L211-216: Is there no surface temperature record available for DOME-C? And if not, why is the surface temperature calculated from ERA-5 model long wave data instead of using the ERA-5 model output of the surface temperature?

- L214: An emissivity of 0.93 seems relatively low to me. Please indicate where this value originates from.

- L216-217: The latent heat flux is calculated based on the averaged meteorological parameters. In my view, it makes more sense to calculate the latent heat flux based on the hourly data and (if needed) stack and average it afterward.

- L217: Please remove the "etc." if no further data is used.

- L220: An average snow density from 2m+ deep snow pits might not be appropriate for the top 1.5 cm. Please provide a sensitivity analysis of the simulation using a range of realistic surface snow densities.

- L234: What does "to fully assess the accumulated isotope effects of atmosphere-snow water vapor exchange." mean? Please rewrite this sentence to clarify on this.

- L250-251: I hardly see any diurnal cycle in the wind speed. In addition, I would argue that the diurnal cycle of the LE differs from the diurnal cycles of Ts and q, since it has a local minimum at 07:00UTC.

- L251, L266: It is not correct to say that the meteorological data are less variable in winter. In fact, all meteorological variables are similarly variable as they have about the same standard deviation. Maybe reformulate to "none of the meteorological variables shows a diurnal cycle" or "in the winter data does not show a diurnal signal."

- L260: Please give the value of the used snow density. How does this value compare to the density taken from Laepple et al. (2018) for the DOME-C simulations?

- L265-266: How is winter defined? Are all hourly data from June-August used?

- L272: Please provide the value of the used $\delta$-T slope in the text.

- L273-274: Where is this comparison presented, and why is this relevant here? Did this comparison influence the initial values of $\delta^{18}Os$? If not, I suggest to remove this.

- L277: Please add the reference (Ma et al., 2020) behind "measurements" again.

- L292: Please clarify: What does the "disequilibrium was included" mean?

- L300-301: The authors mention snow samples for Dome-C in L200-201. An evaluation of the snow isotopic composition development to observations would be very beneficial for the analysis. The simulated changes in snow isotopic composition seem very small compared to variations observed in surface snow samples.

- L314-315: It is not correct to say diurnal cycle here, instead, Fig. 4 shows the simulated change isotopic composition within 24 hours when applying an average summer day observed in January 5-12th.

- L319: What does "diurnal variations" mean? Diurnal maximum minus diurnal minimum? Please define. Maybe the term "diurnal range" is more suitable?

- L339: As mentioned above, the changes in isotopic composition in winter are comparable to the ones in summer

- L354-355: I cannot confirm this statement based on the figures. The different axis ranges make it difficult to compare.

- L359: Please discuss how the simulated results compare to other similar modeling studies, e.g., Wahl et al. (2022) (for Greenland) and Ritter et al. (2016)?

- L356-358: This basically means that the simulated snow isotopic composition does not significantly change after 24 hours of simulation? How much does it change when letting the simulation run longer?

- L364-366: Please reformulate this sentence more clearly.

- L369-370: How is this evident? The authors do not provide evidence for what drives the isotopic composition, neither within their 24-hour simulation nor in a more realistic simulation of a longer time period. The latent heat flux is driven by (1) the near-surface humidity gradient (which, of course, is closely related to the near-surface temperature gradient) and (2) the wind speed. However, this study lacks any evidence that the temperature and humidity drive the surface snow isotopic composition. Please remove this statement.

- L371-372: The authors suggest that wind speed doesn't seem to affect the isotopic composition of the surface snow. However, I'd like to point out that they're using an average wind speed over 11 days, which doesn't show the hourly changes. Thus, such simulation does not allow for a statement that wind speed does not drive the snow isotopic composition at Dome-C. For example, let's say, just to make my point, that 90% of the changes in snow type are due to wind speed. If the wind speed increases linearly from 2 to 7 m/s over the first 5.5 days and then decreases from 7 to 2 m/s in the next 5.5 days, the snow isotopes would change mainly driven by the wind speed. However, the daily average of this wind change would always be 4.5 m/s for all 24 hours. So, when they use the daily average wind speed in their simulation, it makes it seem like wind has no effect on the snow isotopic composition, even though in this example, wind was defined to be the main factor driving the isotopic changes.

- L386: What does this mean: "This could adversely affect changes in atmospheric dynamical conditions between day and night"? Please clarify.

- L387-389: The authors cannot state that: There is no diurnal cycle when averaging, but of course, the wind speed varies on an hourly and daily basis, and the standard deviation is not zero.

- L427-429: Again, the simulated change in the isotopic composition of the vapor is of a comparable magnitude as the changes in summer. What do the authors base this statement on?

- L444-446: The CALVA program states a sentence on its website on how to acknowledge them for the dataset correctly.

- References: The two given references for Ma et al. (2020) can currently not be distinguished in the text.

- Figure 2b: Why is the standard deviation of the latent heat flux so low for cloudy conditions?

- Figure 3: What is $\sigma$ for the simulations? Is it the calculated range from the Monte Carlo simulations, or is it the standard deviation of the Monte Carlo simulations?

- Figure 3 caption: Add water "vapor" isotopic composition.

- Figure 4: Again, please be more precise on what "uncertainty" means.

- Figure 7: Please provide an explanation of the red lines.

- Figure 7 caption: Change "6c and 6d" to "7c and 7d".

- Supplement material S1: The description of the post-processing of the relative humidity ($RH_w$ to $RH_i$) is very difficult to understand.

    - L51-52: Why do you normalize $RH_w$?
    - L52: Which surface temperature is used? The calculated $T_s$ based on ERA-5? If so, please discuss the introduced error by normalizing the observations using model data.
    - L54: (Eq. 15): Do you refer to Eq. 13?
    - L60: What is an "ideal maximum"?
    - L60, L61: What do you mean by "each temperature point"?
    - L63-64: The description of the factor is incomplete (the ratio of $e_s$ with respect to water to $e_s$ with respect to ice. Moreover, why do you only apply this factor for super-saturated conditions? The relative humidity should be corrected with respect to ice for sub-saturation as well.
    - L64: What do you mean by "the rising amplitude of the temperature"?

- Supplement material S2: The description of the uncertainty estimate/error propagation is partly unclear and could be improved. Furthermore, the simulation uncertainties are not sufficiently mentioned and discussed in the main manuscript. A Figure in S2 that shows the calculated uncertainties for all variables could be helpful.

    - L70: How are the "uncertainties" calculated? Is it the standard deviation?
    - L72: Which are "those days"?
    - L75: Which error propagation method is applied? Please provide more details.

R. Berkowicz and L. P. Prahm. Evaluation of the profile method for estimation of surface fluxes of momentum and heat. *Atmospheric Environment (1967)*, 16(12):2809–2819, 1982.

M. Casado, A. Landais, V. Masson-Delmotte, C. Genthon, E. Kerstel, S. Kassi, L. Arnaud, G. Picard, F. Prie, O. Cattani, et al. Continuous measurements of isotopic composition of water vapour on the east antarctic plateau. *Atmospheric Chemistry and Physics*, 16(13):8521–8538, 2016.

J. Jouzel and L. Merlivat. Deuterium and oxygen 18 in precipitation: Modeling of the isotopic effects during snow formation. *Journal of Geophysical Research: Atmospheres*, 89(D7):11749–11757, 1984.

T. Laepple, T. Münch, M. Casado, M. Hoerhold, A. Landais, and S. Kipfstuhl. On the similarity and apparent cycles of isotopic variations in east antarctic snow pits. *The Cryosphere*, 12(1):169–187, 2018.

L. Merlivat and J. Jouzel. Global climatic interpretation of the deuterium-oxygen 18 relationship for precipitation. *Journal of Geophysical Research: Oceans*, 84(C8):5029–5033, 1979.

F. Ritter, H. C. Steen-Larsen, M. Werner, V. Masson-Delmotte, A. Orsi, M. Behrens, G. Birnbaum, J. Freitag, C. Risi, and S. Kipfstuhl. Isotopic exchange on the diurnal scale between near-surface snow and lower atmospheric water vapor at kohnen station, east antarctica. *The Cryosphere*, 10(4):1647–1663, 2016.

A. Touzeau, A. Landais, B. Stenni, R. Uemura, K. Fukui, S. Fujita, S. Guilbaud, A. Ekaykin, M. Casado, E. Barkan, et al. Acquisition of isotopic composition for surface snow in east antarctica and the links to climatic parameters. *The Cryosphere*, 10(2):837–852, 2016.

S. Wahl, H. C. Steen-Larsen, A. Hughes, L. J. Dietrich, A. Zuhr, M. Behrens, A.-K. Faber, and M. Hörhold. Atmosphere-snow exchange explains surface snow isotope variability. *Geophysical Research Letters*, 49(20), 2022. doi:10.1029/2022GL099529.

---

## Author Comment (AC1)

**Response to Reviewer #1's comments**

**General comments**

*This manuscript considers the exchange between water molecules between the firn and the atmosphere, and the impact it can induce on the change of isotopic composition in extremely low accumulation regions of Antarctica. Using the results from Dome C as an analogue for Dome A is a clever strategy that can yield promising results to how to explain the impact of surface processes on the future Dome A ice core. The study takes into account the variations of stability of the atmosphere with systematic calculations of the Richardson number and developed three case studies associated with two sets of summer conditions (clear sky and cloud), and one set of winter conditions.*

*While the authors used a rather classical set of equations to evaluate the isotopic exchanges during sublimation and condensation, it seems not pertinent here, as it ignores major contributors to the boundary layer processes and only consider the system as a closed box without exchange with the free atmosphere. As a result, the results do not match the observations that were made for the surface snow isotopic composition at Dome C, even though, it is supposed to be the case study used to parametrise the model.*

*I suggest profound modifications to the model, which take into account exchanges between the atmospheric boundary layer and the free atmosphere on top of the surface processes, and which would match the surface snow changes, at least in order of magnitude, before considering the manuscript for publication.*

**Response:** We greatly appreciate the reviewer's insightful comment on the physical mechanism of our model. We agree that realistically exchanges between the atmospheric boundary layer and the free troposphere on top of the surface processes should be considered. It was our originally plan that we wanted to explicitly focused on how much changes on snow isotopes can be induced by processes at the air-snow interface alone. This may not reflect the real changes but can reveal the most potential effects associated with the processes at the air-snow interface. Thanks to the reviewer's suggestion, that we realized that it might be better to include the free troposphere which will make the result more comparable with the observations. Therefore, in the revised manuscript, we included the mass exchange between the boundary layer and the free troposphere by adding a third box as illustrated in the revised Figure 1. The calculations and equations were also changed to reflect the modifications to the physical mechanisms in the model. But we wanted to note that, with including the effects of exchanges between the boundary layer and free troposphere, the main conclusion of the manuscript doesn't change (the magnitude of modeled changes are affected but still in the same direction).

[Figure]

**Figure 1:** Schematic diagram of the box model used in this study (Revised version).

**Major Comments:**

**1)** *The box model developed by the authors was parametrised against vapour measurements obtained at Dome C, in order to compensate for lack of measurements at Dome A. The outputs of the model predict changes of vapour isotopic composition that seem realistic, but it is not the case for the changes of snow isotopic composition for which the variations are extremely small (less than 0.02‰) while the observed changes are around 2‰ during a typical night (Casado et al., 2018). The relative changes of snow and vapour isotopic compositions during a typical clear sky night were modelled in this manuscript, and suggested that a closed box model (which is de facto what the authors have implemented since no exchanges between the free atmosphere and the boundary layer are taken into account) is not realistic for this type of event.*

**Response:** Thanks for pointing out this. Indeed in the original model framework, the modeled results on snow isotopic changes cannot match the observations. In the revised manuscript, this is addressed by including the effect of exchanges between boundary and free troposphere. In particular, the new results indicate that simulated changes in snow isotopic composition are significantly larger (i.e., ~ 0.02‰ for $\delta^{18}O$) than the original model (i.e., ~ 0.02‰ for $\delta^{18}O$) at Dome C. Especially, for the case the reviewer mentioned, i.e., a typical night with a frost event on 6-7th January 2015, the diurnal changes of newly simulated results between the maximum and minimum can reach 2‰ for snow $\delta^{18}O$ (as demonstrated in Figure 2). This magnitude is in line with the observations for snow isotopes from Casado et al. (2018) which is ~ 2‰.

In the revised manuscript, we have re-run all simulations under Dome C conditions and three different cases at Dome A using the adjusted model. The simulated results within a 24-hour period were displayed in Figures 3-6 of the revised manuscript.

[Figure]

**Figure 2:** The changes of snow isotopes and water isotopic composition in the near surface atmospheric layer in Jan 6-7[th], 2015 at Dome C

**2)** *Another aspect that suggests that exchanges between the boundary layer and the free atmosphere must happen is the Richardson number. Indeed, for negative Richardson numbers, the atmosphere must be quite convective, which suggest that the boundary layer exchanges with both the surface snow and the free atmosphere.*

*The atmosphere is qualified as stable for any positive Richardson number, yet, it seems that some studies suggest that some amount of mixing remains quite strong for 0 < Ri < 0.1 (Zilitinkevich et al., 2007) . This could be discussed.*

**Response:** Thanks for this valuable suggestion. In our original simulations, we assumed that unstable conditions for atmosphere stability only existed under negative Richardson numbers. Based on this assumption, we considered how mixing between the boundary layer, surface snow, and the free troposphere can affect the water vapor isotopic composition in the near-surface atmospheric layer and snow isotopes during the warming phase with negative Richardson numbers. However, as pointed by the reviewer that Zilitinkevich et al. (2008) suggested mixing can occur under positive Richardson numbers as well. If this is true, our original simulations for the water vapor isotopic composition in the near-surface atmospheric layer may be underestimated in the cooling phase.

To test the relationship between mixing occurrence conditions and Richardson numbers, we ran simulations for Dome C taking into account mixing when Ri<0 and Ri<0.1. As shown in Figure 3, the case with Ri<0 (Case II) indeed underestimates the water vapor isotopic composition in the near-surface atmospheric layer during the cooling time. Based on this comparison, in the revised manuscript, we incorporated mixing into the modeling once Ri<0.1 (Case I) in addition to the original consideration with Ri<0. Discussion on taking into account Ri<0.1 was added in supplementary
information (Text S3) of the revised manuscript, and in the main text all results were
updated with consideration of the mixing when Ri<0.1.

[Figure]

**Figure 3:** The comparison of water vapor isotopic composition between the simulated
and observed changes at Dome C. Two simulated cases are presented here to discuss
the occurrence condition of mixing. In case I, the mixing is assumed to only happen
when Ri<0 in the cooling phase, while case II also considers the occurrence of mixing
when Ri<0.1 in the cooling phase.
**3)** *Some limited vapour data exist at Dome A (Liu et al., 2022). While these data might*
*be difficult to compare to your results, in particular consider how high the d-excess is,*
*which could be associated with calibration issues, it should be discussed.*
**Response:** Thanks for this suggestions. Actually before finalizing the manuscript, we
have discussed with the leading author of the Liu et al. (2022) study, but we noted that
due to the harsh environment, direct observations of water vapor as the Liu et al did is
difficult and the calibration can induce large issues. In addition, their measured sites are
actually not exact the same at Dome A (~100 km away). In the end we didn't choose to
compare this dataset. But since the reviewer asked, in the revised manuscript, we
compared our simulations at Dome A with the data of water vapor $\delta^{18}$O, $\delta$D, and d-
excess from Liu et al. (2022). We found that both our simulations and observations
exhibit diurnal patterns, with high values occurring during the warming phase (daytime)
and low values during the cooling phase (nighttime). However, we note that the
magnitude of the observed diurnal changes in water vapor $\delta^{18}$O and d-excess at sites
near Dome A are very large, over 40‰ and 200‰, respectively. This could be due to
calibration drift caused by the extremely cold and dry conditions during the
measurements at the nearest Dome A site.

Therefore, in Section 4 of the revised manuscript, we only qualitative compare and discuss the similarities and/or differences between simulations and observations, without delving into quantitative details.

**4)** *Considering how fundamental these changes are, an updated version of the manuscript could have completely different conclusions.*

**Response:** We really appreciate the reviewer's comments. By including the effect of exchanges between boundary and free troposphere, the modeled results indeed differ a lot compared to the original model. However, the modeled changes in snow and vapor isotopes are still in the same direction (the magnitude or absolute values differ), and the main conclusion stays the same as that the air-snow exchange would lead to diurnal variations in atmospheric water vapor $\delta^{18}O$ and $\delta D$ by 4.75±2.15 ‰ and 28.79±19.06 ‰ under summer clear-sky conditions at Dome A, with corresponding diurnal variations in surface snow $\delta^{18}O$ and $\delta D$ by 0.81±0.24 ‰ and 1.64±2.71 ‰, respectively. These values become smaller compared to those in the previous simulations. After 24-hour simulation, snow water isotopes were enriched under clear-sky conditions. However, there is no or very little enrichment for snow water isotopes under cloudy conditions, which is different with the previous simulations. Under winter conditions at Dome A, the model still indicates the diurnal change in atmospheric and surface snow water isotopes are not significant, but the model predicts more or less depletions in snow $\delta^{18}O$ and $\delta D$ in the period of 24-hour simulation, opposite to the results under summer clear-sky conditions. This suggests that the air-snow vapor exchange tends to enlarge snow water isotope seasonality.

**Supplementary response**

The revised figures in the main text are as follows:

[Figure]

**Figure 3:** Model simulated diurnal variations of water vapor and snow isotopic
compositions at Dome C along with the observations. (a) water vapor $\delta^{18}O$, (b) water
vapor $\delta D$, (c) water vapor d-excess and (d) snow isotopes. In panels (a)-(c), blue solid
line represents the observations of water vapor isotopic composition ($\delta v_{obs}$) with the
light grey shaded area as the uncertainties ($\pm 1\sigma$). The red solid line and the light red
shaded area depicts the modeled variations of water isotopic composition ($\delta v_{sim}$) and
correspondingly uncertainties ($\pm 1\sigma$). In panel (d), the diurnal variations of modeled
snow $\delta^{18}O$ and d-excess are shown as the black solid line and light blue solid line,
respectively. Their uncertainties are also displayed with shaded areas like $\delta v_{obs}$ and $\delta v_{sim}$
in first three panels. The method for uncertainties estimation can be seen in SI (Texts
S2).

[Figure]

**Figure 4:** The simulated hourly mean vapor exchange flux and variations in atmospheric water vapor and snow isotopes under summer clear-sky conditions at Dome A: (a) Richardson number, (b) friction velocity, (c) vapor exchange flux, (d) snow and water vapor $\delta^{18}O$, (e) snow and water vapor $\delta D$, (f) snow and water vapor d-excess. The uncertainties for each variable are displayed by shaded area in each subpanel.

[Figure]

**Figure 5:** Same to Figure 4 but for Dome A under highly cloudy conditions in summer.

[Figure]

**Figure 6:** Same to Figure 4 but for Dome A under winter conditions.

**End of the responses to Reviewer #1**

**Reference**

Casado, M., Landais, A., Picard, G., Münch, T., Laepple, T., Stenni, B., et al.: Archival processes of the water stable isotope signal in East Antarctic ice cores, The Cryosphere, 12(5), 1745-1766, doi: 10.5194/tc-12-1745-2018, 2018.

Liu J., Du Z., Zhang D., Wang S.: Diagnoses of Antarctic inland water cycle regime: Perspectives from atmospheric water vapor isotope observations along the transect from Zhongshan Station to Dome A, Frontiers in Earth Science, 10, doi: 10.3389/feart.2022.823515, 2022.

Zilitinkevich, S.S., Esau, I.N.: Similarity theory and calculation of turbulent fluxes at the surface for the stably stratified atmospheric boundary layer, Boundary-Layer Meteorology, 125, 193–205, doi: 10.1007/s10546-007-9187-4, 2007.

---

## Author Comment (AC2)

**Response to Reviewer #3's comments**

**General comments**

**1)** *This manuscript describes a closed box model assuming no atmospheric mixing and simulations of the effect of a mean diurnal cycle at Dome C (using observations) and Dome A (using atmospheric reanalyses and assumptions as inputs). The current title does not reflect the content and the conclusions are not well supported by the analyses and the underlying assumptions in the modelling methodology.*

**Response:** Thanks for reviewing the manuscript and the valuable comments. In the revised manuscript, we have addressed the issue of "no atmospheric mixing" by including exchanges between the boundary layer and free troposphere. Additionally, the calculations of isotope mass balance have also been modified following the new model structure. Using this modified model, we have conducted new simulations at Dome C and Dome A. The discussion has been reformulated in the revised manuscript based on the new simulated results and the feedback from the reviewers.

**2)** *The long introduction gives a good scene setting for the study, which addresses an important topic, but fails to describe the modelling framework in the context of other studies, and fails to provide a clear comparison of the meteorological and snow conditions between Dome C and Dome A (and what are the similarities and differences that need to be accounted for in comparing results for these two sites, for diurnal variations, clear and cloud sky, and winter vs summer conditions).*

**Response:** We would like to express our gratitude to the reviewer for this comment. In response to the comment, we have made some revisions to the manuscript. Specifically, in the revised manuscript, we have added a comparison of the meteorological and snow conditions between Dome C and Dome A in Section 2.2.2. The comparison of isotopic values for these two sites were also conducted at the result section (Section 3.2). Additionally, we have included a new paragraph in Section 4 to discuss the similarities and differences in diurnal variations between these two sites. We hope that these revisions will enhance the clarity and comprehensiveness of our work.

The added statement in Section 4 is as follows:

"The diurnal amplitude of water vapor $\delta D$ across East Antarctic interior region appears to vary spatially. The modeled value of 28.79‰ at Dome A is slightly less than the averaged observations of 36±6‰ at Kohnen station and the in-situ measurements of 38±2‰ at Dome C (Ritter et al., 2016; Casado et al., 2016). The difference between the two latter locations can be explained by a smaller amplitude of diurnal temperature cycle (8.7°C) at Kohnen station, relative to that in Dome C (11.1°C). However, there still exists a discrepancy in water vapor $\delta D$ amplitude when the peak-valley gap of diurnal temperature cycle is the same at Dome A and Dome C. Such an anomaly pattern can be attributed to atmospheric dynamical conditions linked with wind speed. At Dome A, the daily mean wind speed of 2.8 m/s is lower than 3.3 m/s in Dome C and 4.5 m/s in Kohnen station during summer. A small wind speed corresponds to the relatively weak air convection in horizontal orientation. Due to the coupling between upper and lower atmospheric layer, vertical turbulent mixing may decrease with the weakened air convection in the atmospheric near-surface layer (Casado et al., 2018).

This change can attenuate molecular exchange between surface snow and water vapor.
In parallel, the decrease of vertical turbulence may result in a less efficient turbulent
diffusion of water molecules and an elevated contribution of molecular diffusion during
air-snow exchange. Changes in water vapor diffusion pathways increase kinetic
fractionation and reduce effective isotopic fractionation of water isotopes, leading to a
muted fluctuation of modelling water vapor δD in combination with less mass exchange.
The surface snow δD displays the synchronization change and different amplitude in
diurnal cycles, in accordance with the comparisons of water vapor δD between Dome
A and Kohnen Station. The similar trend of snow δD is originated from similar
temperature variations on a diurnal scale, because surface snow isotopic composition is
mainly influenced by temperature-controlled fractionation of water isotopes during air-
snow vapor exchange. This relationship also suggests that the difference in temperature
amplitude could be playing a role in the unequal amplitude of snow δD."

**3)** *The description of the model has flaws in the equations for latent heat flux and*
*possibly in the use of relative humidity in the atmosphere and not relative to surface*
*temperature for fractionation coefficients. The information provided in supplementary*
*information is very difficult to understand.*
**Response:** Thanks for this comment. We found that the formulations used in the latent
heat flux calculation is not correct, following Berkowicz and Prahm (1982) (B&P82)
and the suggestion from the reviewer #2. In addition, the fractionation coefficients
calculations should rely on the humidity with respect to the surface layer and surface
temperature, rather than relative humidity and air temperature.
In the revised manuscript, we first have made modifications to the calculations of
latent heat flux. Specifically, we have revised the calculations as follows:

$$Ex = LE/Ls = -\rho_a u^* q^* \tag{1}$$

where $\rho_a$ is dry air density varying with observed air temperature ($T_a$) and pressure ($P_a$), $L_s$ is
sublimation heat constant, u* and q* are friction velocity and specific humidity turbulent scale,
respectively. Where u* and q* are respectively defined as:

$$u^* = \frac{ku_z}{log\left(\frac{z}{z_0}\right) - \Psi_M\left(\frac{z}{L}\right)} \tag{2}$$

$$q^* = \frac{k(q_a - q_s)}{log\left(\frac{z}{z_0}\right) - \Psi_M\left(\frac{z}{L}\right)} \tag{3}$$

We would also like to apologize for any confusion caused by our imprecise
description of the relative humidity correction and fractionation coefficients. In the new
version of the supplementary information, the text description for the relative humidity
correction have been rewritten to be more clear and accurate. Additionally, we would
like to clarify that it is the surface snow temperature ($T_s$) that controls isotopic
fractionation during air-snow vapor exchange. Thus, the surface temperature were used
to calculate fractionation coefficients, instead of air temperature. We will make the
necessary corrections to the related description in Section 2.1.1 of the revised
manuscript.

The revised supplementary information for humidity correction are as follows:

"The raw data of relative humidity (RH) at height z is the relative humidity with respect to the water surface ($RH_w$), measured with the HMP35D humidity probe (Xiao et al., 2008; Ding et al., 2022). The $RH_w$ can be expressed as a percentage:

$$RH_w = e_w/e_w^s \times 100\% \tag{S2}$$

where $e_w$ is the water vapor pressure of air (Pa), and $e_w^s$ is the saturated vapor pressure with respect to the water surface at the air temperature (Pa) which can be calculated using the Clausius-Clapeyron equation. When calculating the effective fractionation factor ($\alpha_f$) in the model (Eq: (15) in the main text), the $RH_w$ were converted to the relative humidity over ice at the temperature of the air ($RH_i$). The conversion between $RH_i$ and $RH_w$ was proposed based on the calibration procedures of Anderson et al. (1984). The details are as follows: 1) The $RH_w$ observations were firstly rescaled using the maximum $RH_w$ of all measured values at each air temperature point ($T_a$),

$$RH_w^{'} = RH_w (T_a)/ RH_w^{max} (T_a) \tag{S3}$$

2) $RH_w^{'}$ values were then converted to $RH_i$ using Eq: (S4) :

$$RH_i = (e_w^s (T_a) /e_i^s (T_a)) \times RH_w^{'} \tag{S4}$$

where $e_i^s$ represents the saturated vapor pressure with respect to ice at the air temperature (Pa). Like $e_w^s$, $e_i^s$ was calculated by the Clausius-Clapeyron equation. Based on Eq: (S3) and Eq: (S4), we obtained $RH_i$ as the final result.

In addition, the relative humidity of the air with respect to the surface temperature (h) in Eq: (14) can also be converted from $RH_w$ observations. The first step of procedures for h conversion is the rescaling $RH_w$ based on Eq: (S3), same to the $RH_i$ conversion. The second step is h calculation using the saturated vapor pressure with respect to ice at the surface temperature (Eq: (S5)).

$$h = (e_w^s (T_a) /e_i^s (T_s)) \times RH_w^{'} \tag{S5}$$

**4)** *The choice of performing simulations driven by a mean diurnal cycle instead of using the actual wealth of observations is unclear and the implications should be discussed. I am puzzled by how wind effects are accounted for when averaging conditions.*

**Response:** Thanks for this comment. We chose to use the mean stacked conditions to conduct simulation since we wanted to highlight the effects of air-snow exchange in a general case. But in order to avoid confusion, in the revised manuscript, the simulations were conducted using continuous meteorological input for each individual day during the studied period at Dome C. This allowed us to calculate the average diurnal changes in water vapor isotopic composition and snow isotopes. However, for the Dome A case, the selected days for clear-sky, cloudy, and winter conditions were not continuous, making it difficult to conduct simulations as was done for Dome C. Instead, we were only able to use the model for one day to simulate the diurnal changes in snow and water vapor isotopes, after a week of spin-up time (as shown in Figures 4-6 in the revised manuscript). This allows to evaluate the effects of air-snow exchange under representative meteorological conditions.

In addition, we also reconsidered the effect of wind speed on simulations during atmosphere-snow water vapor exchange. In the revised manuscript, a new case simulation was presented to test the effect of wind speed variability on atmosphere-snow water vapor exchange. Specifically, we analyzed the response of water vapor and snow isotopic composition to the conditions of 1) a significant diurnal cycle of wind versus that with averaged wind speed. The results, as shown in Figure 1, suggest that strong variability in wind speed will enlarge the variations in latent heat, leading to a more significant diurnal change in water vapor isotopes and snow isotopes, but the for a longer time, there would be days with diurnal wind cycle both smaller or bigger than the mean, so the result with the mean wind pattern is more representative.

[Figure]

**Figure 1:** The comparison of water vapor isotopic composition between two simulated cases at Dome A. The simulations in two cases were driven using the averaged wind speed (Case I) and the strong diurnal changes in wind speed (Case II).

**5)** *There should be at least a more detailed comparison between the Dome C and Dome A characteristics (including comparison of meteorological conditions and ERA5 results at both sites), instead of current Table 1 (where assumptions versus observational based information should be differenciated).*

**Response:** Thanks for this suggestion. In the revised manuscript, we have added content to compare the meteorological conditions at Dome C and Dome A in Section 2.2.2, and the impacts of these conditions on the modeled water vapor and snow isotopes are discussed in Section 4.

**6)** *The assumptions displayed in Figure 1 should be discussed in the context of available information, including the Richardson number, regarding atmospheric exchanges (the closed box assumption validity).*

**Response:** Thanks for the valuable suggestion. We have incorporated these into the revised manuscript by discussing the assumptions related to the occurrence conditions of the air-mass renewal process associated with the Richardson number, as well as the isotopic fractionation during sublimation and deposition. Additionally, we have addressed the setting of initial conditions through some original and new sensitivity tests.

Here, we will provide the discussion of the occurrence conditions of the air-mass renewal process in the supplementary information:

"To determine the correlation between mixing occurrence conditions and Richardson numbers, we ran simulations for Dome C, taking into account mixing when $Ri<0$ and $Ri<0.1$. As shown in Figure 2, the case with $Ri<0$ did indeed underestimate the water vapor isotopic composition in the near-surface atmospheric layer during the cooling time. Based on this comparison, we incorporated mixing into the modeling once $Ri<0.1$."

[Figure]

**Figure 2:** The comparison of water vapor isotopic composition between the simulated and observed changes at Dome C. Two simulated cases are presented here to discuss the occurrence condition of mixing. In the case I, the mixing is assumed to happen when $Ri<0$ in the cooling phase. The case II for the occurrence conditions of mixing is $Ri<0.1$ in the cooling phase.

**7)** *The authors should reflect on what their model explicitly implies in terms of behaviour, and what is effectively "validated" from their approach which does not resolve the diurnal variations in snow measured at Dome C. This physics-based approach is missing.*

**Response:** Thank you for bringing this to our attention. We have resolved the issue by making modifications to the physical mechanism of our model (Figure 1), as outlined in our previous response to general comments. We then conducted simulations under Dome C conditions and three different cases at Dome A using the updated model. The simulated results for a 24-hour period are presented in Figures 3-6 of the main text (at the end of this response). The new results indicate that the changes in snow isotopic composition are significantly greater than the original $\delta^{18}O$ simulations of 0.02‰ at Dome C. During a typical night, such as the frost event on January 6-7, 2015, the diurnal changes of the newly simulated results between the maximum and minimum can reach 2‰ for snow $\delta^{18}O$ (as shown in Figure 3). This magnitude is consistent with the observations for snow isotopes from Casado et al. (2018).

[Figure]

**Figure 3:** The changes of snow isotopes and water isotopic composition in the near surface atmospheric layer during the 6-7th Jan, 2015 at Dome C.

**8)** *For these reasons, major revisions are needed, first to ensure accurate equations in the model, and then to reflect on the limitations and suitability of the core assumptions of the closed box model to address these questions, and third regarding the average diurnal cycle approach, and fourth regarding the detailed comparison between Dome C and Dome A (well beyond "validating" and "applying" this model at the two sites).*

**Response:** Thank you for the helpful comment. Several significant changes were made to the model structure to reflect reviewer's suggestions. Specifically, we have added a third box to represent the free atmosphere layer. The calculations and equations were also updated to reflect the modifications made to the physical mechanism of the model. We also have presented new assumptions for initial conditions and air mass renewal occurrence conditions, which enable the model to run effectively. Furthermore, the simulations were continuously conducted using meteorological observations recorded hourly. Finally, we have included a comparison between Dome C and Dome A in the Discussion section of the revised manuscript (Details can be seen in response to Comment #2 and Comment #6). After all of these modifications, in addition to that arisen by other reviewers, the main conclusion of the manuscript stays the same: The diurnal variations in atmospheric water vapor $\delta^{18}O$ and $\delta D$ can reach 4.75±2.15 ‰ and 28.79±19.06 ‰ under summer clear-sky conditions at Dome A, with corresponding diurnal variations in surface snow $\delta^{18}O$ and $\delta D$ by 0.81±0.24 ‰ and 1.64±2.71 ‰, respectively. After 24-hour simulation, snow water isotopes were enriched under clear-sky conditions. However, there is no or very little enrichment for snow water isotopes under cloudy conditions. Under winter conditions at Dome A, the model still indicates the diurnal change in atmospheric and surface snow water isotopes are not significant, but the model predicts more or less depletions in snow $\delta^{18}O$ and $\delta D$ in the period of 24-hour simulation, opposite to the results under summer clear-sky conditions. This suggests that the air-snow vapor exchange tends to enlarge snow water isotope seasonality.

**Supplementary response**

The revised figures in the main text are as follows:

[Figure]

**Figure 2:** Stacks of diurnal cycles of meteorological parameters and the calculated latent heat under summer clear-sky conditions (a), summer highly cloudy conditions (b), and winter conditions (c) at Dome A. The hourly data for air temperature, relative humidity, air pressure and wind speed were averaged by AWS observations over those selected days. The diurnal variations for other three parameters were calculated based on hourly observations. In each panel, the solid line with marks represents the average and the grey shadow is the standard deviation. The background color of pink and blue corresponds to the period dominated by sublimation and deposition, respectively, in a diurnal cycle.

[Figure]

**Figure 3:** Model simulated diurnal variations of water vapor and snow isotopic
compositions at Dome C along with the observations. (a) water vapor $\delta^{18}O$, (b) water
vapor $\delta D$, (c) water vapor d-excess and (d) snow isotopes. In panels (a)-(c), blue solid
line represents the observations of water vapor isotopic composition ($\delta v_{obs}$) with the
light grey shaded area as the uncertainties ($\pm 1\sigma$). The red solid line and the light red
shaded area depicts the modeled variations of water isotopic composition ($\delta v_{sim}$) and
correspondingly uncertainties ($\pm 1\sigma$). In panel (d), the diurnal variations of modeled
snow $\delta^{18}O$ and d-excess are shown as the black solid line and light blue solid line,
respectively. Their uncertainties are also displayed with shaded areas like $\delta v_{obs}$ and $\delta v_{sim}$
in first three panels. The method for uncertainties estimation can be seen in SI (Texts
S2).

[Figure]

**Figure 4:** The simulated hourly mean vapor exchange flux and variations in
atmospheric water vapor and snow isotopes under summer clear-sky conditions at
Dome A: (a) Richardson number, (b) friction velocity, (c) vapor exchange flux, (d) snow
and water vapor $\delta^{18}O$, (e) snow and water vapor $\delta D$, (f) snow and water vapor d-excess.
The uncertainties for each variable are displayed by shaded area in each subpanel.

[Figure]

**Figure 5:** Same to Figure 4 but for Dome A under highly cloudy conditions in summer.

[Figure]

**Figure 6:** Same to Figure 4 but for Dome A under winter conditions.

**End of the responses to Reviewer #3**

**Reference**

Casado, M., Landais, A., Picard, G., Münch, T., Laepple, T., Stenni, B., et al.: Archival processes of the water stable isotope signal in East Antarctic ice cores, *The Cryosphere*, 12(5), 1745-1766, doi: 10.5194/tc-12-1745-2018, 2018.

Ritter, F., Steen-Larsen, H. C., Werner, M., Masson-Delmotte, V., Orsi, A., Behrens, M., et al.: Isotopic exchange on the diurnal scale between near-surface snow and lower atmospheric water vapor at Kohnen station, East Antarctica, *The Cryosphere*, 10(4), 1647-1663, doi: 10.5194/tc-10-1647-2016, 2016.

Wahl, S., Steen-Larsen, H. C., Reuder, J., & Hörhold, M.: Quantifying the Stable Water Isotopologue Exchange Between the Snow Surface and Lower Atmosphere by Direct Flux Measurements, *Journal of Geophysical Research: Atmospheres,* 126(13), doi: 10.1029/2020jd034400, 2021.

---

## Author Response (AR1)

Dear Dr. Smith
Please find our revised manuscript "**A model framework on atmosphere-snow water**
**vapor exchange and the associated isotope effects at Dome Argus, Antarctica: part**
**I the diurnal changes** " by *Ma et al.* We are grateful to you and the reviewers for the
constructive comments and suggestions which significantly improve the manuscript. In
the revised manuscript, we have made substantial revisions according to the
comments/suggestions. Below we briefly described the main comments and our
responses. Detailed responses can be found in the point-to-point response file
One of the main comments/suggestions was on the components and the physical
mechanisms of the model. The reviewers suggested the exchanges between the
atmospheric boundary layer and the free troposphere should also be considered in the
model. In response, we added a third box into the model structure and then modified
the calculations of mass and isotopic balance during atmosphere-snow vapor exchange
accordingly. The calculations of the latent heat flux and humidity in the model were
also modified according to the reviewer's comments. After these modifications, the
model performance was improved as now it was able to reproduce the observed isotope
changes in surface snow at Dome C, in addition to the good agreements between the
modeled and observed isotope changes in vapor water. However, although the absolute
values of the modelled results are changed, we note the patterns of the results stay the
same, so as the conclusion.
The other main comment was on the designs of the simulations, i.e., how long the
simulations should be performed, and what types of input data (i.e., stacked means or
daily data) should be used. In the revised manuscript, for Dome C simulations we used
daily meteorological data during the studied period as input. However, for Dome A
simulations, in order to obtain representative results for summer clear-sky, cloudy and
winter conditions, we still used the stacked means as input and focused on the diurnal
variations and changes.
We confirm that all authors have approved the revised manuscript and its submission
to The Cryosphere. Please address all correspondence to genglei@ustc.edu.cn. We look
forward to hearing from you at your earliest convenience.
Sincerely,
Lei Geng
Professor
School of Earth and Space Sciences
University of Science and Technology of China
Hefei, 230026, China

**Response to Reviewer #1's comments**
**General comments**
*This manuscript considers the exchange between water molecules between the firn and the atmosphere, and the impact it can induce on the change of isotopic composition in extremely low accumulation regions of Antarctica. Using the results from Dome C as an analogue for Dome A is a clever strategy that can yield promising results to how to explain the impact of surface processes on the future Dome A ice core. The study takes into account the variations of stability of the atmosphere with systematic calculations of the Richardson number and developed three case studies associated with two sets of summer conditions (clear sky and cloud), and one set of winter conditions.*

*While the authors used a rather classical set of equations to evaluate the isotopic exchanges during sublimation and condensation, it seems not pertinent here, as it ignores major contributors to the boundary layer processes and only consider the system as a closed box without exchange with the free atmosphere. As a result, the results do not match the observations that were made for the surface snow isotopic composition at Dome C, even though, it is supposed to be the case study used to parametrise the model.*

*I suggest profound modifications to the model, which take into account exchanges between the atmospheric boundary layer and the free atmosphere on top of the surface processes, and which would match the surface snow changes, at least in order of magnitude, before considering the manuscript for publication.*

**Response:** We greatly appreciate the reviewer's insightful comment on the physical mechanism of our model. We agree that realistically exchanges between the atmospheric boundary layer and the free troposphere on top of the surface processes should be considered. It was our originally plan that we wanted to explicitly focused on how much changes on snow isotopes can be induced by processes at the air-snow interface alone. This may not reflect the real changes but can reveal the most potential effects associated with the processes at the air-snow interface. Thanks to the reviewer's suggestion, that we realized that it might be better to include the free troposphere which will make the result more comparable with the observations. Therefore, in the revised manuscript, we included the mass exchange between the boundary layer and the free troposphere by adding a third box as illustrated in the revised Figure 1. The calculations and equations were also changed to reflect the modifications to the physical mechanisms in the model. But we wanted to note that, with including the effects of exchanges between the boundary layer and free troposphere, the main conclusion of the manuscript doesn't change (the magnitude of modeled changes are affected but still in the same direction).

[Figure]

**Figure 1:** Schematic diagram of the box model used in this study (Revised version).

**Major Comments:**

**1)** *The box model developed by the authors was parametrised against vapour measurements obtained at Dome C, in order to compensate for lack of measurements at Dome A. The outputs of the model predict changes of vapour isotopic composition that seem realistic, but it is not the case for the changes of snow isotopic composition for which the variations are extremely small (less than 0.02‰) while the observed changes are around 2‰ during a typical night (Casado et al., 2018). The relative changes of snow and vapour isotopic compositions during a typical clear sky night were modelled in this manuscript, and suggested that a closed box model (which is de facto what the authors have implemented since no exchanges between the free atmosphere and the boundary layer are taken into account) is not realistic for this type of event.*

**Response:** Thanks for pointing out this. Indeed in the original model framework, the modeled results on snow isotopic changes cannot match the observations. In the revised manuscript, this is addressed by including the effect of exchanges between boundary and free troposphere. In particular, the new results indicate that simulated changes in snow isotopic composition are significantly larger than the original model (i.e., $\sim 0.02$‰ for $\delta^{18}O$) at Dome C. Especially, for the case the reviewer mentioned, i.e., a typical night with a frost event on Jan 6-7th, 2015, the diurnal changes of newly simulated results between the maximum and minimum can reach 2‰ for snow $\delta^{18}O$ (as demonstrated in Figure 2). This magnitude is in line with the observations for snow isotopes from Casado et al. (2018) which is $\sim 2$‰.

In the revised manuscript, we have re-run all simulations under Dome C conditions and three different cases at Dome A using the adjusted model. The simulated results within a 24-hour period were displayed in Figures 3-6 of the revised manuscript.

[Figure]

**Figure 2:** The changes of snow isotopes and water vapor isotopic composition (relative to the average value within a 24-h simulation) in the boundary layer in Jan 6-7[th], 2015 at Dome C.

**2)** *Another aspect that suggests that exchanges between the boundary layer and the free atmosphere must happen is the Richardson number. Indeed, for negative Richardson numbers, the atmosphere must be quite convective, which suggest that the boundary layer exchanges with both the surface snow and the free atmosphere.*
*The atmosphere is qualified as stable for any positive Richardson number, yet, it seems that some studies suggest that some amount of mixing remains quite strong for $0 < Ri < 0.1$ (Zilitinkevich et al., 2007) . This could be discussed.*

**Response:** Thanks for this valuable suggestion. In our original simulations, we assumed that unstable conditions for atmosphere stability only existed under negative Richardson numbers. Based on this assumption, we considered how mixing between the boundary layer, surface snow, and the free troposphere can affect the water vapor isotopic composition in the near-surface atmospheric layer and snow isotopes during the warming phase with negative Richardson numbers. However, as pointed by the reviewer that Zilitinkevich et al. (2008) suggested mixing can occur under positive Richardson numbers as well. If this is true, our original simulations for the water vapor isotopic composition in the near-surface atmospheric layer may be underestimated in the cooling phase.

To test the relationship between mixing occurrence conditions and Richardson
numbers, we ran simulations for Dome C taking into account mixing when Ri<0 and
Ri<0.1. As shown in Figure 3, the case with Ri<0 (Case II) indeed underestimates the
water vapor isotopic composition in the near-surface atmospheric layer during the
cooling time. Based on this comparison, in the revised manuscript, we incorporated
mixing into the modeling once Ri<0.1 (Case I) in addition to the original consideration
with Ri<0. Discussion on taking into account Ri<0.1 was added in supplementary
information (Texts S4) of the revised manuscript, and in the main text all results were
updated with consideration of the mixing when Ri<0.1.

[Figure]

**Figure 3:** The comparison of water vapor isotopic composition between the simulated
and observed changes at Dome C. Two simulated cases are presented here to discuss
the occurrence condition of mixing. In case I, the mixing is assumed to only happen
when Ri<0 in the cooling phase, while case II also considers the occurrence of mixing
when Ri<0.1 in the cooling phase.

**3)** *Some limited vapour data exist at Dome A (Liu et al., 2022). While these data might*
*be difficult to compare to your results, in particular consider how high the d-excess is,*
*which could be associated with calibration issues, it should be discussed.*
**Response:** Thanks for this suggestions. Actually before finalizing the manuscript, we
have discussed with the leading author of the Liu et al. (2022) study, but we noted that
due to the harsh environment, direct observations of water vapor as the Liu et al did is
difficult and the calibration can induce large issues. In addition, their measured sites are
actually not exact the same at Dome A (~100 km away). In the end we didn't choose to
compare this dataset. But since the reviewer asked, in the revised manuscript, we
compared our simulations at Dome A with the data of water vapor $\delta^{18}O$, $\delta D$, and d-
excess from Liu et al. (2022). We found that both our simulations and observations
exhibit diurnal patterns, with high values occurring during the warming phase (daytime)

and low values during the cooling phase (nighttime). However, we note that the magnitude of the observed diurnal changes in water vapor $\delta^{18}O$ and d-excess at sites near Dome A are very large, over 40‰ and 200‰, respectively. This could be due to calibration drift caused by the extremely cold and dry conditions during the measurements at the nearest Dome A site.

Therefore, in Section 4 of the revised manuscript, we only qualitative compare and discuss the similarities and/or differences between simulations and observations, without delving into quantitative details.

**4)** *Considering how fundamental these changes are, an updated version of the manuscript could have completely different conclusions.*

**Response:** We really appreciate the reviewer's comments. By including the effect of exchanges between boundary and free troposphere, the modeled results indeed differ a lot compared to the original model. However, the modeled changes in snow and vapor isotopes are still in the same direction (the magnitude or absolute values differ), and the main conclusion stays the same as that the air-snow exchange would lead to diurnal variations in atmospheric water vapor $\delta^{18}O$ and $\delta D$ by 4.75±2.15 ‰ and 28.79±19.06 ‰ under summer clear-sky conditions at Dome A, with corresponding diurnal variations in surface snow $\delta^{18}O$ and $\delta D$ by 0.81±0.24 ‰ and 1.64±2.71 ‰, respectively. These values become smaller compared to those in the previous simulations. After 24-hour simulation, snow water isotopes were enriched under clear-sky conditions. However, there is no or very little enrichment for snow water isotopes under cloudy conditions, which is different with the previous simulations. Under winter conditions at Dome A, the model still indicates the diurnal change in atmospheric and surface snow water isotopes are not significant, but the model predicts more or less depletions in snow $\delta^{18}O$ and $\delta D$ in the period of 24-hour simulation, opposite to the results under summer clear-sky conditions. This suggests that the air-snow vapor exchange tends to enlarge snow water isotope seasonality.

**End of the responses to Reviewer #1**

**Reference**

Casado, M., Landais, A., Picard, G., Münch, T., Laepple, T., Stenni, B., et al.: Archival processes of the water stable isotope signal in East Antarctic ice cores, The Cryosphere, 12(5), 1745-1766, doi: 10.5194/tc-12-1745-2018, 2018.

Liu J., Du Z., Zhang D., Wang S.: Diagnoses of Antarctic inland water cycle regime: Perspectives from atmospheric water vapor isotope observations along the transect from Zhongshan Station to Dome A, Frontiers in Earth Science, 10, doi: 10.3389/feart.2022.823515, 2022.

Zilitinkevich, S.S., Esau, I.N.: Similarity theory and calculation of turbulent fluxes at the surface for the stably stratified atmospheric boundary layer, Boundary-Layer Meteorology, 125, 193–205, doi: 10.1007/s10546-007-9187-4, 2007.

**Response to Reviewer #2's comments**
**General comments**
**1)** *My main concern is that the way the authors word their conclusions and their title suggests they provided model estimates of the diurnal variations in the snow and vapor isotopes. In fact, presented simulations are driven by average diurnal cycles of the meteorological parameters. Thus, instead, the authors provide the impact of an average day on initialized snow and vapor isotopes. The presented current results show how a given initial surface snow and vapor isotopic composition could develop within the first 24 hours when applying water vapor exchange.*

*It is unclear to me why the authors didn't run the simulation based on the meteorological input of individual days instead of stacking and averaging the input data. This limits the simulation time to only 24 hours. Such a short time does not allow for the development of the snow surface over several days. I would consider a minimum of a week spinup time to perform a model simulation in a more equilibrated state as could be expected in nature.*

*The intuitive approach to obtain an estimate of the average diurnal impact on the isotopes would be to run a longer simulation over several days and give the average daily impact. It seems to me that the authors have the needed data and tools to provide a model simulation over several days, as suggested above. This will improve the manuscript's relevance and provide better applicability of their results to explain observed changes in the snow isotopic composition.*

**Response:** We appreciate the reviewer's insightful comments. In the original manuscript, we chose to use the mean stacked conditions to conduct simulation since we wanted to highlight the effects of air-snow exchange in a general case. But in order to avoid confusion, in the revised manuscript, the simulations were conducted using continuous meteorological input for each individual day during the studied period at Dome C, where the model was run during the entire studied period (Jan 5th to Jan 16th, 2015), and the simulated results were stacked and averaged to evaluate the changes in snow and water vapor isotopes within a 24-hour period, as shown in Figure 3 of the revised manuscript. The model performance in water vapor isotopic variations is better than the simulations in the original manuscript. For snow isotopic composition, the diurnal evolution of simulated results can basically match with observations in the order of magnitude during a typical frost event (Figure 2 in this response).

In the Dome A simulations, however, the selected days for clear-sky, cloudy, and winter conditions were not continuous, making it difficult to conduct simulations as was done for Dome C. Instead, we were only able to use the model for one day to simulate the diurnal changes in snow and water vapor isotopes, after a week of spin-up time. This allows to evaluate the effects of air-snow vapor exchange under representative meteorological conditions. It is important to note that the input meteorological conditions and latent heat flux during both the spin-up time and the simulated period at Dome A were obtained from stacking observations or calculations on selected days, due to the non-continuous clear-sky and cloudy days in the studied period. Furthermore, the choice of the modeling running day and duration can significantly influence the final results of snow and water vapor isotopic composition, as meteorological conditions and latent heat flux vary significantly between two
different days within a season. To mitigate this effect, it is recommended to use the
averaged meteorological conditions to run simulations at Dome A. These approaches at
least provide some, on average, quantitative information on the isotopic effects of
atmospheric-snow water vapor exchanges at Dome A.

[Figure]

**Figure 1**. Schematic diagram of the box model used in this study (Revised version).

**2)** *Secondly, there are errors in the calculation of the latent heat flux as well as the*
*calculation of the isotopic flux. Please see the details below. In addition, to my*
*understanding, the latent heat flux is calculated based on already stacked and averaged*
*meteorological data. Since the latent heat flux is non-linearly dependent on these*
*meteorological parameters, the resulting flux based on the averages can diverge*
*severely from a diurnal average of the latent heat flux resulting from hourly calculations.*
*The presented simulations need to be re-run using the corrected latent heat flux*
*calculation.*

**Response:** We would like to express our gratitude to the reviewer for bringing to our
attention the errors in the calculations of latent heat flux and isotope flux. We have taken
into account the detailed comments provided in this response and have made the
necessary corrections to the equations for these parameters in the revised manuscript.

As part of our revisions, we have also changed the calculation method for the latent
heat flux and isotope flux for Dome C. Instead of using stacked and averaged
meteorological data within 24 hours, we now use continuous meteorological input for
individual days over the studied period. For the Dome A simulations, the latent heat
flux calculations remain the same as the Dome C simulation cases. However, the
isotope flux was obtained by stacked and averaged latent heat flux data due to the
selection of cloud conditions (Comment #1). These changes in the calculation method
can provide more accurate changes in the flux parameters on a diurnal scale.
Furthermore, the uncertainties of these parameters can be easily estimated by
calculating the standard deviation of the simulated results on the given days. More
details on this can be found in Comment #52 of this response.

**3)** *Another concern is that even when the above-mentioned errors in the latent heat flux calculation are corrected, the conditions for the Monin-Obukov similarity theory (MOST) are often violated under polar conditions. The present study does not discuss the quality of the calculated latent heat flux. If the authors pursue the goal of providing as realistic estimates of the water vapor exchange on the isotopes as possible, they have to make sure that the quality of the driving parameter, the latent heat flux, is well evaluated for similar conditions.*

**Response:** Thanks the reviewer for this comments. Indeed, the eddy covariance (EC) technique is a more robust method for quantifying latent heat fluxes and calculating isotopic fluxes at the atmosphere-snow interface, as demonstrated by Whal et al. (2021). However, this technique heavily relies on specialized measurement instruments, making it difficult to determine the latent heat flux in the absence of such instruments. As a result, high-quality latent heat flux data is not available at most polar sites.

Alternatively, the Monin-Obukhov similarity theory (MOST) are widely applied in polar regions because it calculates the latent heat flux based solely on meteorological parameters. While it seems not to be very suitable under polar conditions especially in winter, some previous studies have used the bulk method and MOST to calculate surface fluxes and the results were reasonable. For example, the King and Anderson (1994) study indicated that MOST can well describe the winter heat and water vapor fluxes at the Halley station of the Brunt Ice Shelf. Van den Broeke et al. (2005) calculated the year-round turbulent fluxes with MOST along a traverse line from coastal to inland region in Dronning Maud Land, Antarctica. Based on these, we think it is acceptable to use MOST and the bulk method if we intend to predict the potential mass and isotope changes that can be caused by atmosphere-snow vapor exchange.

When it comes to the quality of model calculations, the key factor is whether the model has been built using appropriate physical processes and meteorological parameters. If such a model can accurately reproduce observations at Dome C, it is highly likely that it will also be able to make predictions for Dome A within some degree of uncertainty. We hope we can have more observational data from Dome A to constrain the model, which is on progress but not available currently.

**Detailed comments**

**1)** *L20-22: This is misleading because the given values refer to the simulated changes when applying one average summer day. The way it is currently written suggests that the given values correspond to the average daily impact on the isotopes when simulating many different summer days.*

**Response:** Thank you for bringing the misleading information to our attention. We have revised the manuscript by re-simulating the continuous variations for snow isotopes and water vapor isotopes at the atmosphere-snow interface. Using the new simulated results obtained from Dome C and Dome A, we have calculated the daily impact of atmosphere-snow water vapor exchange on water isotopes. This was done by averaging the hourly values during summer clear-sky, cloudy and winter days. Based on these new results, we have rewritten the Abstract to reflect our findings accurately.

**2)** *L26: I disagree with this statement. Although, in contrast to summer, the meteorological variables don't seem to have a diurnal cycle in winter, the simulation of the isotopic changes shows similar magnitudes to the simulated changes in summer. How do you come to the conclusion that there are no relevant isotopic changes simulated on a diurnal scale in winter? Please clarify what this statement refers to. In that context, please reconsider the use of the term "diurnal cycle" or "diurnal pattern" in the manuscript. For me, a diurnal cycle is a repetitive pattern, i.e., similar values are found at the same time of the day. However, the authors use that term when describing the simulated isotopic change within 24 hours (e.g., L26, L295, L296, L310, L314, L319, L328, L330, L335, L339, L353-355, L361, L403,. . . ). But since the simulated isotopic values are different at 00:00 and 24:00 of the simulated day, the isotopes do not show a diurnal cycle but a change during one day.*

**Response:** Thanks for pointing out this. Our simulations at Dome A indicate that the water vapor isotopic composition during winter exhibits similar magnitudes of change to those observed during summer. However, the variations in snow isotopic composition during winter are significantly smaller than those observed during summer. This difference can be attributed to the more pronounced changes in meteorological conditions and latent heat flux that occur within a 24-hour period during summer days. As a result, we have revised the Abstract to emphasize the significance of meteorological conditions on the impact of atmosphere-snow water vapor exchange. Additionally, we have rephrased the sentences in L26 to provide a more explicit statement in the revised manuscript.

"Under winter conditions at Dome A, the model predicts that more or less depletions in snow $\delta^{18}O$ and $\delta D$ can be caused by atmosphere-snow water vapor exchange in the period of 24-hour simulation, opposite to the results under summer conditions.."

We also appreciate the feedback regarding the misnomer and have thus replaced the term "diurnal cycle" or "diurnal pattern" with the more accurate term "diurnal changes" or "diurnal variations" in the revised manuscript.

**3)** *L114-116: This sentence lacks clarity, please reformulate it. The calculation of sublimation and deposition is based on the same formula in the model, so why are two different formulations used here? And please change "followed by a mixing procedure and then uptake of surface snow", e.g., to "and the deposit is mixed into the snow surface layer".*

**Response:** Thank you for your comment. Previous studies have shown that there are differences in isotopic fractionation between sublimation and deposition (Ritter et al., 2016; Hughes et al., 2021). It is important to note that during deposition, the dominant process is equilibrium fractionation, whereas sublimation is significantly influenced by kinetic fractionation, except for equilibrium fractionation. Therefore, it is necessary to use two different formulations to describe the isotopic balance between snow and water vapor in Section 2.2. In case of mass changes in sublimation and deposition, the same formula as shown in Eq: (1) can be used.

However, we agree that the statement mentioned in the comment was confusing, and we have rewritten it in the revised manuscript as follows:

"During sublimation, water vapor is released from snow, transported into the atmospheric layer via turbulent mixing and molecular diffusion, and immediately mixed with the water vapor already in the boundary layer. During deposition, water vapor is influenced by aerodynamic resistance from turbulence and molecular diffusion, and the deposit is mixed with the surface snow layer."

**4)** *L124: What does "mainly" and "etc" refer to? Are further input parameters required*

*to run the model? If so, please provide a complete list of all input parameters. If not,*

*please remove the "etc".*

**Response #4:** Remove, Thanks.

**5)** *L129-130: Please provide a sufficient discussion of the uncertainty of the calculated*

*latent heat fluxes beyond what is presented in S2 in the supplements. Is there a way to*

*evaluate the quality of the latent heat flux calculations using another dataset (e.g.,*

*measured with an eddy covariance system)?*

**Response:** Thanks for your comment. We have made significant updates to the revised manuscript, particularly regarding the estimation method for the uncertainty of the latent heat flux calculations. The original Monte Carlo method has been replaced with a more straightforward approach that involves stacked and averaged simulations over multiple days. This new method relies on continuous calculations for the latent heat flux using meteorological input data from individual days. We have provided a detailed explanation of this new method in the Texts S2 of the supplements (details can be seen in Comment #52), where we also analyze the impact of the uncertainty of the calculated latent heat fluxes.

It is crucial to assess the accuracy of the latent heat flux calculations. However, there were no available measurements from the eddy covariance system to validate the calculations at Dome A. Therefore, we had to rely on comparing our calculations with those in previous publications. Ma Y. et al. (2011) had previously estimated the latent heat flux at this site. According to their findings, the latent heat flux calculations exhibited significant cycles on the diurnal scale and its diurnal ranges are 2.7 W/M$^2$

during summertime. These features and the order of magnitude for latent heat flux are consistent with the calculations in our study. Moreover, both the previous studies and our study found that the diurnal changes in latent heat flux are not significant during winter days. Based on these similarities, we are confident that the latent heat flux calculations in our study are reliable.

**6)** *L134, Eq 1.: The formula that the authors use to calculate the latent heat flux is not*

*correct. Following Berkowicz and Prahm (1982) (B&P82) from solving Eq. 22 for LE,*

*then using H from Eq. 11d with u and Θ∗ from Eqs. 11a and 11b, Δu = u$_{air}$ − u$_{surface}$*

*with u$_{surface}$ = 0, and γ =cp/Ls you obtain:*

$$LE = \rho L_s \kappa^2 \cdot \frac{u_{air}}{log\left(\frac{z_{u,a}}{z_{u,0}}\right) - \Psi_m\left(\frac{z_{u,2}}{L}\right) + \Psi_m\left(\frac{z_{u,1}}{L}\right)} \cdot \frac{q_a - q_s}{R \cdot log\left(\frac{z_{t,a}}{z_{t,0}}\right) - \Psi_h\left(\frac{z_{t,2}}{L}\right) + \Psi_h\left(\frac{z_{t,1}}{L}\right)} \qquad (1)$$

*Additionally, Ls should not show up on the right side of the formula when giving the*
*expression for LE/Ls. Please correct the theory of the box model calculation and re-run*
*all simulations of the study. Furthermore, in Eq. 1, in L134 and L138: There is no time*
*derivative given in B&P82, they use Δ to indicate the vertical gradient. When using the*
*MOST, the latent heat flux depends on the wind speed as well as the vertical humidity*
*gradient (qa-qs).*
**Response:** We are grateful to the reviewer for this valuable suggestion. Based on this
feedback, we have made necessary corrections to Eq: (1) in the revised manuscript.
However, for simplification of calculation, we ignored the corrected parameters in Eq:
(1) during modeling. Using the revised model, we generated new simulations and the
updated results are presented in Figures 2-6 of the main text (at the end of this response).

**7)** *L135: Please change "$\rho_V$" to "$\rho_a$".*
**Response:** Thanks, correct.

**8)** *L145: Where does the chosen value of 0.244 mm for the roughness length come from?*
*The latent heat flux is highly sensitive to the choice of the roughness length. Please*
*provide a sensitivity analysis of the simulated results to the choice of a range of*
*roughness lengths, e.g., 0.1 mm to 2 mm.*
**Response:** The roughness length ($z_0$) at Dome A was calculated in this study using the
least square method and wind observations at three levels (1 m, 2 m, and 4 m) under
neutral conditions, which typically vary between $10^{-5}$ to $10^{-3}$ m. To simplify the
calculations, a constant value of $z_0 = 2.44 \times 10^{-4}$ m was used in the modeling. This
estimate was determined using all wind speed data (397 groups) under neutral
conditions. It is worth noting that $z_0$ in this study is close to the previous calculation of
$1.45 \times 10^{-4}$ m from Ma et al., (2011).
We acknowledge the importance of $z_0$ value in obtaining accurate results. In
response to the reviewer's suggestion, we have added a sensitivity test for $z_0$ in the
supplementary section (Texts S5). Additionally, we have provided detailed explanations
and cautions for $z_0$ calculations in the supplementary.
The added texts S5 are shown as follows:
"Besides the initial parameters, changes in $z_0$ might influence the isotopic effects
of atmosphere-snow water vapor exchange. Thus, we also conducted the sensitivity test
for $z_0$ and run for a 24-h period under summer clear-sky conditions at Dome A. The test
was focused on the sensitivity of surface snow and water vapor $\delta^{18}O$ to varying $z_0$
between 0.01 to 10 mm. All other simulation settings were the same as in Section 2.2.4
of the main text.
The results of sensitivity tests for $z_0$ are shown in Fig. S4. As shown in the figure,
the magnitude of the diurnal variations in water vapor $\delta^{18}O$ ($\delta^{18}O_v$) is very sensitive to
$z_0$ (Fig. S4a) because $z_0$ determines the latent heat flux. This is consistent with Ritter et
al. (2016) who pointed out that diurnal variations in water vapor isotopic composition
decrease with the increase of boundary layer height. The magnitude of diurnal
variations in snow $\delta^{18}O$ ($\delta^{18}O_s$) is also sensitive to $z_0$ (Fig. S4b and S4c). However, the
changes in $\delta^{18}O_s$ is smaller than $\delta^{18}O_v$."

**9)** *L172: Above (in L138), RH$_i$ is defined as the relative humidity over ice, not for the specific humidity.*
**Response:** Thanks, correct.

**10)** *L182, L183: The "h" in Merlivat and Jouzel (1979) (M&J79) does not refer to the relative humidity of the air, but to the relative humidity of the air with respect to the surface temperature, i.e., h =qair qsat,surface (instead of RHair =qair qsat,air). The formulation in M&J79 is really confusing, but their qs in the formula of h =q/qs (below Eq. 9 in M&J79), in fact, refers to the "saturated specific humidity at the air-water interface (z=0)", i.e., the saturation specific humidity with respect to the surface temperature, while q is the air specific humidity. It is, thus, not correct to use the relative humidity here, but instead h =qair qs,surface. If this was not the case in the simulations, please correct and re-run them. Otherwise, please be more precise in the description of RH$_i$.*
**Response:** Thanks for the valuable feedback provided by the reviewer regarding the term 'humidity'. We have carefully reviewed our equations and made the necessary corrections based on the definition provided in Merlivat and Jouzel (1979). The revised equations have been used to generate new simulated results. Furthermore, we have improved the clarity of the description of RH$_i$ in the supplementary material. For more information on the corrections made, please kindly refer to our response to Comment # 51.

**11)** *L450 and 454. The authors state that the air temperature is controlling the isotopic fraction. This is not correct. It is the snow surface temperature, which is governing the isotopic fractionation. L189: Where does the expression for Rt EX come from? Because Eq. 2 in Jouzel and Merlivat (1984) is RtEX =af (Rtv + 1) − 1. Please correct this*
**Response:** Thanks for pointing out these mistakes. The necessary corrections have been done in the revised manuscript, including revising the Eq: (13) and updating L450 and L454.

$$R_{Ex}^t = \alpha_f (R_v^t + 1) - 1 \qquad (13)$$

**12)** *L200-201: Casado et al. (2016) does not present a snow dataset. If the authors refer to the Touzeau et al. (2016) dataset, please add the reference.*
**Response:** Thanks, we have added the reference.

**13)** *L209: I suggest replacing "representative" with "average". It was initially unclear to me what the authors meant by "stacking" the observed cycles.*
**Response:** We agree. The "representative" has been replaced by "average" in the revised manuscript.

**14)** *L210: Please remove the "e.g." and "etc." in the parenthesis since the given parameters are the only ones that can be downloaded from the CALVA program.*
**Response:** Thanks, delete.

**15)** *L211-216: Is there no surface temperature record available for DOME-C? And if not, why is the surface temperature calculated from ERA-5 model long wave data instead of using the ERA-5 model output of the surface temperature?*

**Response:** During the modelled period, surface temperature data were available for Dome C, as measured by a Campbell Scientific IR120 infrared probe and reported by Casado et al. (2016). In the revised manuscript, we used these observations as input for simulations at Dome C instead of the calculations based on the method from Brun et al. (2011).

However, for Dome A, surface temperature observations were not available from 2005 to 2011. Therefore, we used the method from Brun et al. (2011) to calculate surface temperature (Eq: (17) in the main text). We chose this method because it can accurately represent the observations at Dome C. To validate the calculations at Dome A, we compared them with observed 10cm firn temperature at the same location. The calculations matched well with the observed snow temperature for the top 10cm layer, as shown in Figure 2a.

$$Ts = \left(\frac{LW_{up} + (\epsilon-1)LW_{dn}}{\epsilon\sigma}\right)^{0.25} \tag{17}$$

Furthermore, the direct output of surface temperature from the ERA-5 model can also be used as input for our model because the ERA-5 model output at Dome C is comparable to the surface temperature calculations based on the method used in this study, as well as the long-wave radiation data from the ERA-5 reanalysis data (Figure 2b).

[Figure]

**Figure 2.** The comparison of the $T_s$ results of different methods. (a) The calculated $T_s$ and the observed snow temperature for top 10 cm snow at Dome A, during the period of 2005-2011 (b) The calculated $T_s$, the ERA-5 model output of $T_s$ and the observed $T_s$ at Dome C, during the period of 5th-16th January, 2015

**16)** *L214: An emissivity of 0.93 seems relatively low to me. Please indicate where this value originates from.*

**Response:** Thanks for this comment. The value of 0.93 for snow emissivity was cited from the Doctoral thesis of Ma et al. (2012), which calculated the surface snow temperature at Dome A. This value is lower than the snow emissivity of 0.99 at Dome

C (Brun et al., 2011; Vignon et al., 2017). Despite the significant difference between these two values, we still use the value of 0.93 as the snow emissivity for Dome A simulations. We have now included this difference between Dome A and Dome C in the revised Table S1.

**17)** *L216-217: The latent heat flux is calculated based on the averaged meteorological parameters. In my view, it makes more sense to calculate the latent heat flux based on the hourly data and (if needed) stack and average it afterward.*
**Response:** We concur that the fluctuations in latent heat flux over a period of multiple days are significant for subsequent simulations related to water isotopes. To that end, we recalculated the latent heat flux and then computed the average, which is illustrated in Figure 2 of the primary text (please see the revised version at the end of this response).

**18)** *L217: Please remove the "etc." if no further data is used.*
**Response**: Thanks, remove.

**19)** *L220: An average snow density from 2m+ deep snow pits might not be appropriate for the top 1.5 cm. Please provide a sensitivity analysis of the simulation using a range of realistic surface snow densities.*
**Response**: Thanks for the suggestion. We will test the isotopic values in response to varying snow density at Dome A and add results to the Section 2.2.4 and Section 3.4 of the main text.

**20)** *L234: What does "to fully assess the accumulated isotope effects of atmosphere-snow water vapor exchange." mean? Please rewrite this sentence to clarify on this.*
**Response**: In order to illustrate the impact of cloud presence on the simulation results at Dome A, we have conducted two simulated cases: one with cloud and one without cloud. However, we understand that the original sentence in L234 may have been unclear. Therefore, we have completely rewritten the sentence as follows:
"Therefore, in the model simulations for Dome A, we simulated two representative cases with and without cloud (i.e., cloudy vs. clear-sky conditions) in order to accurately assess the isotopic variations associated with atmosphere-snow water vapor exchange."

**21)** *L250-251: I hardly see any diurnal cycle in the wind speed. In addition, I would argue that the diurnal cycle of the LE differs from the diurnal cycles of Ts and q, since it has a local minimum at 07:00UTC.*
**Response**: Thanks for providing a different perspective, as suggested by the reviewer. The wind has a diurnal cycle under clear-sky conditions at Dome A. However, due to the large range of the y-axis in Figure 2a of main text, the significant pattern for wind was unclear. We have made necessary corrections to Figure 2 of main text to improve its clarity.
    Regarding LE, we recalculated it following the reviewer's suggestion. The results show that high LE values are observed during the warming phase, and lower values during the cooling phase, similar to $T_s$ and q as depicted in Figure 2 of the main text (at the end of this response). We acknowledge that the original manuscript may have had unclear sentences or descriptions for LE changes. We have revised the manuscript by rewriting the sentences to make it more precise and clear in expressing our viewpoint.

**22)** *The argument that the use of Pang et al. 2019 is a reliable approach is a circular argument since you are using the estimate of Pang et al. 2019 to compare with the data that Pang et al uses to create the relationship between isotope and temperature.*

**Response**: Thanks for this comment. To support our estimate, we used simulation data from ECHAM5-wiso (Werner et al., 2011), which calculated precipitation isotopes based on temperature and other factors. We compared the results of our calculation with the simulation data, and the comparison is presented in Figure 2 of the main text. As shown in the figure, the two methods agree with each other quite well.

[Figure]

**Figure 3**. The estimated precipitation $\delta^{18}O$ and its standard deviation during the period of 2005-2011. Blue solid line with star marks represents the calculations using the temperature-isotope slope, and the light blue shaded area is the uncertainties. Black solid line with x marks and light grey shaded area displays the ECHAM5-wiso simulation data and its uncertainties, respectively.

**23)** *L251, L266: It is not correct to say that the meteorological data are less variable in winter. In fact, all meteorological variables are similarly variable as they have about the same standard deviation. Maybe reformulate to "none of the meteorological variables shows a diurnal cycle" or "in the winter data does not show a diurnal signal."*

**Response**: We appreciate your valuable suggestion. The sentences mentioned in the comment have been revised in the new version of the manuscript.

**24)** *L260: Please give the value of the used snow density. How does this value compare to the density taken from Laepple et al. (2018) for the DOME-C simulations?*

**Response:** In Table 1 of main text, we have listed the snow density values at Dome A and Dome C. The snow density value at Dome A (380 kg/m$^3$) is slightly higher than that at Dome C (329 kg/m$^3$).

**25)** *L265-266: How is winter defined? Are all hourly data from June-August used?*

**Response**: Yes, the winter period corresponds to June-August in Antarctica. During the winter period in Antarctica, hourly meteorological data from clear-sky days were retrieved and then averaged for running simulations at Dome A.

**26)** *L272: Please provide the value of the used δ-T slope in the text.*

**Response:** For non-summer seasons, the isotopes of precipitation were also estimated using the regression line (slope of 0.64±0.02, R$^2$=0.59) of the non-summer precipitation isotopic composition and near surface air temperature at Dome F, Vostok and Dome C

compiled by Pang et al. (2019). In the main text, we added the used δ-T slope following the comment.

**27)** *L273-274: Where is this comparison presented, and why is this relevant here? Did*

*this comparison influence the initial values of δ$^{18}$O$_s$? If not, I suggest to remove this.*

**Response:** We appreciate this suggestion. We used a comparison of δ$^{18}$O$_s$ values between the ECWMF-wiso dataset and linear calculations using the δ-T slope to validate the δ$^{18}$O$_s$ estimation. The results of this comparison are presented in Figure 3.

We observed a strong correlation between the monthly δ$^{18}$O$_s$ variations in these two data sources, and their values were similar in each month, indicating that the linear calculations are reliable. Based on this finding, we can confidently state in the main text that the setting of δ$^{18}$O$_s$ values are accurate at Dome A. Thus, it is necessary to mention the comparison between δ$^{18}$O$_s$ calculations from the δ-T slope and the ECWMF-wiso dataset in the text.

**28)** *L277: Please add the reference (Ma et al., 2020) behind "measurements" again*

**Response:** Thanks for reminding this. We have checked and added the reference.

**29**) *L292: Please clarify: What does the "disequilibrium was included" mean?*

**Response:** The term "disequilibrium" in the original manuscript refers to the isotopic composition of water vapor being in thermodynamic imbalance with the snow isotopes at the snow-atmosphere interface. During modeling, we assumed that the isotopic composition of water vapor was in equilibrium with the snow isotopes under the initial conditions. However, published observations from other polar sites indicate that

"disequilibrium" conditions are common. To test how "disequilibrium" conditions affect simulations of water vapor isotopic composition and snow isotopes, we designed sensitivity experiments. In the section 2.4 of main text, we used the phrase

"disequilibrium was included" to accurately describe the case. However, this description may not be clear to readers. In the revised manuscript, we replaced it with

"the isotopic composition of water vapor being in thermodynamic imbalance with the snow isotopes was included" to make it easier to understand.

**30**) *L300-301: The authors mention snow samples for Dome-C in L200-201. An*

*evaluation of the snow isotopic composition development to observations would be very*
*beneficial for the analysis. The simulated changes in snow isotopic composition seem*
*very small compared to variations observed in surface snow samples.*

**Response:** We acknowledge that the simulated changes in the isotopic composition of snow do not match well with the observations at Dome C. This error can be attributed to the absence of certain physical mechanisms in the original model. To address this issue, we utilized an updated model, which is mentioned in Figure 1, to re-run simulations during the Jan 5th -16th, 2015 at Dome C. As depicted in Figure 3 of the main text (see details at end of this response), the averaged magnitude of the simulated snow isotopic variations aligns with the stacked observations within 24 hours.

**31)** *L314-315: It is not correct to say diurnal cycle here, instead, Fig. 4 shows the simulated change isotopic composition within 24 hours when applying an average summer day observed in January 5-12th.*

**Response:** Thanks, we corrected the L314-315 following the reviewer's suggestion. The details are as follows:

"The modelled snow $\delta^{18}O$ and $\delta D$ follow a diurnal pattern where higher values occur during the warming phase and lower values during the cooling phase (Fig. 3d). The diurnal range of simulated snow $\delta^{18}O$ are ~2‰ on average. This value is close to the observations in the order of magnitude during a typical frost event, but smaller than that of the simulated water vapor $\delta^{18}O$."

**32)** *L319: What does "diurnal variations" mean? Diurnal maximum minus diurnal minimum? Please define. Maybe the term "diurnal range" is more suitable?*

**Response:** Thanks for this helpful suggestion. The "diurnal variations" in this sentence means the diurnal maximum minus diurnal minimum. To make it more clear, we used the "diurnal range" to replace the "diurnal variations".

**33)** *L339: As mentioned above, the changes in isotopic composition in winter are comparable to the ones in summer.*

**Response:** Thanks for the comment. We have revised this sentence as follows:

"As a result, in comparison with the simulated results in summer, there is no significant diurnal variations in snow isotopes in winter, but the changes in water vapor isotopic composition in winter are comparable to the ones in summer."

**34)** *L354-355: I cannot confirm this statement based on the figures. The different axis ranges make it difficult to compare.*

**Response:** Thanks for pointing out this. In the revised manuscript, we replotted the Figure 7 to clearly show the sensitivity of simulated results to changes in initial conditions.

**35)** *L359: Please discuss how the simulated results compare to other similar modeling studies, e.g., Wahl et al. (2022) (for Greenland) and Ritter et al. (2016)?*

**Response:** Thank you for your helpful comment. We have revised the manuscript to include a discussion of the similarities and differences between our calculations and the simulated results of other studies. One significant similarity we found with two similar studies you mentioned is that diurnal variations in snow isotopes and water vapor isotopic composition in the boundary layer can be mainly explained by the atmosphere-snow water vapor exchange through modeling results. Additionally, these studies suggest that the accumulation of isotopic effects from the atmosphere-snow water vapor exchange can lead to isotopic enrichment of the snow layer during the summer, if the snow layer remains consistently exposed at the surface. One main difference we noticed between these studies is the magnitude of diurnal changes in water vapor isotopic composition and snow isotopes. For instance, the diurnal range of snow isotopic composition at Dome C is larger than that at Kohnen station and Dome A, which can be attributed to the stronger variability of humidity gradient and wind speed at Dome C. We have added these comparisons and related discussions to the main text's Discussion section.

The detailed comparison in the main text is shown as follows:

"We also compared modelled water vapor $\delta^{18}O$, $\delta D$, and d-excess data at Dome A with those observations from other East Antarctic interior sites, such as Kohnen station, Dome C, and a location about 100 km away from Dome A (Ritter et al., 2016; Casado et al., 2016; Liu et al., 2022). In general, both our simulations and observations show diurnal patterns, with high values during the daytime warming phase and low values during the night-time cooling phase. However, we noticed that the observed diurnal changes in water vapor $\delta^{18}O$ and d-excess at sites near Dome A are very large, over 40‰ and 200‰, respectively. This is probably due to calibration drifts caused by the extremely cold and dry conditions during the measurements at the nearest Dome A site which influence the measurements (Liu et al., 2022). The averaged $\delta D$ observations of 36±6‰ at Kohnen station and the in-situ measurements of 38±2‰ at Dome C are higher than our modeled $\delta D$ value of 28.78±19.06‰ at Dome A. This difference can be attributed to atmospheric dynamical conditions linked with wind speed in addition to other meteorological conditions. At Dome A, the daily mean wind speed of 2.8 m/s is lower than 3.3 m/s in Dome C and 4.5 m/s in Kohnen station during summer. A lower wind speed corresponds to relatively weak air convection in the horizontal orientation. Due to the coupling between upper and lower atmospheric layers, vertical turbulent mixing may decrease with the weakened air convection in the atmospheric boundary layer (Casado et al., 2018). This change can attenuate molecular exchange between surface snow and water vapor. In parallel, the decrease of vertical turbulence may result in a less efficient turbulent diffusion of water molecules and an elevated contribution of molecular diffusion during atmosphere-snow water vapor exchange. Changes in water vapor diffusion pathways increase kinetic fractionation and reduce effective isotopic fractionation of water isotopes, leading to a muted fluctuation of modelled water vapor $\delta D$ in combination with less mass exchange."

**36)** *L356-358: This basically means that the simulated snow isotopic composition does not significantly change after 24 hours of simulation? How much does it change when letting the simulation run longer?*

**Response:** Thanks for this constructive suggestion. We have conducted simulations for
Dome A over the course of one week during summer, using the updated model. We
observed that the isotopic composition of snow became more enriched compared to its
initial state (Figure 4).

[Figure]

**Figure S4:** The simulated changes in snow and water vapor isotopes in an 11-day period
(Jan 5-16$^{th}$, 2015) under Dome C conditions

**37)** *L364-366: Please reformulate this sentence more clearly.*
**Response:** Thanks for this suggestion. We reformulated this sentence as following:
"In general, in the period of mass exchange dominated by sublimation, snow $\delta^{18}O$ and
$\delta D$ are enriched as lighter isotopes are preferentially sublimated to the atmosphere.
Meanwhile, sublimates mixing with vapor water lead to increases in vapor $\delta^{18}O$ and $\delta D$
because the sublimates are of higher $\delta^{18}O$ and $\delta D$ than atmospheric vapor.".

**38)** *L369-370: How is this evident? The authors do not provide evidence for what drives*
*the isotopic composition, neither within their 24-hour simulation nor in a more realistic*
*simulation of a longer time period. The latent heat flux is driven by (1) the near-surface*
*humidity gradient (which, of course, is closely related to the near-surface temperature*
*gradient) and (2) the wind speed. However, this study lacks any evidence that the*
*temperature and humidity drive the surface snow isotopic composition. Please remove*
*this statement.*
**Response:** Thanks for this suggestion. We acknowledge that original manuscript did
not accurately reflect the relationship between temperature, humidity, and water vapor
isotopic composition. After calculating the latent heat flux, we agree that the water
vapor and snow isotopic composition are likely controlled by the near-surface humidity
gradient and wind speed. We have revised this statement to reflect the discussion after
this sentence, rather than deleting it. The new statement is as follows:
"Based on Fig. 2, 4c, and 5c, it is clear that the diurnal isotope cycles in surface snow
and vapor water have a strong correlation with temperature and humidity."

**39)** *L371-372: The authors suggest that wind speed doesn't seem to affect the isotopic*
*composition of the surface snow. However, I'd like to point out that they're using an*

*average wind speed over 11 days, which doesn't show the hourly changes. Thus, such*
*simulation does not allow for a statement that wind speed does not drive the snow*
*isotopic composition at Dome-C. For example, let's say, just to make my point, that 90%*
*of the changes in snow type are due to wind speed. If the wind speed increases linearly*
*from 2 to 7 m/s over the first 5.5 days and then decreases from 7 to 2 m/s in the next 5.5*
*days, the snow isotopes would change mainly driven by the wind speed. However, the*
*daily average of this wind change would always be 4.5 m/s for all 24 hours. So, when*
*they use the daily average wind speed in their simulation, it makes it seem like wind*
*has no effect on the snow isotopic composition, even though in this example, wind was*
*defined to be the main factor driving the isotopic changes.*

**Response:** We completely agree with the reviewer's viewpoint. The original
simulations, which used averaged meteorological conditions over a 24-hour period,
failed to accurately reflect the impact of wind on the water vapor and snow isotopic
composition at the atmosphere-snow interface. To address this issue, we re-ran the
simulations to obtain continuous isotopic variations during the studied period.

Furthermore, we conducted a sensitivity test by varying with a significant diurnal
cycle of wind and comparing it with the ones with averaged wind speed. The results, as
shown in Figure 5 (i.e., Figure S2 of the supplementary information), suggest that
strong variability in wind speed will enlarge the variations in latent heat, leading to a
more significant diurnal change in water vapor isotopes and snow isotopes.

[Figure]

**Figure 5:** The comparison of water vapor isotopic composition between two simulated
cases at Dome A. The simulations in two cases were driven using the averaged wind
speed (Case I) and the strong diurnal changes in wind speed (Case II).

**40)** *L386: What does this mean: "This could adversely affect changes in atmospheric*
*dynamical conditions between day and night"? Please clarify*

**Response:** The statement in this comment suggests that smaller temperature changes
within a cloudy day can create relatively stable atmospheric dynamical conditions. As a result, the diurnal variations of latent heat flux in summer cloudy days are less significant than those in summer clear-sky days. This leads to less mass exchange as well as isotope effects during atmosphere-snow water vapor exchanges. To make the statement clearer, we have reformulated it as follows:

"With the presence of cloud, the differences between the air temperature and surface temperature during the day and night become less pronounced (as shown in Fig. 2). This could have a negative impact on the changes in atmospheric dynamics between day and night, as evidenced by the relatively small magnitude of diurnal variations in Richardson number (as shown in Figs. 4a and 5a)."

**41)** *L387-389: The authors cannot state that: There is no diurnal cycle when averaging, but of course, the wind speed varies on an hourly and daily basis, and the standard deviation is not zero.*

**Response:** Thanks for pointing out this inappropriate statement. After careful consideration, we have decided to remove it as this sentence does not contribute to the following discussion.

**42)** *L427-429: Again, the simulated change in the isotopic composition of the vapor is of a comparable magnitude as the changes in summer. What do the authors base this statement on?*

**Response:** It is unclear for the statement in the L427-429 of the original manuscript. We have revised it based on the response to Comment #33.

**"The results indicate there is small diurnal changes for snow isotopes over the 24-hour simulation period".**

**43)** *L444-446: The CALVA program states a sentence on its website on how to acknowledge them for the dataset correctly.*

**Response:** Thanks for reminding this. We will use the standard way to express the acknowledgement for the CALVA program in the revised manuscript.

"We also acknowledge using Dome C data from the CALVA project and CENECLAM and GLACIOCLIM observatories (http://www-lgge.ujf-grenoble.fr/~christo/calva/)."

**44)** *References: The two given references for Ma et al. (2020) can currently not be distinguished in the text.*

**Response:** Thanks for the comment. We would like to clarify that the two papers referenced are published by Ma Bin et al. (2020) and Ma Tianming et al. (2020), respectively. To avoid confusion, we have used the formulation "Ma B. et al. (2020)" and "Ma T. et al. (2020)" when citing these two studies in the text.

**45)** *Figure 2b: Why is the standard deviation of the latent heat flux so low for cloudy conditions?*

**Response:** Under cloudy conditions, the relatively low values in the standard deviation of the latent heat flux is mainly attributed to the calculated method (Monte-Carlo method). In the revised manuscript, we directly estimated the standard deviation by stacking the simulated diurnal variations of the latent heat flux at the given days. The
corrected results can be seen in the Figure 2b of the revised manuscript.
**46)** *Figure 3: What is σ for the simulations? Is it the calculated range from the Monte*
*Carlo simulations, or is it the standard deviation of the Monte Carlo simulations?*
**Response:** The σ in Figure 3 represents the standard deviation of the Monte Carlo
simulations. According to the reviewers, the estimates for uncertainty provided in the
original manuscript is inappropriate. In the revised manuscript, we have directly
estimated the standard deviation by stacking the simulated diurnal variations of snow
and water vapor isotopic composition in the individual days. The details can be seen in
the Text S2 of the supplemental information (response to Comment #52) and Figure 3
of the main text (at the end of this response).
**47)** *Figure 3 caption: Add water "vapor" isotopic composition.*
**Response:** Thanks, Correct.
**48)** *Figure 4: Again, please be more precise on what "uncertainty" means.*
**Response:** We have given a detailed explanation in the Comment #46. Please see the
response to that comment.
**49)** *Figure 7: Please provide an explanation of the red lines.*
**Response:** The red lines in Figure 7 represent the modeled magnitudes of $\delta^{18}O$ diurnal
variations in water vapor and snow with the changes in initial conditions. They in fact
show the same meanings as the color bar in each panel. Given that, we remove these
red lines in the revised manuscript.
**50)** *Figure 7 caption: Change "6c and 6d" to "7c and 7d".*
**Response:** Thanks, Correct.
**51)** *Supplement material S1: The description of the post-processing of the relative*
*humidity (RHw to RHi) is very difficult to understand. – L51-52: Why do you normalize*
*RHw? – L52: Which surface temperature is used? The calculated Ts based on ERA-5?*
*If so, please discuss the introduced error by normalizing the observations using model*
*data. – L54: (Eq. 15): Do you refer to Eq. 13? – L60: What is an "ideal maximum"? –*
*L60, L61: What do you mean by "each temperature point"? – L63-64: The description*
*of the factor is incomplete (the ratio of es with respect to water to es with respect to ice.*
*Moreover, why do you only apply this factor for super-saturated conditions? The*
*relative humidity should be corrected with respect to ice for sub-saturation as well. –*
*L64: What do you mean by "the rising amplitude of the temperature"?*
**Response:** We appreciate a lot for the reviewer#2's careful checking and valuable
comments for Supplement material S1. This part has been rewritten as follows:
"The raw data of relative humidity (RH) at height z is the relative humidity with
respect to the water surface ($RH_w$), measured with the HMP35D humidity probe (Xiao
et al., 2008; Ding et al., 2022). The $RH_w$ can be expressed as a percentage:

$$RH_w = e_w/e_w^s \times 100\% \qquad\qquad (S2)$$

where $e_w$ is the water vapor pressure of air (Pa), and $e_w^s$ is the saturated vapor pressure with respect to the water surface at the air temperature (Pa) which can be calculated using the Clausius-Clapeyron equation. When calculating the effective fractionation factor ($\alpha_f$) in the model (Eq: (15) in the main text), the $RH_w$ were converted to the relative humidity over ice at the temperature of the air ($RH_i$). The conversion between $RH_i$ and $RH_w$ was proposed based on the calibration procedures of Anderson et al. (1984). The details are as follows: 1) The $RH_w$ observations were firstly rescaled using the maximum $RH_w$ of all measured values at each air temperature point ($T_a$),

$$RH_w^{'} = RH_w(T_a)/RH_w^{max}(T_a) \qquad\qquad (S3)$$

2) $RH_w^{'}$ values were then converted to $RH_i$ using Eq: (S4) :

$$RH_i = (e_w^s(T_a)/e_i^s(T_a)) \times RH_w^{'} \qquad\qquad (S4)$$

where $e_i^s$ represents the saturated vapor pressure with respect to ice at the air temperature (Pa). Like $e_w^s$, $e_i^s$ was calculated by the Clausius-Clapeyron equation. Based on Eq: (S3) and Eq: (S4), we obtained $RH_i$ as the final result.

In addition, the relative humidity of the air with respect to the surface temperature (h) in Eq: (14) can also be converted from $RH_w$ observations. The first step of procedures for h conversion is the rescaling $RH_w$ based on Eq: (S3), same to the $RH_i$ conversion. The second step is h calculation using the saturated vapor pressure with respect to ice at the surface temperature (Eq: (S5)).

$$h = (e_w^s(T_a)/e_i^s(T_s)) \times RH_w^{'} \qquad\qquad (S5)"$$

**52)** *Supplement material S2: The description of the uncertainty estimate/error propagation is partly unclear and could be improved. Furthermore, the simulation uncertainties are not sufficiently mentioned and discussed in the main manuscript. A Figure in S2 that shows the calculated uncertainties for all variables could be helpful. – L70: How are the "uncertainties" calculated? Is it the standard deviation? – L72: Which are "those days"? – L75: Which error the standard deviation is applied? Please provide more details.*

**Response:** We would like to express our gratitude to the reviewer for reviewing the supplement material S2. The term "uncertainties" in our study represents the standard deviation of each variable. We have estimated them directly by stacking the observations and calculations on the given days in the revised manuscript. The corrections have thus been made in the supplementary document as we have updated our method of estimating uncertainties. The revised Text S2 is as follows:

"At each time step, we first calculated the standard deviation as the uncertainties ($1\sigma$) of wind speed, air temperature, relative humidity by stacking the hourly observations from AWS on the selected days for each parameter. The same method was then applied to determine the uncertainty of surface temperature using hourly
calculations from Brun et al., (2012). We also used the stacking method to estimate the
uncertainties of other calculations such as the latent heat flux ($Q_{LE}$). These estimated
uncertainties were plotted in Figures 2 of the main text (shaded areas).

The standard deviations of water vapor and surface snow $\delta^{18}O$, $\delta D$, and d-excess
serve as the uncertainties of simulated isotopic values ($Q_\delta$). In the Dome C simulations,
these values were calculated by stacking continuous simulations of water isotopes for
each day between January 5th and January 16th in 2015 (as indicated by the shaded
area in Figure 3). However, clear-sky and cloudy days selected for Dome A simulations
are not continuous. Therefore, we were only able to use the model for one day to
simulate the diurnal changes in snow and water vapor isotopes, after a week of spin-up
time. This make it difficult to estimate the uncertainties of water isotopes using the
simple stacking method. To determine the uncertainties, we used error propagation
method as an alternative solution, as referred to by Radic et al. (2017). First, we
calculated the uncertainties of the fractionation coefficient ($Q_\alpha$) based on the standard
deviation of surface temperature. Then, we used the uncertainties of latent heat ($Q_{LE}$)
and $Q_\alpha$ to determine $Q_\delta$. The equations used to calculate $Q_\alpha$ and $Q_\delta$ are shown as below:

$$Q_\alpha = \alpha' * Q_{Ts} \tag{S6}$$

$$Q_\delta = \sqrt{(\frac{\partial \delta}{\partial \alpha} * Q_\alpha)^2 + (\frac{\partial \delta}{\partial LE} * Q_{LE})^2} \tag{S7}$$

where $\alpha'$ is the derivative of fractionation coefficient (Eq:(13) of the main text), the

$\frac{\partial \delta}{\partial \alpha}$ and $\frac{\partial \delta}{\partial LE}$ represents the derivative of fractionation coefficient and latent heat flux in the equation of isotopic balance of the model (Eq: (10) of the main text). The final
results are shown in the Figures 4-6 of the main text."
**End of the responses to Reviewer #2**

**Reference**
Brun, E., Six, D., Picard, G., Vionnet, V., Arnaud, L., Bazile, E., et al.:
Snow/atmosphere coupled simulation at Dome C, Antarctica, *Journal of*
*Glaciology*, 57(204), 721-736, doi: 10.3189/002214311797409794, 2011.
Casado, M., Landais, A., Picard, G., Münch, T., Laepple, T., Stenni, B., et al.: Archival
processes of the water stable isotope signal in East Antarctic ice cores, *The*
*Cryosphere*, 12(5), 1745-1766, doi: 10.5194/tc-12-1745-2018, 2018.
Hughes, A. G., Wahl, S., Jones, T. R., Zuhr, A., Hörhold, M., White, J. W. C., et al: The
role of sublimation as a driver of climate signals in the water isotope content of
surface snow Laboratory and field experimental results, *The Cryosphere*, 15(10),
4949-4974, doi: 10.5194/tc-15-4949-2021, 2021.
King, J. C., & Anderson, P. S.: Heat and water vapour fluxes and scalar roughness
lengths over an Antarctic ice shelf, *Boundary-Layer Meteorology*, 69, 101–121, doi: org/10.1007/BF00713297, 1994.

Ma, B., Shang, Z., Hu, Y., Hu, K., Wang, Y., Yang, X., et al.: Night-time measurements
of astronomical seeing at Dome A in Antarctica, *Nature*, 583(7818), 771–774, doi:
10.1038/s41586-020-2489-0, 2020.

Ma, T., Li, L., Li, Y., An, C., Yu, J., Ma, H., et al.: Stable isotopic composition in
snowpack along the traverse from a coastal location to Dome A (East Antarctica):
Results from observations and numerical modelling, *Polar Science*, 24, 100510,
doi: 10.1016/j.polar.2020.100510, 2020.

Ma Y..: Evaluation of Polar WRF Simulations of Atmospheric Circulation. 2012.
Chinese Academy of Meteorological Sciences, PhD dissertation, 2012.

Merlivat, L., & Jouzel, J.: Global climatic interpretation of the deuterium-oxygen 18
relationship for precipitation, *Journal of Geophysical Research: Oceans*, 84(C8),
5029, doi: 10.1029/JC084iC08p05029, 1979.

Pang, H., Hou, S., Landais, A., Masson-Delmotte, V., Jouzel, J., Steen-Larsen, H. C., et
al.: Influence of Summer Sublimation on $\delta D$, $\delta 18O$, and $\delta 17O$ in Precipitation,
East Antarctica, and Implications for Climate Reconstruction from Ice Cores,
*Journal of Geophysical Research: Atmospheres*, 124(13), 7339-7358, doi:
10.1029/2018JD030218, 2019.

Ritter, F., Steen-Larsen, H. C., Werner, M., Masson-Delmotte, V., Orsi, A., Behrens, M.,
et al.: Isotopic exchange on the diurnal scale between near-surface snow and lower
atmospheric water vapor at Kohnen station, East Antarctica, *The Cryosphere*,
10(4), 1647-1663, doi: 10.5194/tc-10-1647-2016, 2016.

Touzeau, A., Landais, A., Stenni, B., Uemura, R., Fukui, K., Fujita, S., et al.:
Acquisition of isotopic composition for surface snow in East Antarctica and the
links to climatic parameters, *The Cryosphere*, 10(2), 837-852, doi:10.5194/tc-10-
837-2016, 2016.

van den Broeke, M., van As, D., Reijmer, C. & van de Wal, R.: Sensible heat exchange
at the Antarctic snow surface: a study with automatic weather stations.
I*nternational Journal of Climatology*, 25, 1081-11010-, doi:10.1002/joc.1152,
2005.

Vignon, E., Genthon, C., Barral, H., Amory, C., Picard, G., Gallée, H. et al.:
Momentum- and Heat-Flux Parametrization at Dome C, Antarctica: A Sensitivity
Study. *Boundary-Layer Meteorology,* 162, 341–367, doi: 10.1007/s10546-016-
0192-3, 2017.

Wahl, S., Steen-Larsen, H. C., Reuder, J., & Hörhold, M.: Quantifying the Stable Water
Isotopologue Exchange Between the Snow Surface and Lower Atmosphere by
Direct Flux Measurements, *Journal of Geophysical Research: Atmospheres,*
126(13), doi: 10.1029/2020jd034400, 2021.

Werner, M., Langebroek, P. M., Carlsen, T., Herold, M., & Lohmann, G.: Stable water
isotopes in the ECHAM5 general circulation model: Toward high-resolution
isotope modeling on a global scale, *Journal of Geophysical Research: Atmosphere*,
116, 14, doi: 10.1029/2011jd015681, 2011.

**Response to Reviewer #3's comments**
**General comments**
**1)** *This manuscript describes a closed box model assuming no atmospheric mixing and simulations of the effect of a mean diurnal cycle at Dome C (using observations) and Dome A (using atmospheric reanalyses and assumptions as inputs). The current title does not reflect the content and the conclusions are not well supported by the analyses and the underlying assumptions in the modelling methodology.*

**Response:** Thanks for reviewing the manuscript and the valuable comments. In the revised manuscript, we have addressed the issue of "no atmospheric mixing" by including exchanges between the boundary layer and free troposphere. Additionally, the calculations of isotope mass balance have also been modified following the new model structure. Using this modified model, we have conducted new simulations at Dome C and Dome A. The discussion has been reformulated in the revised manuscript based on the new simulated results and the feedback from the reviewers.

**2)** *The long introduction gives a good scene setting for the study, which addresses an important topic, but fails to describe the modelling framework in the context of other studies, and fails to provide a clear comparison of the meteorological and snow conditions between Dome C and Dome A (and what are the similarities and differences that need to be accounted for in comparing results for these two sites, for diurnal variations, clear and cloud sky, and winter vs summer conditions).*

**Response:** We would like to express our gratitude to the reviewer for this comment. In response to the comment, we have made some revisions to the manuscript. Specifically, in the revised manuscript, we have added a comparison of the meteorological and snow conditions between Dome C and Dome A in Section 2.2.2. The comparison of isotopic values for these two sites were also conducted at the result section (Section 3.2). Additionally, we have included a new paragraph in Section 4 to discuss the similarities and differences in diurnal variations between these two sites. We hope that these revisions will enhance the clarity and comprehensiveness of our work.

The added statement in Section 4 is as follows:
"We compared our Dome A simulations with water vapor $\delta^{18}O$, $\delta D$, and d-excess data from other East Antarctic interior sites, such as Kohnen station, Dome C, and a location about 100 km away from Dome A (Ritter et al., 2016; Casado et al., 2016; Liu et al., 2022). Both our simulations and observations show diurnal patterns, with high values during the daytime warming phase and low values during the nighttime cooling phase. However, we noticed that the observed diurnal changes in water vapor δ18O and d-excess at sites near Dome A are very large, over 40‰ and 200‰, respectively. This could be due to calibration drift caused by the extremely cold and dry conditions during the measurements at the nearest Dome A site. The averaged δD observations of 36±6‰ at Kohnen station and the in-situ measurements of 38±2‰ at Dome C are higher than our modeled δD value of 28.78±19.06‰ at Dome A. This difference can be attributed to atmospheric dynamical conditions linked with wind speed. At Dome A, the daily mean wind speed of 2.8 m/s is lower than 3.3 m/s in Dome C and 4.5 m/s in Kohnen station during summer. A lower wind speed corresponds to relatively weak air convection in the horizontal orientation. Due to the coupling between upper and lower
atmospheric layers, vertical turbulent mixing may decrease with the weakened air
convection in the atmospheric near-surface layer (Casado et al., 2018). This change can
attenuate molecular exchange between surface snow and water vapor. In parallel, the
decrease of vertical turbulence may result in a less efficient turbulent diffusion of water
molecules and an elevated contribution of molecular diffusion during atmosphere-snow
water vapor exchange. Changes in water vapor diffusion pathways increase kinetic
fractionation and reduce effective isotopic fractionation of water isotopes, leading to a
muted fluctuation of modeled water vapor δD in combination with less mass exchange."

**3)** *The description of the model has flaws in the equations for latent heat flux and*
*possibly in the use of relative humidity in the atmosphere and not relative to surface*
*temperature for fractionation coefficients. The information provided in supplementary*
*information is very difficult to understand.*

**Response:** Thanks for this comment. We found that the formulations used in the latent
heat flux calculation is not correct, following Berkowicz and Prahm (1982) (B&P82)
and the suggestion from the reviewer #2. In addition, the fractionation coefficients
calculations should rely on the humidity with respect to the surface layer and surface
temperature, rather than relative humidity and air temperature.

In the revised manuscript, we first have made modifications to the calculations of
latent heat flux. Specifically, we have revised the calculations as follows:

"$Ex = LE/Ls = -\rho_a u^* q^*$ (1)

where $\rho_a$ is dry air density varying with observed air temperature ($T_a$) and pressure ($P_a$),
$L_s$ is sublimation heat constant, u* and q* are friction velocity and specific humidity
turbulent scale, respectively. Where u* and q* are respectively defined as:

$u^* = \frac{k u_z}{log\left(\frac{z}{z_0}\right) - \Psi_M\left(\frac{z}{L}\right)}$ (2)

$q^* = \frac{k(q_a - q_s)}{log\left(\frac{z}{z_0}\right) - \Psi_M\left(\frac{z}{L}\right)}$ (3)"

We would also like to apologize for any confusion caused by our imprecise
description of the relative humidity correction and fractionation coefficients. In the new
version of the supplementary information, the text description for the relative humidity
correction have been rewritten to be more clear and accurate. Additionally, we would
like to clarify that it is the surface snow temperature ($T_s$) that controls isotopic
fractionation during air-snow vapor exchange. Thus, the surface temperature were used
to calculate fractionation coefficients, instead of air temperature. We will make the
necessary corrections to the related description in Section 2.1.1 of the revised
manuscript.

The revised supplementary information for humidity correction are as follows:
"The raw data of relative humidity (RH) at height z is the relative humidity with respect
to the water surface (RH$_w$), measured with the HMP35D humidity probe (Xiao et al.,
2008; Ding et al., 2022). The RH$_w$ can be expressed as a percentage:

$RH_w = e_w/e_w^s \times 100\%$ (S2)

where $e_w$ is the water vapor pressure of air (Pa), and $e_w^s$ is the saturated vapor pressure with respect to the water surface at the air temperature (Pa) which can be calculated using the Clausius-Clapeyron equation. When calculating the effective fractionation factor ($\alpha_f$) in the model (Eq: (15) in the main text), the $RH_w$ were converted to the relative humidity over ice at the temperature of the air ($RH_i$). The conversion between $RH_i$ and $RH_w$ was proposed based on the calibration procedures of Anderson et al. (1984). The details are as follows: 1) The $RH_w$ observations were firstly rescaled using the maximum $RH_w$ of all measured values at each air temperature point ($T_a$),

$$RH_w^{'} = RH_w(T_a)/RH_w^{max}(T_a) \tag{S3}$$

2) $RH_w^{'}$ values were then converted to $RH_i$ using Eq: (S4) :

$$RH_i = (e_w^s(T_a)/e_i^s(T_a))\times RH_w^{'} \tag{S4}$$

where $e_i^s$ represents the saturated vapor pressure with respect to ice at the air temperature (Pa). Like $e_w^s$, $e_i^s$ was calculated by the Clausius-Clapeyron equation. Based on Eq: (S3) and Eq: (S4), we obtained $RH_i$ as the final result.

In addition, the relative humidity of the air with respect to the surface temperature (h) in Eq: (14) can also be converted from $RH_w$ observations. The first step of procedures for h conversion is the rescaling $RH_w$ based on Eq: (S3), same to the $RH_i$ conversion. The second step is h calculation using the saturated vapor pressure with respect to ice at the surface temperature (Eq: (S5)).

$$h = (e_w^s(T_a)/e_i^s(T_s))\times RH_w^{'} \tag{S5}$$

**4)** *The choice of performing simulations driven by a mean diurnal cycle instead of using the actual wealth of observations is unclear and the implications should be discussed. I am puzzled by how wind effects are accounted for when averaging conditions.*

**Response:** Thanks for this comment. We chose to use the mean stacked conditions to conduct simulation since we wanted to highlight the effects of air-snow exchange in a general case. But in order to avoid confusion, in the revised manuscript, the simulations were conducted using continuous meteorological input for each individual day during the studied period at Dome C. This allowed us to calculate the average diurnal changes in water vapor isotopic composition and snow isotopes. However, for the Dome A case, the selected days for clear-sky, cloudy, and winter conditions were not continuous, making it difficult to conduct simulations as was done for Dome C. Instead, we were only able to use the model for one day to simulate the diurnal changes in snow and water vapor isotopes, after a week of spin-up time (as shown in Figures 4-6 in the revised manuscript). This allows to evaluate the effects of air-snow exchange under representative meteorological conditions.

In addition, we also reconsidered the effect of wind speed on simulations during atmosphere-snow water vapor exchange. In the revised manuscript, a new case simulation was presented to test the effect of wind speed variability on atmosphere-snow water vapor exchange. Specifically, we analyzed the response of water vapor and snow isotopic composition to the conditions of a significant diurnal cycle of wind versus that with averaged wind speed. The results, as shown in Figure 1, suggest that strong variability in wind speed will enlarge the variations in latent heat, leading to a more significant diurnal change in water vapor isotopes and snow isotopes, but for a longer time, there would be days with diurnal wind cycle both smaller or bigger than the mean, so the result with the mean wind pattern is more representative. These discussion has been added into the Supplementary Information (Text S3)

[Figure]

**Figure 1:** The comparison of water vapor isotopic composition between two simulated cases at Dome A. The simulations in two cases were driven using the averaged wind speed (Case I) and the strong diurnal changes in wind speed (Case II).

**5)** *There should be at least a more detailed comparison between the Dome C and Dome A characteristics (including comparison of meteorological conditions and ERA5 results at both sites), instead of current Table 1 (where assumptions versus observational based information should be differenciated).*

**Response:** Thanks for this suggestion. In the revised manuscript, we have added content to compare the meteorological conditions at Dome C and Dome A in Section 2.2.2, and the impacts of these conditions on the modeled water vapor and snow isotopes are discussed in Section 4.

**6)** *The assumptions displayed in Figure 1 should be discussed in the context of available information, including the Richardson number, regarding atmospheric exchanges (the closed box assumption validity).*

**Response:** Thanks for the valuable suggestion. We have incorporated these into the revised manuscript by discussing the assumptions related to the occurrence conditions of the air-mass renewal process associated with the Richardson number, as well as the isotopic fractionation during sublimation and deposition. Additionally, we have addressed the setting of initial conditions through some original and new sensitivity
tests.
Here, we will provide the discussion of the occurrence conditions of the air-mass
renewal process in the supplementary information (Text S3):
"To determine the correlation between mixing occurrence conditions and Richardson
numbers, we ran simulations for Dome C, taking into account mixing when Ri<0 and
Ri<0.1. As shown in Figure 2, the case with Ri<0 did indeed underestimate the water
vapor isotopic composition in the near-surface atmospheric layer during the cooling
time. Based on this comparison, we incorporated mixing into the modeling once Ri<0.1."

[Figure]

**Figure 2:** The comparison of water vapor isotopic composition between the simulated
and observed changes at Dome C. Two simulated cases are presented here to discuss
the occurrence condition of mixing. In the case I, the mixing is assumed to happen when
Ri<0 in the cooling phase. The case II for the occurrence conditions of mixing is Ri<0.1
in the cooling phase.

**7)** *The authors should reflect on what their model explicitly implies in terms of
behaviour, and what is effectively "validated" from their approach which does not
resolve the diurnal variations in snow measured at Dome C. This physics-based
approach is missing.*

**Response:** Thank you for bringing this to our attention. We have resolved the issue by
making modifications to the physical mechanism of our model (Figure 1), as outlined
in our previous response to general comments. We then conducted simulations under
Dome C conditions and three different cases at Dome A using the updated model. The
simulated results for a 24-hour period are presented in Figures 3-6 of the main text (at
the end of this response). The new results indicate that the changes in snow isotopic
composition are significantly greater than the original $\delta^{18}O$ simulations of 0.02‰ at
Dome C. During a typical night, such as the frost event on January 6-7, 2015, the diurnal
changes of the newly simulated results between the maximum and minimum can reach

2‰ for snow δ¹⁸O (as shown in Figure 3). This magnitude is consistent with the
observations for snow isotopes from Casado et al. (2018).

[Figure]

**Figure 3:** The changes of snow isotopes and water isotopic composition in the near
surface atmospheric layer during the 6-7th Jan, 2015 at Dome C.
**8)** *For these reasons, major revisions are needed, first to ensure accurate equations in*
*the model, and then to reflect on the limitations and suitability of the core assumptions*
*of the closed box model to address these questions, and third regarding the average*
*diurnal cycle approach, and fourth regarding the detailed comparison between Dome*
*C and Dome A (well beyond "validating" and "applying" this model at the two sites).*
**Response:** Thank you for the helpful comment. Several significant changes were made
to the model structure to reflect reviewer's suggestions. Specifically, we have added a
third box to represent the free atmosphere layer. The calculations and equations were
also updated to reflect the modifications made to the physical mechanism of the model.
We also have presented new assumptions for initial conditions and air mass renewal
occurrence conditions, which enable the model to run effectively. Furthermore, the
simulations were continuously conducted using meteorological observations recorded
hourly. Finally, we have included a comparison between Dome C and Dome A in the
Discussion section of the revised manuscript (Details can be seen in response to
Comment #2 and Comment #6). After all of these modifications, in addition to that
arisen by other reviewers, the main conclusion of the manuscript stays the same: The
diurnal variations in atmospheric water vapor δ¹⁸O and δD can reach 4.75±2.15 ‰ and
28.79±19.06 ‰ under summer clear-sky conditions at Dome A, with corresponding
diurnal variations in surface snow δ¹⁸O and δD by 0.81±0.24 ‰ and 1.64±2.71 ‰,
respectively. After 24-hour simulation, snow water isotopes were enriched under clear-
sky conditions. However, there is no or very little enrichment for snow water isotopes under cloudy conditions. Under winter conditions at Dome A, the model still indicates the diurnal change in atmospheric and surface snow water isotopes are not significant, but the model predicts more or less depletions in snow $\delta^{18}O$ and $\delta D$ in the period of 24-hour simulation, opposite to the results under summer clear-sky conditions. This suggests that the air-snow vapor exchange tends to enlarge snow water isotope seasonality.

**End of the responses to Reviewer #3**

**Reference**

Casado, M., Landais, A., Picard, G., Münch, T., Laepple, T., Stenni, B., et al.: Archival processes of the water stable isotope signal in East Antarctic ice cores, *The Cryosphere*, 12(5), 1745-1766, doi: 10.5194/tc-12-1745-2018, 2018.

Ritter, F., Steen-Larsen, H. C., Werner, M., Masson-Delmotte, V., Orsi, A., Behrens, M., et al.: Isotopic exchange on the diurnal scale between near-surface snow and lower atmospheric water vapor at Kohnen station, East Antarctica, *The Cryosphere*, 10(4), 1647-1663, doi: 10.5194/tc-10-1647-2016, 2016.

Wahl, S., Steen-Larsen, H. C., Reuder, J., & Hörhold, M.: Quantifying the Stable Water Isotopologue Exchange Between the Snow Surface and Lower Atmosphere by Direct Flux Measurements, *Journal of Geophysical Research: Atmospheres,* 126(13), doi: 10.1029/2020jd034400, 2021.

---

## Referee Report (RR1)

**Review of manuscript tc-2023-76 "A model framework on atmosphere-snow water vapor exchange and the associated isotope effects at Dome Argus, Antarctica: part I the diurnal changes" by Tianming Ma et al.**

**General comments:**

The authors have mostly (see Comment L.147) addressed the flaws in the latent heat flux calculation and the model theory. Sufficient uncertainty and sensitivity analyses are performed, yet the outcomes of these analyses or their implications for the results' robustness are not addressed in the discussion. I suggest adding a discussion of the results' robustness based on the sensitivity analyses in 3.4, Text S4, and Text S5 before publication.

The authors have now simulated the impact of vapor fluxes for a continuous time series at Dome C. However, for Dome A, their simulation is still based on average diurnal cycles of the input data. To assess how averaging the input data might affect the results (see Comment L.416-417), I suggest one additional simulation for Dome A with continuous input data, regardless of the cloudiness.

The authors make statements at several points in the manuscript (e.g., Comments L23., L.397, L.463-464, L.475-477) without providing sufficient evidence. I kindly ask the authors to revise such statements and reformulate them appropriately. I further suggest conducting a comprehensive language revision of the manuscript, as occasional imprecise formulations may lead to misinterpretation. Lastly, in its current form, the manuscript is missing references to figures and supplemental material wherever relevant. This makes it difficult to follow the authors' explanations, and I strongly advise providing all references before the publication of this manuscript.

**Detailed comments:**

- L.23: Please add: under "average" summer clear-sky conditions. It is important to distinguish the isotopic impact of an average clear-sky from the average impact during all clear-sky days, as both cases could differ significantly.

- L.28: Please clarify what is meant by "more or less".

- L.54: Estimates of the long-term effect of atmosphere-snow water vapor exchange on the snow isotopic composition in Greenland have been done by Dietrich et al. (2023), but not yet in Antarctica.

- L.144 (and others): I suppose the used Formula is either the "August–Roche–Magnus Formula" or the "Magnus Formula" to calculate the saturation-specific humidity since $q_s$ cannot be directly calculated from the Clausius-Clapeyron Equation? Please add the name of the used formula.

- L.144, L.146, Table S1: Please correct to "Clausius-Clapeyron".

- L.147: If I understood correctly, you set the correction term $\Psi_M$ to zero. Firstly, this needs to be stated in the manuscript. Secondly, under the mostly stable conditions in polar regions, the stability correction terms $\Psi_M$ and $\Psi_q$ cannot be neglected. Please include $\Psi_M$ and $\Psi_q$ in your latent heat flux calculation. E.g. following Holtslag and De Bruin (1988) for stable conditions (assuming $\Psi_M=\Psi_q$), and Paulson (1970) unstable conditions.

- L.261: Typo: negative -31.01°C

- L.269: Why is it relevant to mention stellar images here? Please remove or clarify.

- L.291: I presume data from the model "ECHAM5-wiso" is used (Werner et al. 2011).

- L.360: Diurnal changes, not cycles.

- L.397: Neither of the three figures supports this statement since none shows isotopic values.

- L.407-408: Please reference Text S4 here.

- L.411: Liu et al. (2022) are not in the reference list.

- L.413-416: "We noticed": Where are these diurnal changes shown? Or is it Liu et al. (2022) who show these changes? I furthermore assume that it is a 200‰ change in $\delta D$, not in d-excess.

- L.416-417: The given numbers correspond to the diurnal variations, not the absolute values of $\delta D$. Please correct. In addition, where do the values for Kohnen and Dome C come from? Please update the text with the references.

- L.416-417: In the supplements is shown that an increased wind speed variability leads to a larger diurnal magnitude of the vapor $\delta^{18}$O. I suspect the lower diurnal magnitude to be a consequence of the averaged meteorological input. This could be tested by running an additional simulation for Dome A with continuous meteorological input without distinguishing between cloudy and clear-sky days. Please add a sentence that references and discusses the results from Figure S3.

- L.420: Vertical convection or horizontal advection?

- L.439-440: Figures 4 and 5 show no general vapor depletion. I suggest removing the second part of this sentence.

- L.463-464: The presented results and figures do not provide sufficient evidence that allows a statement regarding the long-term isotopic impact of vapor exchange. Please provide evidence or remove this statement.

- L.475-477: The evidence for this statement is missing.

- Figure 4d: The line for the snow isotopic composition is missing in the figure.

- Figure 4 and Figure 5: Please choose the same y-axis ranges for Fig. 4 and 5.

- Text S4, L.123-126: Figure S2 shows that the vapor $\delta^{18}$O is strongly underestimated in Case II (turbulent mixing for Ri¡0.1). The written text suggests the opposite.

- Figure S3: Typo in legend of "Water vapor $\delta^{18}$O-Case II"
* * *
L. J. Dietrich, H. C. Steen-Larsen, S. Wahl, T. R. Jones, M. S. Town, and M. Werner. Snow-atmosphere humidity exchange at the ice sheet surface alters annual mean climate signals in ice core records. *Geophysical Research Letters*, 50(20):e2023GL104249, 2023. doi:10.1029/2023GL104249.

A. Holtslag and H. De Bruin. Applied modeling of the nighttime surface energy balance over land. *Journal of Applied Meteorology and Climatology*, 27(6):689–704, 1988.

C. A. Paulson. The mathematical representation of wind speed and temperature profiles in the unstable atmospheric surface layer. *Journal of Applied Meteorology and Climatology*, 9(6):857–861, 1970.

M. Werner, P. M. Langebroek, T. Carlsen, M. Herold, and G. Lohmann. Stable water isotopes in the echam5 general circulation model: Toward high-resolution isotope modeling on a global scale. *Journal of Geophysical Research: Atmospheres*, 116(D15), 2011. doi:10.1029/2011JD015681.

---

## Referee Report (RR2)

**Comments on "A model framework on atmosphere-snow water vapor exchange and the associated isotope effects at Dome Argus, Antarctica: part I the diurnal changes" by Ma et al.**

The authors developed a box model to quantify the atmosphere-snow water vapor exchange and the isotopic effect at sites with very low snow accumulation rates where the atmosphere-snow exchange is an important post-depositional process that can significantly influence the isotopic compositions in surface snow. The model is better than the simple Rayleigh distillation models in previous studies because it parameterizes atmosphere-snow water vapor exchange by using the bulk aerodynamic method. After reading the revised paper carefully, I suggest to publish it after a minor revision. I gave several minor comments below.

Line 85, "box model." changes to 'box model'.

Line 99-100, the authors indicate that at Dome A the time interval between two precipitation events can reach ~80 days. Please give a reference here.

Line 235-237, the authors indicate that the isotopic composition of vapor in the free atmosphere layer ($\delta f0$) is greater than the isotopic composition of vapor in the boundary layer ($\delta v0$). This is due to the contribution from the free atmosphere can increase the ratio of $H_2^{18}O$ molecules in the boundary layer (Casado et al., 2018). Why the contribution from the free atmosphere can increase the ratio of $H_2^{18}O$ molecules in the boundary layer? The authors should explain it in more details.

Line 237, 'Casado et al. 2018' should be 'Casado et al., 2018'.

Line 410-411, the authors indicate that as $\delta^{18}O_{s0}$ decreases, the magnitude of $\delta^{18}O_s$ diurnal changes decreases. But in Fig. 8e, we can see that as $\delta^{18}O_{s0}$ decreases, the magnitude of $\delta^{18}O_s$ diurnal changes increases (not decrease). Is that right?

Line 426, 'Fig. 2, 4c, and 5c' should be 'Figs. 2, 4c, and 5c'.

Line 449, 'data from Dome A simulations' changes to 'data at Dome A'.

Line 526, 'by those in summer' changes to 'by those in winter'; 'snow isotopes..'' changes to 'snow isotopes.'.

---

## Author Response (AR2)

Dear Dr. Smith,

Please find our revised manuscript "A model framework on atmosphere-snow water vapor exchange and the associated isotope effects at Dome Argus, Antarctica: part I the diurnal changes " by Ma et al. We have explicitly addressed all the comments and suggestions from the two reviewers. Below we briefly described the main comments and our responses.

One of the main comments/suggestions from the first reviewer was that the conclusions would require significant modifications if the influence with free atmospheric layer were incorporated into the model. When involving the effects of the free atmosphere in the revised manuscript, in summer, the diurnal variations in snow isotopes become larger, so as the enrichments after 24 hours and/or longer duration. In winter, the modeled diurnal variation and changes (i.e., depletion) after 24 hours and/or longer duration also become larger, though the absolute values are still much smaller compared to that in summer conditions. As a result, the key conclusion that vapor exchange at the atmosphere-snow interface leads to a larger seasonality of snow isotopes holds the same. This is why in the response we state "the conclusions are unchanged".

This reviewer also questioned why the modeled the snow $\delta D$ amplitude is so small. We compared our modeled of snow $\delta D$ amplitude of $(1.6 \pm 2.71)$ ‰ with the observed amplitude of $\sim 3$ ‰ at the Konhen Station. They are in fact comparable, and the lower value at Dome A is due to the lower wind speed at Dome A. This reviewer also has pointed out some errors or questions, which we have explicitly addressed as stated in the response file.

The comments from the second reviewer were focused on the continuous simulations at Dome A and sensitivity tests. We have included two additional simulations that utilize continuous meteorological inputs for summer and winter days at Dome A. The details of these simulations can be found in Section 2.2 and Section 3.2.4 of the revised manuscript. We also reformulated the description of sensitivity test results in Section 3.4, discussing how the factors tested influenced the simulations of diurnal variations in water vapor isotopes and snow isotopes. In addition to these technical improvements, we conducted a comprehensive language revision of the manuscript and the addition of some new references in the manuscript.

We have also conducted a comprehensive language revision of the manuscript with the assistance of Nature AI language tool. We hope this would improve the writing.

We confirm that this manuscript has not been published elsewhere and is not under consideration by another journal. All authors have approved the manuscript and agree with its submission to The Cryosphere. Please address all correspondence to genglei@ustc.edu.cn. We look forward to hearing from you at your earliest convenience.

Sincerely,

      Lei Geng, Ph.D
      Professor
      School of Earth and Space Sciences
University of Science and Technology of China
      96 Jinzhai Road, Hefei, Anhui, 230026
      Email: genglei@ustc.edu.cn

**Response to Reviewer #1's comments**

**General comments:**

*1) While the authors have been updated the models that they used in the manuscript, the conclusions are unchanged, as stated by the authors in the response. This is surprising, because the added influence with the exchange with the free atmosphere should create extremely significant change to the vapour boundary layer. It's difficult to evaluate how this has actually been computed by the authors without in depth evaluation of what was done, which is beyond my duty as a reviewer.*

**Response:** We thank the reviewer for this question. To clarify the point, that the key conclusion of the original manuscript without involving the effects of the free atmosphere is vapor exchange at the air-snow interface would tend to enlarge the magnitude of seasonal snow isotope variation, as it causes enrichments in surface snow isotopes in summer, while more or less depletions in winter.

When involving the effects of the free atmosphere in the revised manuscript, in summer, the diurnal variations in snow isotopes become larger, so as the enrichments after 24 hours and/or longer duration. In winter, the modeled diurnal variation and changes (i.e., depletion) after 24 hours and/or longer duration also become larger, though the absolute values are still much smaller compared to that in summer conditions.

As a result, **the key conclusion that vapor exchange at the atmosphere -snow interface leads to a larger seasonality of snow isotopes holds the same**. This is why in the response we state "the conclusions are unchanged".

Other than the above mentioned points, we did revised a bit of the conclusion: since based on the simulated results with or without the effects of free troposphere, the modeled changes in winter is not comparable to (i.e., lower than) those in summer, due to much stable boundary layer condition in winter. This makes the effects in summer can't be offset by summer, leading to overall enrichments in snow isotopes.

*2) It's a bit surprising that the snow isotopic composition is shown as anomaly in Figure 3d, which makes me suspect that the simulation estimates were not matching with the observations in order to obtain the observed values for the vapour.*

**Response:** Thanks for this question. Yes, in Figure 3d, we chose to show the modeled difference instead of the absolute values. It is indeed that the modeled absolute snow $\delta^{18}O$ doesn't make the observed surface $\delta^{18}O$ at Dome C. However, the reason is that in the model we used the isotopic composition of fresh summer snow (~-47‰ reported from Touzeau et al., (2016)) as initial snow isotope composition, which is higher than $\delta^{18}O$ observed in surface snow (i.e., -51.16‰). if we replaced the model initial value of -47‰ with -51.16‰, then the modeled absolute values are consistent with the observations (Figure 1 in this response).

In the previous versions, we chose $\delta^{18}O$ of fresh snow as the initial value to better constrain the changes due to air-snow exchange. To avoid confusion, in the revised manuscript, we replotted Figure 3 with $\delta^{18}O$ observed in surface snow (i.e., -51.16‰) as the model initial values, to make the results more consistent with the observations reported by Casado et al., (2018).

[Figure]

**Figure 1:** The simulations and observations of snow and water isotopic composition in the near surface atmospheric layer during the Jan 6-7$^{th}$, 2015 at Dome C (the initial snow isotopic composition is -51.16‰ for running simulations).

*3)The snow isotopic composition is missing in Figure 4d, so it's difficult to know what was to be seen there, but it seems like the amplitude of the snow dD variations are extremely small, which is not very realistic.*

**Response:** Sorry, yes we forgot the put the snow $\delta^{18}$O data there, and in the revised manuscript, we have added it. Regarding the amplitude of snow $\delta$D, the model gave an estimate of 1.6±2.71‰ for Dome A clear-sky conditions (Figure 4e of the main text). This value is a little smaller than the observed peak-to-peak amplitude of ∼ 3‰ at Konhen Station in January. Such a difference can be attributed to the lower wind speed at Dome A (Text S4 in the Supplementary).

At Dome C, there is no available data of snow $\delta$D on the diurnal scale. Only one paper reported an observed value of ~2‰ for a peak-to-peak amplitude of diurnal variations in snow $\delta^{18}$O during a frost event (Casado et al., 2018). This reported value is significantly higher than the average of simulated snow $\delta^{18}$O variations at Dome A (0.8±0.35‰). The averaged meteorological input (including days in December, January, and February) used in Dome A simulations is the reason of the smaller diurnal amplitude of snow $\delta^{18}$O. For example, the averaged wind speed at Dome A (2.8 m/s) is lower than that in Dome C (3.3 m/s), leading to a less effective exchange between snow and water vapor. We consider individual day simulation, the diurnal amplitude in snow $\delta^{18}$O diurnal variations can exceed 1.5‰ at Dome A, as evidenced by continuous simulations in January (Figure 7 of main text). This is also comparable to the observations at Dome C.

*4)The winter conditions shown in Fig. 6 are clearly started with non matching vapour and snow isotopic composition since the vapour isotopic composition is converging toward a different value.*

**Response:** We have re-examined Figure 6 and the winter simulation results using the updated model. It has come to our attention that there is a discrepancy between the simulation results in snow d-excess and the curve depicted in the submitted figure. This error has arisen from the extensive modifications made in this study, leading to our confusion between the calculation results with the updated model and those with the previous model. In the revised manuscript, we have corrected this mistake. The corrected result does not have the issue of converging toward a different value (Figure 2 of this response or Figure 6 of main text).

[Figure]

**Figure 2:** The simulated hourly mean vapor exchange flux and variations in atmospheric water vapor and snow isotopes under winter conditions at Dome A: (a) Richardson number, (b) friction velocity, (c) vapor exchange flux, (d) snow and water vapor $\delta^{18}$O, (e) snow and water vapor $\delta$D, (f) snow and water vapor d-excess. The uncertainties for each variable are displayed by shaded area in each subpanel.

*5) Overall, I'm sure that the authors undertook a tremendous amount of work, and that this could potentially be an interesting manuscript, but I feel like the rigour of model shown here, and the application to Dome A conditions, is not sufficient.*

**Response:** We appreciate the reviewer's time and efforts to evaluate this manuscript, we agree that there could be still rooms to improve the manuscript even after we have revised significantly according to the reviewer's constructive suggestions. But nevertheless, we think the results of this manuscript are new, since the results indicate this special kind of post-depositional processing, i.e., vapor exchange at the air snow interface is tending to enlarge the seasonal variations in snow isotopes, not as other processing tending to smooth the variability. We think this is a good enough point to elucidate and it shall inspire new observations or experiments to confirm this in the future.

**End of the responses to Reviewer #1**

**Reference**

Casado, M., Landais, A., Picard, G., Münch, T., Laepple, T., Stenni, B., et al.: Archival processes of the water stable isotope signal in East Antarctic ice cores, The Cryosphere, 12(5), 1745-1766, doi: 10.5194/tc-12-1745-2018, 2018.

Touzeau, A., Landais, A., Stenni, B., Uemura, R., Fukui, K., Fujita, S. et al.: Acquisition of isotopic composition for surface snow in East Antarctica and the links to climatic parameters, The Cryosphere, 10(2), 837-852, doi: 10.5194/tc-10-837-2016, 2016.

**Response to Reviewer #2's comments**

**General comments:**

*1) The authors have mostly (see Comment L.147) addressed the flaws in the latent heat flux calculation and the model theory. Sufficient uncertainty and sensitivity analyses are performed, yet the outcomes of these analyses or their implications for the results' robustness are not addressed in the discussion. I suggest adding a discussion of the*

*results' robustness based on the sensitivity analyses in 3.4, Text S4, and Text S5 before publication.*

**Response:** Thank you for the feedbacks, and we also appreciate the valuable suggestions. In the revised manuscript, we have revised the description of the sensitivity analysis results in Section 3.3. Based on the sensitivity test results, we have added two paragraphs in the discussion section of main text to discuss the results from sensitivity tests.

   The revised Section 3.3 are as follows:

[revised manuscript text omitted]

*2) The authors have now simulated the impact of vapor fluxes for a continuous time series at Dome C. However, for Dome A, their simulation is still based on average diurnal cycles of the input data. To assess how averaging the input data might affect the results (see Comment L.416-417), I suggest one additional simulation for Dome A with continuous input data, regardless of the cloudiness.*

**Response:** Thanks for your nice suggestion. We have added two additional simulation cases running with continuous meteorological inputs at Dome A site. One case is realized on summer days disregarding the influence of clouds, the other one is on winter days. The running duration for two cases are 11 days, consistent with the Dome C simulations. The selected period for summer simulation is from 5th to 16th of January for each year during 2006-2011(data were not observed in 2005). The winter period for simulations is 5th-16th, July. Thus, 6 groups of simulated results for each season can be obtained to calculate the average of continuous changes in water vapor and snow $\delta^{18}O$, as shown in Figure 1.

The continuous simulations at Dome A show opposite trends for changes in snow isotopes between summer and winter simulation conditions (Figure 1b and 1d). This supports that the seasonal snow isotope variations can be enlarged due to the snow-atmosphere water vapor exchange process. Also, the annual net effects can lead to an increase in the annual mean value of snow isotopic composition, in consideration of a more significant isotopic effects in summer. These important simulations and conclusions have been added into the main text (Method (Section 2.2), Results (Section 3.2), and Discussion).

[Figure]

**Figure 1:** The continuous simulations in snow and water vapor isotopes at Dome A. Panel a) and b) respectively represents summer simulations in a 11-day period (Jan 5-16$^{th}$, 2006-2011), Panel c) and d) are same to Panel a) and b), but for wintertime (Jul 5-16$^{th}$, 2006-2011). In all panels, the light lines represent the simulated results of water vapor $\delta^{18}O$ for each year during the simulation period. The bold solid line and the light blue shadow are the averages (AVG) and standard deviations (SD) of $\delta^{18}O$ simulations in each year, respectively.

*3) The authors make statements at several points in the manuscript (e.g., Comments L23., L.397, L.463-464, L.475-477) without providing sufficient evidence. I kindly ask the authors to revise such statements and reformulate them appropriately. I further suggest conducting a comprehensive language revision of the manuscript, as occasional imprecise formulations may lead to misinterpretation. Lastly, in its current form, the manuscript is missing references to figures and supplemental material wherever relevant. This makes it difficult to follow the authors' explanations, and I strongly advise providing all references before the publication of this manuscript.*

**Response:** We are really grateful to the reviewer for the rigorous considerations. We have made the necessary revisions following the detailed comments provided by reviewer #2. A comprehensive check to the manuscript has been conducted, ensuring that erroneous sections have been rectified to the best of our ability. Additionally, efforts have been made to add the references as much as possible. Given the substantial modifications made to the revised manuscript, the individual sentence or section revisions are not listed one by one here. The reviewers can refer to the tracked changes version for a detailed overview of specific modifications. It is our sincere belief that these revisions will enhance clarity and comprehension of the article for all readers.

**Detailed comments:**
*L.23: Please add: under "average" summer clear-sky conditions. It is important to distinguish the isotopic impact of an average clear-sky from the average impact during all clear-sky days, as both cases could differ significantly.*
**Response:** Thanks, added.

*L.28: Please clarify what is meant by "more or less".*
**Response:** Thanks, we meant to express that the changes are small or negligible, but removed *"more or less"* to make this sentence more clear.

*L.54: Estimates of the long-term effect of atmosphere-snow water vapor exchange on the snow isotopic composition in Greenland have been done by Dietrich et al. (2023), but not yet in Antarctica.*
**Response:** Thanks for this suggestion. We reformulated this sentence as following:
*Isotopic effects associated with atmosphere-snow water vapor exchange at longer time*
*scales have been done at Greenland Ice Sheet (Dietrich et al., 2023), but not yet in*
*Antarctica.*

*L.144 (and others): I suppose the used Formula is either the "August–Roche–Magnus Formula" or the "Magnus Formula" to calculate the saturation-specific humidity since*
*$q_s$ cannot be directly calculated from the Clausius-Clapeyron Equation? Please add the name of the used formula.*
**Response:** Thanks for pointing out this inappropriate statement. The $q_s$ calculations was based on August–Roche–Magnus Formula in this study. We have added the name of the used formula in the L.144 of revised manuscript.

*L.144, L.146, Table S1: Please correct to "Clausius-Clapeyron".*
**Response:** Thanks, correct.

*L.147: If I understood correctly, you set the correction term $\Psi M$ to zero. Firstly, this*
*needs to be stated in the manuscript. Secondly, under the mostly stable conditions in polar regions, the stability correction terms $\Psi M$ and $\Psi q$ cannot be neglected. Please include $\Psi M$ and $\Psi q$ in your latent heat flux calculation. E.g. following Holtslag and De Bruin (1988) for stable conditions (assuming $\Psi M = \Psi q$), and Paulson (1970) unstable conditions.*
**Response:** Thanks for your rigorous consideration. In fact, we have used the correction term $\Psi M$ and $\Psi q$ when calculating latent heat flux calculations in the revised version. The $\Psi M$ and $\Psi q$ was set following Louis et al., (1979) for stable and unstable conditions. This chosen parameterization scheme is characterized by high calculating efficiency, compared to the iteration method like Paulson (1970) for unstable conditions. However,
the correction terms were not added into the equations listed in the manuscript due to our carelessness. Also, there is no sufficient statement on the setting of $\Psi M$ and $\Psi q$ and the chosen of parameterization scheme in the Section 2.2.2. In the revised manuscript, we have corrected the equations and added some sentences to bring convenience for readers. The details are as follows:

*The ΨM is calculated for stable, unstable and neutral boundary layer using the functions taken from Louis (1979).*

*L.261: Typo: negative -31.01℃*

**Response:** Thanks, correct.

*L.269: Why is it relevant to mention stellar images here? Please remove or clarify.*
**Response:** Thanks for this comment. Previous studies have used two different methods, i.e., sonic radar and seeing—the angular size of stellar images to determine the boundary layer height at Dome A. The measurements from sonic radar were only conducted from 2009 February to 2009 August, whereas the seeing—the angular size
of stellar images were mainly performed during 2019. All of them confirm a median thickness of approximately 14 metres for the boundary layer at Dome A. Thus, we mentioned the stellar images here to ensure the credibility of estimation for the boundary layer height at Dome A.

*L.291: I presume data from the model "ECHAM5-wiso" is used (Werner et al. 2011).*
**Response:** No, when deriving the δ-T slope, we didn't use ECHAM5-wiso data. The compiled data of precipitation isotopic composition in Pang et al. (2019) were collected from previously published papers, including Landais et al., (2012), Touzeau et al., (2016), Stenni et al., (2016), Touzeau et al., (2016), Casado et al., (2016) and Ritter et
al., (2016). These observations have been used to obtain the δ-T slope and then calculate the $\delta^{18}O_{s0}$ in winter season at Dome A. The ECHAM5-wiso data was then used to compare with the calculated $\delta^{18}O_{s0}$ in winter season at Dome A, to verify the calculations. These explanations have been stated in the previous response and added to the main text.

*L.360: Diurnal changes, not cycles.*
**Response:** Thanks, correct.

*L.397: Neither of the three figures supports this statement since none shows isotopic*
*values.*
**Response:** Thanks for this valuable suggestion. We changed other 3 images (Fig.2, Fig.4c and Fig.4d) to support our statement in the revised manuscript.

*L.407-408: Please reference Text S4 here.*
**Response:** Thanks, added.

*L.411: Liu et al. (2022) are not in the reference list.*
**Response:** Thank you so much for your careful check. We have added this reference in the list.

*L.413-416: "We noticed": Where are these diurnal changes shown? Or is it Liu et al. (2022) who show these changes? I furthermore assume that it is a 200‰ change in δD, not in d-excess.*

**Response:** We very appreciate the reviewer's comment. The observed diurnal changes
in water vapor isotopic composition at the nearest Dome A site are shown in Liu et al. (2022). For the d-excess, it has large diurnal variations with an amplitude of ~200‰ (Fig.2). The exact reason is still unclear, but could be the calibration drift caused by the extremely cold and dry conditions during the measurements. In the revised manuscript, we added the reference at L.413-416 and a sentence to make it more clear for readers.

[Figure]

**Figure 2:** Diurnal cycles of water vapor d-excess during the measuring method from Zhongshan to Dome A (cited from Liu et al., 2022). The color successive change represents gradual distance variation from near coastal to interior inland Antarctica. All the signals are dominated by the presence of diurnal cycles with the isotope variation
amplitude increased to interior.

*L.416-417: The given numbers correspond to the diurnal variations, not the absolute values of δD. Please correct. In addition, where do the values for Kohnen and Dome C come from? Please update the text with the references.*

**Response:** We would like to express our gratitude to the reviewer for pointing out these errors. The descriptions of δD observations have been reformulated according to the reviewer's suggestions. The related references have also been added to the end of each value. The revised sentences are as follows:

*Our modeled δD variations at Dome A (28.78±19.06‰) are lower than the observed*
*diurnal variations in water vapor δD at Kohnen station (36±6‰ from Ritter et al., (2016)) and at Dome C (38±2‰ from Casado et al., (2016)).*

*L.416-417: In the supplements is shown that an increased wind speed variability leads to a larger diurnal magnitude of the vapor δ18O. I suspect the lower diurnal magnitude*
*to be a consequence of the averaged meteorological input. This could be tested by running an additional simulation for Dome A with continuous meteorological input without distinguishing between cloudy and clear-sky days. Please add a sentence that references and discusses the results from Figure S3.*

**Response:** Thanks for the reviewer's nice suggestion. The simulations at Dome A site have been done with continuous meteorological input from 5 to 16 January in each year (2006-2011). As shown in Figure 1 of this response, we find the water vapor $\delta^{18}O$ has a clear diurnal variation with higher values in sublimation period and lower values in deposition period. This pattern is consistent with the simulations using the averaged meteorological input. For the diurnal magnitude, the continuous simulations are indeed higher than those from the averaged meteorological input on several individual days. However, the diurnal variations in most of days are close or lower than 4.75‰, which was calculated with the averaged meteorological input in summer clear-sky days. This comparison suggests that the data processing method for model input will not cause a lower diurnal magnitude showing in water vapor $\delta^{18}O$.

Additionally, the discussion on the results from Figure S2 (Figure S3 in the original edition) and related reference have been added into the manuscript. The details are as follows:

*Wind speed also plays a key role in driving isotopic variations at Dome A, because its increase can amplify the variations in latent heat, leading to more pronounced diurnal changes in water vapor and snow isotopic composition (Supplementary Text S4, Bréant et al., 2019).*

*L.420: Vertical convection or horizontal advection?*
**Response:** It is vertical convection. This error has been corrected in the revised manuscript.

*L.439-440: Figures 4 and 5 show no general vapor depletion. I suggest removing the second part of this sentence.*
**Response:** Thanks for this suggestion. The second part of L.439-440 has been removed from the manuscript.

*L.463-464: The presented results and figures do not provide sufficient evidence that allows a statement regarding the long-term isotopic impact of vapor exchange. Please provide evidence or remove this statement.*
**Response:** From the diurnal simulations, it is apparent that the surface snow $\delta^{18}O$ and $\delta D$ would become enriched compared to fresh snow in summer, while in winter surface snow isotopes would be depleted compared to fresh snow. If the diurnal changes could be accumulated and other isotopic modifications were not taken into account, an amplification of the snow isotope seasonality would be caused by atmospheric vapor-snow exchange. This assumption indeed needs to be supported by more calculations. We originally planned to do the seasonal simulations at Dome A, but these work are out of the scope of this paper. Thus, the statement on long-term isotopic impact of vapor change has been removed from this manuscript right now. We expected to provide more evidence to support this statement in the future work.

*L.475-477: The evidence for this statement is missing.*
**Response:** Thanks for this comment. Based on the simulations in the Figure 4 and

Figure 6, we found that the snow isotopes become more enriched after a 24-h period during summer. In contrast, the winter snow layer has an opposite change in $\delta^{18}O/\delta D$ on the diurnal scale. If other post-depositional processes and precipitation intermittency are not considered, the diurnal changes in snow isotopes induced by atmosphere-snow vapor exchange will be accumulated under the ideal condition. Considering the opposite effect between summer and winter, the annual net effect from atmosphere-snow vapor exchange will be small on the snow isotopes. While this inference holds true from a qualitative perspective, it remains to be more explored in the future work. These explanations have been added into the discussion.

*Figure 4d: The line for the snow isotopic composition is missing in the figure.*
**Response:** Sorry for this mistake. We have added the line representing snow isotopic composition in the Figure 4d of revised manuscript.

*Figure 4 and Figure 5: Please choose the same y-axis ranges for Fig. 4 and 5.*
**Response:** Thanks for this valuable comment. After adjustment, the y-axis ranges for Figure 4 are in accord with those of Figure 5.

*Text S4, L.123-126: Figure S2 shows that the vapor $\delta^{18}O$ is strongly underestimated in Case II (turbulent mixing for Ri¡0.1). The written text suggests the opposite.*
**Response:** We apologize for our carelessness in the Text S4, L.123-126, and Figure S2. The Case I represents the simulations under the turbulent mixing for Ri<0.1, whereas the Case II shows the modeled results when Ri<0. In the revised edition, we have corrected this error in the Figure S1 (Figure S2 in the original edition).

*Figure S3: Typo in legend of "Water vapor $\delta$ 18O-Case II"*
**Response:** Thanks, corrected.

**End of the responses to Reviewer #2**

---

## Author Response (AR3)

Dear Dr. Smith,

Please find our revised manuscript "A Model Framework on Atmosphere-snow Water vapor Exchange and the Associated Isotope Effects at Dome Argus, Antarctica: Part I the Diurnal Changes " by Ma et al. We really appreciate that you take the time to handle this manuscript. We have received comments from three new reviewers, and you can see they are all relatively minor. In the revised manuscript, we have addressed all the comments and made corresponding changes/corrections.

The revisions to the manuscript were concentrated on three key areas: 1) incorporating a comparison of the estimated initial snow isotopic composition (Section 2.2.3 and Supplementary); 2) providing additional details in the determination of water vapor isotopic composition in the free atmosphere layer (Section 2.2.1); and 3) discussing the suitability of continuous simulations at Dome A (Section 4).

In addition, we have undertaken a thorough language revision of the manuscript and corrected other writing errors. We hope these efforts will further improve manuscript.

We confirm that this manuscript has not been published elsewhere and is not under consideration by another journal. All authors have approved the manuscript and agree with its submission to The Cryosphere. Please address all correspondence to genglei@ustc.edu.cn. We look forward to hearing from you at your earliest convenience.

Sincerely,

Lei Geng, Ph.D
Professor
School of Earth and Space Sciences
University of Science and Technology of China
96 Jinzhai Road, Hefei, Anhui, 230026
Email: genglei@ustc.edu.cn

**Response to Referee #1/Report #4's comments**

*1) Line 105: Think that this should be clouds vs cloud*

**Response:** Thanks, we used 'clouds' to replace 'cloud' in this sentence.

*2) Figure 1: why dotted lines on arrows in deposition column?*

**Response:** Thank you. During deposition, water vapor exchange between the boundary layer and free atmosphere only occurs when Ri < 0.1 (i.e., the unstable condition for atmospheric layer). In comparison, this process can consistently take place during the sublimation phase when the atmospheric conditions remain unstable throughout the daytime and warming periods. In order to highlight the difference, we used the dotted arrows with a label (Ri < 0.1) to indicate that this exchange only occurs when Ri < 0. To make it more clear, we have added an explanatory sentence in the title of Figure 1 in the revised manuscript.

*3) Line 177, Eqn 8, define variable S*

**Response:** S is the unit area. We have added a definition of S in the main text and supplementary.

*4) Line 304: The comparison of the estimation based on the Pang et al data should be presented, as should the estimation of the initial snow isotopic composition from the ECHAM5 model, given the sensitivity to this parameter.*

**Response:** We are really grateful to the reviewer for the rigorous considerations. The comparison between the estimation the of initial snow isotopic composition based on the Pang et al. (2019) data and the ECHAM5 model has already been addressed in our previous response to Reviewer #2. The results were shown in Figure 1 of this response. We put this figure into the supplementary and added one sentence to describe it in the main texts.

[Figure]

**Figure 1**. The estimated precipitation $\delta^{18}O$ and its standard deviation during the period of 2005-2011. Blue solid line with star marks represents the calculations using the temperature-isotope slope according to data from Pang et al. (2019), and the light blue shaded area is the uncertainties. Black solid line with x marks and light grey shaded area displays the ECHAM5-wiso simulation data and its uncertainties, respectively.

**End of Response to Referee #1**

**Response to Referee #2/Report #5's comments**

*1) Line 85, "box model." changes to 'box model'.*

**Response:** Thanks, the superfluous full stop was deleted from this sentence.

*2) Line 99-100, the authors indicate that at Dome A the time interval between two precipitation events can reach ~80 days. Please give a reference here.*

**Response:** Thank you. This statement is based on the ERA-5 dataset since there are no observations at Dome A. In the revised manuscript, we have stated the data sources at the end of this sentence. The details are follows:

*In addition, reanalysis data indicate that at Dome A the time interval between two precipitation events can reach ~ 80 days (estimated based on ERA5 reanalysis dataset), which means that snow can sit at the surface for a substantially long period before burial, and is subject to experience extensive atmosphere-snow water vapor exchange, which consequently affects the isotopic composition of the buried snow.*

*3) Line 235-237, the authors indicate that the isotopic composition of vapor in the free atmosphere layer ($\delta f0$) is greater than the isotopic composition of vapor in the boundary layer ($\delta v0$). This is due to the contribution from the free atmosphere can increase the ratio of H218O molecules in the boundary layer (Casado et al., 2018). Why the contribution from the free atmosphere can increase the ratio of H218O molecules in the boundary layer? The authors should explain it in more details.*

**Response:** Thanks for this question. Based on the vertical isotopic profiles ($\delta^{18}O$) observed at the summit of Greenland (Berkelhammer et al., 2016), it is noted that the isotopic composition of water vapor in the free atmosphere adjacent to the boundary layer is nearly equivalent to, or only slightly greater than, that within the boundary layer throughout the entire year (Figure 2 of this response). Although there are currently no vertical observations of water vapor isotopic composition in central Antarctica, Casado et al. (2018) assumed that this feature may be universally present in polar inland regions. With assumption, Casado et al. (2018) found that the water vapor and snow isotopic compositions under the influence of exchange with the free atmosphere at Dome C can be explained. In this study, we followed the same assumption and considered that the isotopic composition of water vapor in the free atmosphere is higher than that within the boundary layer at Dome A.

In the revised manuscript, we have made this more clear in Section 2.2.1:

*Here we expect that $\delta f0$ is greater than $\delta v0$. Although there are currently no vertical observations of water vapor isotopic composition in Antarctica, vertical isotopic profiles ($\delta^{18}O$) observed at the summit of Greenland have indicated that the isotopic composition of water vapor in the free atmosphere is slightly higher than that within the boundary layer (Berkelhammer et al., 2016). In order to explain the water vapor and snow isotope observations at Dome C, Casado et al. (2018) assumed that the contribution from the free atmosphere can increase the ratio of $H_2^{18}O$ molecules in the boundary layer (Casado et al., 2018) and set $\delta f0$ as the highest observed value of water vapor isotopic composition at Dome C.*

[Figure]

**Figure 2**. Vertical profiles of the isotopic ratio ($\delta^{18}O$) at Summit Camp, Greenland (Cited from Berkelhammer et al., 2016).

*4) Line 237, 'Casado et al. 2018' should be 'Casado et al., 2018'.*
**Response:** Thanks, correct.

*5) Line 410-411, the authors indicate that as δ18Os0 decreases, the magnitude of δ18Os diurnal changes decreases. But in Fig. 8e, we can see that as δ18Os0 decreases, the magnitude of δ18Os diurnal changes increases (not decrease). Is that right?*
**Response:** Thanks for pointing out this issue. We have overlooked the pattern and a comment by the reviewer, the correct trend is that the magnitude of $\delta^{18}O_s$ diurnal changes increases as the $\delta^{18}O_{s0}$ decreases. In the revised manuscript, we have corrected the description in this sentence.

*6) Line 426, 'Fig. 2, 4c, and 5c' should be 'Figs. 2, 4c, and 5c'.*
**Response:** Thanks, correct.

*7) Line 449, 'data from Dome A simulations' changes to 'data at Dome A'.*
**Response:** Correct.

*8) Line 526, 'by those in summer' changes to 'by those in winter'; 'snow isotopes.''' changes to 'snow isotopes.'.*
**Response:** Thanks, correct.

**End of Response to Referee #2**

**Response to Referee #3/Anonymous reviewer #2's comments**

*1) Figure 7: The authors kindly provided a continuous simulation at Dome A. However, this simulation looks considerably different from the continuous simulation at Dome C presented in Figure 3e. In particular, the large variations in the vapor isotopic composition seem unrealistic and I would like to ask the authors to check and explain what causes these large variations and comment on the appropriateness of using this vapor data for the Dome A simulations in the manuscript's discussion.*

**Response**: Thank you. The significant fluctuations observed in the modeled vapor isotopic composition at Dome A are primarily attributed to abrupt shifts in temperature measurements. Upon re-evaluation, we noted that substantial variations in water vapor isotopes can arise when temperature changes exceed 5°C within a given time interval (1 hour). Such marked temperature fluctuations can profoundly impact atmospheric conditions and the isotopic fractionation coefficient. Although these changes may result in only minor variations in the snow layer due to its large reservoir of water molecules, they can significantly affect the isotopic composition of water vapor. Furthermore, we did not separate clear-sky and cloudy conditions during continuous simulations at Dome A. This simplified parameterization is likely to enhance the variability of our calculated results. In the revised manuscript, we added the statement in the deficient of continuous simulation.

*However, it should be noted that the continuous simulation in this study was conducted without differentiating between clear-sky and cloudy conditions and was considerably affected by abrupt temperature fluctuations observed at Dome A. Therefore, further exploration of continuous simulations is required, which can be achieved through improvements in model refinement and the capabilities of observational techniques with more precise data available.*

*2) L. 331: Please correct to ECHAM5-wiso.*
**Response**: Thanks, correct.

*3) L. 468: As the authors suggested in response to reviewers, I suggest to reference Fig. 2, 4c, d and 5c, d, since Fig. 2, 4c, and 5c do not show isotopic values and, thus, do not sufficiently support the statement. The referenced Figures in the manuscript differ from the response to the reviewers.*
**Response:** Thanks for this advice. We agree that the relationship between meteorological factors and simulations in water isotopes needs to be demonstrated by 5 figures, namely Figs. 2, 4c, 4d, 5c, and 5d, instead of only the 3 figures mentioned in the original manuscript. We have made correspondingly revisions in the main text.

*4) L. 524: As mentioned in a previous reviewer's comment, Figures 4 and 5 show no general vapor depletion in summer. I suggest removing the second part of this sentence. The authors indicated in the response to the reviewers that they agree with removing this line, however, the lines are not removed in the uploaded manuscript.*
**Response:** Thanks, delete.

**End of Response to Referee #3**

**Reference**

Berkelhammer, M., Noone, D. C., Steen-Larsen, H. C., Bailey, A., Cox, C. J., O'Neill, M. S., Schneider, D., Steffen, K., and White, J. W. C.: Surface-atmosphere decoupling limits accumulation at Summit, Greenland, Science Advance, 2, e1501704, doi: 10.1126/sciadv.1501704, 2016.

Casado, M., Landais, A., Picard, G., Münch, T., Laepple, T., Stenni, B., et al.: Archival processes of the water stable isotope signal in East Antarctic ice cores, The Cryosphere, 12(5), 1745-1766, doi: 10.5194/tc-12-1745-2018, 2018.

Pang, H., Hou, S., Landais, A., Masson-Delmotte, V., Jouzel, J., Steen-Larsen, H. C.: Influence of Summer Sublimation on $\delta D$, $\delta 18O$, and $\delta 17O$ in Precipitation, East Antarctica, and Implications for Climate Reconstruction from Ice Cores, Journal of Geophysical Research: Atmospheres, 124(13), 7339-7358, doi: 10.1029/2018JD030218, 2019.

---

## Author Response (AR4)

Dear Dr. Smith,

Please find our revised manuscript "A Model Framework on Atmosphere-snow Water vapor Exchange and the Associated Isotope Effects at Dome Argus, Antarctica: Part I the Diurnal Changes " by Ma et al. We really appreciate that you take the time to handle this manuscript and give the comments/recommendations. In the revised manuscript, we have addressed all the comments and made corresponding changes/corrections.

The revisions to the manuscript were concentrated on the precision of numerical values. The inappropriate words or phrases have also been corrected following the suggestions. In addition, we have undertaken a thorough language revision of the manuscript and corrected other writing errors. We hope these efforts will further improve manuscript.

We confirm that this manuscript has not been published elsewhere and is not under consideration by another journal. All authors have approved the manuscript and agree with its submission to The Cryosphere. Please address all correspondence to genglei@ustc.edu.cn. We look forward to hearing from you at your earliest convenience.

Sincerely,

Lei Geng, Ph.D
Professor
School of Earth and Space Sciences
University of Science and Technology of China
96 Jinzhai Road, Hefei, Anhui, 230026
Email: genglei@ustc.edu.cn

**Response to Editor's Comments**

*1) L.26-27: Please check significant figures. Variabilities should be expressed to at most two-digit precision, values should be expressed to no more precision than the variability: thus,*
*4.75 +- 2.57 -> 4.8 +- 2.6*
*28.8 +- 19.06 -> 29 +- 19*
*Please fix this throughout the manuscript. I will mention it where I see it, but will likely not see everything.*

**Response:** Thanks for this suggestion. We have examined all numerical values throughout the manuscript and have revised their precision accordingly. The specific modifications and their corresponding lines in the manuscript are as follows:

*Line 26: 4.75 +- 2.57 -> 4.8 +- 2.6, 28.8 +- 19.06 -> 29 +- 19*
*Line 27: 1.64 +- 2.71 -> 1.6 +- 2.7*
*Line 90: 1-1.5 ->1.0-1.5*
*Line 104: 1.99 ->2.0*
*Line 277: 10.38 ->10, -31.01 ->-31*
*Line 278: -38.69 ->-39, -27.67 ->-28*
*Line 281: 2.98 ->3.0, 3.34 ->3.3*
*Line 346: 2 ->2.0*
*Line 369: 4.75 ->4.8, 28.75->29, 9.25->9.3*
*Line 371: 1.64 ->1.6, 4.85->4.9*
*Line 372: 2.35 ->2.4, 15.67->16, 3.13->3.1*
*Line 373: 1.09 ->1.1, 1.26->1.3*
*Line 382: 3.00 ->3.0, 21.15->21, 4.02->4.0*
*Line 383: 2.21 ->2.2*
*Line 461: 28.78 +- 19.06 ->29 +- 19*

*2)L.91: By 2015/2016 -> "By the 2015/2016"*
**Response:** Corrected.

*3)L.95: "processing, especially" -> "processing. In particular,"*
**Response:** Corrected.

*4)L.96: add comma after important*
**Response:** Thanks, added.

*5)L.101: delete "experience"*
**Response:** Corrected.

*6)L.104: "found on average" ->"found that on average"*
**Response:** Corrected.

*7)L.105: 1.99 -> 2.0*
**Response:** See details in the Response to Comment 1).

*8)L.123: "i.e." – ":"*
**Response:** Corrected.

*9)L.144: Ls should be L-sub-s. One of the referees found this in the last review! Please check that all edits suggested by referees were actually made.*
**Response:** Sorry for our careless. This mistake has been corrected. We also have reviewed all the comments provided by the referees and are confident that no issues have been overlooked in the revised manuscript.

*10) L.260: "across different cases": Please give an example or two of the kind of cases compared.*
**Response:** In Section 3.2.4 and Figure 7, we have shown the results of all six cases for continuous simulations at Dome A. To make it clear in the section of method, the reminder for readers has been added in the end of this phrase.

*11)L.261: "can be" -> were*
**Response:** Corrected.

*12)L.277-278: Check significant figures.*
**Response:** See details in the Response to Comment 1).

*13)L.285: "seeing—" -> "visual observations of"*
**Response:** Thanks, corrected.

*14)L.287: "mm w. eq. y." -> mm yr-1*
**Response:** Corrected.

*15) L.305: do you mean negative values of the latent heat flux?*
**Response:** Yes. In order to avoid ambiguity, we deleted the word of 'calculated' in front of 'latent heat flux'.

*16)L.305-06: "within diurnal variations in the wintertime": Please be more specific about the sign of the variations as it relates to the sign of the heat flux (if that is what is under discussion here)*
**Response:** We rephrased the paragraph to emphasize that the latent heat fluxes in Dome A winter are persistently negative and exhibit stable diurnal fluctuations. The revised paragraph is as follows:
*The stacked hourly mean values of winter meteorological conditions at Dome A were extracted in the same way as we did for the summer conditions. As shown in Fig. 2c, the average temperature, specific humidity, and atmospheric pressure are lower than those in summer, but the relative humidity increases during winter. These changes result*

*in the negative values of latent heat flux during winter. In addition, the winter meteorological parameters and latent heat flux do not show any apparent diurnal variations.*

*17)L.340-341: increased-> increases, decreased -> decreases*
**Response:** Thanks, corrected.

*18)L.342: Please explain what variables are being correlated here.*
**Response:** The correlations between water vapor $\delta^{18}O$ and $\delta D$ are not clearly described in L342. We rephrased this sentence as follows:
*The diurnal patterns in water vapor $\delta D$ are similar to that in water vapor $\delta^{18}O$ and their max-min difference is ~54‰ (Fig. 3b).*

*19)L.370-374: check significant figures*
**Response:** See details in the Response to Comment 1).

*20)L.390: "comparison with " -> "contrast to"*
**Response:** Corrected.

*21)L.392: "unchanged" -> constant"*
**Response:** Corrected.

*22)L.400: "However" -> "In addition,"*
**Response:** Corrected.

*23)L.402: "its" -> their*
**Response:** Corrected.

*24)L.403: Please specify what tendency. Also, please consider whether tendency or trend is the more appropriate word.*
**Response:** Thanks for this suggestion. We used the 'diurnal cycles' to specifically describe the pattern of continuous simulations at Dome A. The revised sentence is as follows:
*The diurnal cycles shown in the Dome A continuous simulations are consistent with the simulated results at Dome C.*

*25)L.407-422: Please remind the reader what parameters were tested in each group of tests. This should be done in the topic sentence for each paragraph.*
**Response:** We appreciate this valuable suggestion from the editor. In the Section 3.3, we have revised the topic sentence of each paragraph to explicitly state the parameters that were evaluated in the respective groups of tests.

*26)L.408: Provide a reminder for what H0 and h0 represent*
**Response:** To remind the reader, we used the full description of $H_0$ and $h_0$ in this

sentence. The details are as follows:0

*In the first group of tests (Fig. 8a), the magnitude of the diurnal variations in water vapor $\delta^{18}O$ ($\delta^{18}O_v$) is highly influenced by the boundary height ($H_0$) but not by snow depth ($h_0$).*

*27)L.425: delete "two"*
**Response:** Thanks, corrected.

*28)L.441: add "with" after "vapor reservoir"*
**Response:** Added.

*29)L.452: Please specify which two input parameters*
**Response:** In the revised manuscript, we rephrased the sentence as follows: *However, in the model employed for this study, the boundary layer height ($H_0$) and water vapor isotopic composition in the free atmosphere layer ($\delta f_0$) are maintained as constants to simplify the calculations, whereas they vary daily in reality.*

*30)L.460: Check significant figures*
**Response:** See details in the Response to Comment 1).

*31)L.464: orientation-> direction*
**Response:** Thanks, corrected.

*32)L.469: Please explain what calibration might be drifting*
**Response:** As stated in Liu et al., (2022), the extremely low water vapor content at Dome A is a crucial reason for calibration drifting. Here we made this point very clear in the sentence as follows:
*This large discrepancy may be due to calibration drifts caused by the low water vapor content during the measurements at the nearest Dome A site (Liu et al., 2022).*

*33)L.475: "only variable": do you mean "only" or "most significant"?*
**Response:** The phrase of 'most significant' is more accurate. Thus, we made a revision in this sentence.

*34)L.476: join the two sentences, replacing "with the presence of clouds" with "and, ".*
**Response:** Corrected.

*35)L.506: replace 2 occurrences of "would" with "should"*
**Response:** Corrected.

*36)L.511: "in terms of evaluating" -> "To evaluate"*
**Response:** Corrected.

*37)L.513: would -> should*
**Response:** Thanks, corrected.

*38)L.514 "it should be noted" -> "we note"*
**Response:** Corrected.

*39)L.533: "not comparable to (i.e. lower than)" -> "smaller than"*
**Response:** Corrected.

*40)L.535: delete "wanted to"*
**Response:** Thanks, corrected.

*41)L.539: "On the other hand" -> "Further,"*
**Response:** Corrected.

**End of Response**